# ABKD: Pursuing a Proper Allocation of the Probability Mass in Knowledge Distillation via $\alpha$-$\beta$-Divergence

**Guanghui Wang** [1]  **Zhiyong Yang** [1]  **Zitai Wang** [2]  **Shi Wang** [2]  **Qianqian Xu** [2]  **Qingming Huang** [1 3 2]

## Abstract

Knowledge Distillation (KD) transfers knowledge from a large teacher model to a smaller student model by minimizing the divergence between their output distributions, typically using forward Kullback-Leibler divergence (FKLD) or reverse KLD (RKLD). It has become an effective training paradigm due to the broader supervision information provided by the teacher distribution compared to one-hot labels. We identify that the core challenge in KD lies in balancing two mode-concentration effects: the ***Hardness-Concentration*** effect, which refers to focusing on modes with large errors, and the ***Confidence-Concentration*** effect, which refers to focusing on modes with high student confidence. Through an analysis of how probabilities are reassigned during gradient updates, we observe that these two effects are entangled in FKLD and RKLD, but in extreme forms. Specifically, both are too weak in FKLD, causing the student to fail to concentrate on the target class. In contrast, both are too strong in RKLD, causing the student to overly emphasize the target class while ignoring the broader distributional information from the teacher. To address this imbalance, we propose ABKD, a generic framework with $\alpha$-$\beta$-divergence. Our theoretical results show that ABKD offers a smooth interpolation between FKLD and RKLD, achieving an effective trade-off between these effects. Extensive experiments on 17 language/vision datasets with 12 teacher-student settings confirm

its efficacy. The code is available at https://github.com/ghwang-s/abkd.

## 1. Introduction

Knowledge Distillation (KD) (Hinton, 2015) is a widely-adopted technique for transferring knowledge from large models (teachers) to smaller models (students). In this setup, the student model, with a predictive distribution $q_\theta$, learns to mimic the predictive distribution $p$ of the teacher model. This imitation is typically achieved by minimizing a predefined divergence $\mathbb{D}$ between the teacher distribution $p$ and the student distribution $q_\theta$: $\ell_{\text{KD}} \triangleq \mathbb{D}(p \| q_\theta)$. This way, KD allows the student to leverage richer soft label information from $p$ compared to one-hot labels, often leading to better performance than traditional supervised fine-tuning. This has been shown in tasks like image classification (Dosovitskiy, 2020; Radford et al., 2021; Yang et al., 2023b; Wang et al., 2022b) and text generation (Vaswani, 2017; Touvron et al., 2023a).

A key step in KD is to choose a proper divergence $\mathbb{D}$ for distribution matching. One popular choice in previous works (Cho & Hariharan, 2019; Mirzadeh et al., 2020; Zhou et al., 2021; Zhao et al., 2022; Jin et al., 2023; Sun et al., 2024; Zheng & Yang, 2024) is the *forward* Kullback-Leibler divergence (FKLD). However, FKLD's asymmetry often results in a student distribution $q_\theta$ that is overly smooth, spreading across the entire support of $p$. To address this, recent studies (Lee et al., 2023; Gu et al., 2024a; Kim et al., 2024; Gu et al., 2024b) have explored the *reverse* KLD (RKLD), which allows $q_\theta$ to focus on a few prominent modes of $p$. Despite the effectiveness, empirical results (Wen et al., 2023; Wu et al., 2024; Ko et al., 2024) suggest that RKLD often yields suboptimal performance across a range of tasks. What is worse, there is no systematic approach to identify the essential issues hidden behind, which hinders the development of a more generic and effective KD framework. To get out of this dilemma, we first pose the following question:

> *What underlying factors contribute to the suboptimal performance of FKLD and RKLD?*

To answer this, we analyze how different divergence func-

[1] School of Computer Science and Technology, University of Chinese Academy of Sciences, Beijing, China [2] Key Laboratory of Intelligent Information Processing, Institute of Computing Technology, Chinese Academy of Sciences, Beijing, China [3] Key Laboratory of Big Data Mining and Knowledge Management (BDKM), University of Chinese Academy of Sciences, Beijing, China. Correspondence to: Zhiyong Yang <yangzhiyong21@ucas.ac.cn>, Qingming Huang <qmhuang@ucas.ac.cn>.

*Proceedings of the $42^{nd}$ International Conference on Machine Learning*, Vancouver, Canada. PMLR 267, 2025. Copyright 2025 by the author(s).

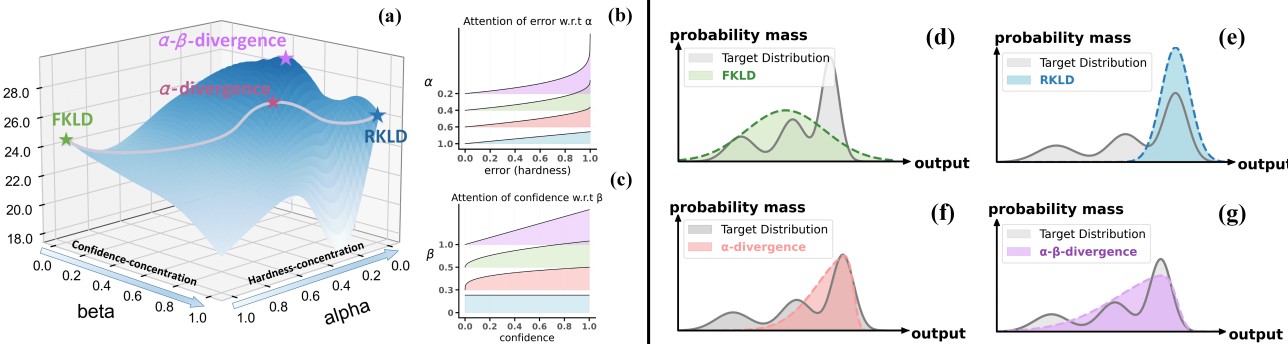

*Figure 1.* **(a)** Illustration of the unified search space for our proposed ABKD, where height (color) represents performance ($\uparrow$). The FKLD and RKLD are special cases of ABKD when selecting $(\alpha = 1, \beta = 0)$ and $(\alpha = 0, \beta = 1)$, respectively. The $\alpha$-divergence can only search along the submanifold $\alpha + \beta = 1$ in the ABKD space. **(b)-(c)** illustrate how adjusting $\alpha$ and $\beta$ affects hardness-concentration and confidence-concentration. **(d)-(g)** illustrate how different divergences learn a student distribution from the given teacher distribution. The $\alpha$-$\beta$-divergence, compared to others, can more effectively learn soft label information while maintaining focus on the target class.

tions affect the allocation of probability mass in the student distribution during training by tracking the log mass ratio LogR. Notably, LogR is proportional to the gradient of the loss function *w.r.t* the logits. This insight allows us to frame the problem as understanding how divergence algorithms influence the reduction of LogR. Through this lens, we identify two key mode-concentration effects: ***Hardness-Concentration*** and ***Confidence-Concentration***. ***Hardness-Concentration*** refers to focusing on modes in the loss where there is a large error between $p$ and $q_\theta$, while ***Confidence-Concentration*** refers to focusing on modes in the loss where $q_\theta$ has high confidence.

On top of this, we find that the limitations of FKLD and RKLD stem from the extreme ways they utilize these concentration effects: a) FKLD exhibits **weak** concentration effects, treating mismatches equally from all classes, which fails in guiding the student to concentrate on the target class and causes incorrect predictions (Fig. 1d). b) RKLD exhibits **strong** concentration effects, focusing on both hard classes with large errors and classes where the student has high confidence. This often leads to a trivial solution, where the well-trained student focuses exclusively on the target class and ignores broader knowledge from $p$ (Fig. 1e). With the limitations revealed, we continue to seek an answer to the following question:

> *Can we find a generic, theoretically grounded method to balance hardness-concentration and confidence-concentration?*

In pursuit of this, we introduce the $\alpha$-$\beta$-divergence, a general extension of divergences that unifies FKLD and RKLD, while also extending to previously unexplored divergences like the Hellinger distance and $\beta$-divergence. Our theoretical results demonstrate that the $\alpha$-$\beta$-divergence provides

a flexible mechanism to smoothly interpolate between the extremes of FKLD and RKLD by controlling the trade-off between hardness-concentration (Fig. 1b) and confidence-concentration (Fig. 1c) via the hyperparameters $\alpha$ and $\beta$. This mechanism ensures a more proper allocation of probability mass (Fig. 1g). Motivated by these insights, we propose ABKD, a generic distillation framework based on $\alpha$-$\beta$-divergence. Empirical results across a variety of tasks, including instruction-following and image classification, demonstrate ABKD's generality and effectiveness. For instance, by modifying only the loss function, ABKD achieves performance improvements of 0.81 to 3.31 over FKLD and RKLD on five instruction-response datasets when distilling *GPT-2 XL* (1.5B) into *GPT-2* (0.1B).

In summary, the contributions of this work are three-fold:

- **Theoretically**: We analyze the limitations of FKLD and RKLD from novel perspectives of hardness-concentration and confidence-concentration, and show that the $\alpha$-$\beta$-divergence offers a flexible approach to balance these effects.

- **Methodologically**: We propose ABKD, a flexible distillation framework that unifies FKLD and RKLD and generalizes to several other divergences, offering greater versatility and applicability.

- **Empirically**: Extensive experiments on 17 language and vision datasets with 12 teacher-student configurations (0.85M-0.46M to 7B-3B) validate the theoretical insights. ABKD outperforms or matches state-of-the-art methods *without extra trainable parameters* and allows further gains by rectifying their loss functions.

**Prior Arts.** We discuss related work and defer a concentrated account to App. A.

## 2. Preliminaries

KD involves using a *fixed teacher model* $f_T$ to improve the performance of a *parameterized student model* $f_S$. Given an input $\boldsymbol{x}$, the teacher $f_T$ and student $f_S$ produce probability distributions $p$ and $q_\theta$, respectively.

The goal of KD can be achieved by letting $q_\theta$ mimic $p$ for all samples in dataset $\mathcal{D}$. A direct way to do this is minimizing:

$$\ell_{\text{KD}} \triangleq \mathbb{D}(p\|q_\theta), \tag{1}$$

where $\mathbb{D}$ is a distribution measure. Optionally, practitioners can substitute $p$ with the one-hot vector $\boldsymbol{y}$, where $\boldsymbol{y} \triangleq [0, \ldots, 1, \ldots, 0]$ with 1 at ground-truth label $y$ and 0 elsewhere. In this case, the loss is $\ell_{\text{CE}} \triangleq \mathbb{D}(\boldsymbol{y}\|q_\theta)$, where $\mathbb{D}$ is typically the FKLD. The final training loss is:

$$\ell = \ell_{\text{CE}} + \lambda \ell_{\text{KD}}, \tag{2}$$

where $\lambda$ is a hyperparameter. Since $p$ provides richer information (*i.e.*, soft label) than the one-hot vector $\boldsymbol{y}$, KD outperforms traditional supervised fine-tuning on many downstream tasks, such as instruction-following and image classification. The settings for these tasks in KD are as follows.

**Instruction-following**. Let $\boldsymbol{x}$ and $\boldsymbol{y}$ represent the input and output sequences, respectively. A token-level autoregressive model produces an $C$-dimensional probability distribution for the $n$-th token over the vocabulary $\mathbb{V}$, conditioned on $\boldsymbol{x}$ and $\boldsymbol{y}_{<n}$, where $\boldsymbol{y}_{<n} \triangleq (y_1, y_2, \ldots, y_{n-1})$ denote the generated output sequence up to the $(n-1)$-th token, The discrepancy between token-level distributions of $p$ and $q_\theta$ is defined as $\mathbb{D}(p\|q_\theta) \triangleq \frac{1}{L_y} \sum_{n=1}^{L_y} \mathbb{D}(p(\cdot \mid \boldsymbol{y}_{<n}, \boldsymbol{x})\|q_\theta(\cdot \mid \boldsymbol{y}_{<n}, \boldsymbol{x}))$, where $L_y$ denotes the sequence length.

**Image classification**. Let $\boldsymbol{x} \in \mathbb{R}^{H \times W}$ be an image and $\boldsymbol{y} \in \mathbb{R}^C$ its one-hot label, with $H$, $W$, and $C$ representing the image dimensions and number of classes. A vision model produces a $C$-dimensional probability distribution conditioned on $\boldsymbol{x}$. The discrepancy between $p$ and $q_\theta$ is defined as $\mathbb{D}(p\|q_\theta)$.

## 3. The Limitations of FKLD and RKLD

Prior arts primarily use **FKLD** $\mathbb{D}_{\text{KL}}(p\|q_\theta)$ or **RKLD** $\mathbb{D}_{\text{KL}}(q_\theta\|p)$ to measure distribution discrepancy:

$$\mathbb{D}_{\text{KL}}(p\|q_\theta) = \sum_k p(k) \log \frac{p(k)}{q_\theta(k)}, \tag{3}$$

$$\mathbb{D}_{\text{KL}}(q_\theta\|p) = \sum_k q_\theta(k) \log \frac{q_\theta(k)}{p(k)}. \tag{4}$$

Despite promising success, recent empirical studies show that these two divergences cause suboptimal performance (Wen et al., 2023; Ko et al., 2024; Wu et al., 2024). Next,

we uncover the underlying factors that contribute to the limitations of FKLD and RKLD by tracking how they allocate probability mass in the student distribution during gradient updates. These insights will guide us in Sec. 4 to identify a more suitable divergence for KD.

### 3.1. Tracking Probability Allocation with Log Mass Ratio

To find a proper probability matching scheme, KD algorithms must keep allocating the probability mass of the student distribution during training. The key in our theory is to keep track of the probability mass change in each gradient update step. To do this, we define a monitoring quantity called **log mass ratio** inspired by Tajwar et al. (2024):

$$\text{LogR}_t^{\mathcal{A}}(y) \triangleq \log \left( \frac{q_{t+1}^{\mathcal{A}}(y)}{q_t(y)} \right),$$

where $q_{t+1}^{\mathcal{A}}(y)$ is the probability mass for class $y$ obtained from algorithm $\mathcal{A}$ at step $t+1$; $q_t(y)$ is original probability mass for class y at step $t$.

To define probability mass $q_t$, we follow the most popular convention that the class probability is approximated by a softmax function such that: $q_t(y) \propto \exp(f_y^t)$, where $f_y^t$ is the logit for the $y$-class channel at step $t$. Interestingly, one can show that (see App. C.1) $\text{LogR}_t^{\mathcal{A}}(y)$ is proportional to the gradient of the logit $f_y^t$ as follows:

$$\text{LogR}_t^{\mathcal{A}}(y) = -\eta \cdot \nabla_{f_y^t} \ell + \text{N}_t^{\mathcal{A}}(y), \tag{5}$$

where $\eta$ denotes the learning rate and $\text{N}_t^{\mathcal{A}}(y)$ is a normalizing factor independent of y, which vanishes to zero when all the class channel gradients $\nabla_{f_y} \ell$ vanish. Note that under a mild assumption, we can show that (see App. C.2) when $\nabla_{\boldsymbol{W}} \ell$, the overall gradient *w.r.t.* the model weights $\boldsymbol{W}$, goes to zero, $\nabla_{f_y} \ell$ also goes to zero. In this sense, to reach a local minimum, the algorithm automatically reduces the magnitude of $\nabla_{f_y} \ell$, and also the $|\text{LogR}_t^{\mathcal{A}}(y)|$.

Based on the discussion above, **we next show how reducing $|\text{LogR}_t^{\mathcal{A}}(y)|$ in different divergences affects hardness-concentration and confidence-concentration**.

### 3.2. FKLD and RKLD as Two Extreme Cases

First, we have the following upper bounds for the log mass ratio for FKLD (Eq. 3) and RKLD (Eq. 4).

**Proposition 3.1.** *The updates induced by FKLD and RKLD for $q_t$ within one gradient descent step are given by:*

*FKLD:* $\quad \left| \text{LogR}_t^{\mathcal{F}}(y) \right| \leq \eta \cdot \underbrace{1}_{(a)} \cdot \underbrace{\left| p(y) - q_t(y) \right|}_{(b)} + \left| \text{N}_t^{\mathcal{F}}(y) \right|,$

*RKLD:*

$$\left|\mathsf{LogR}_t^{\mathcal{R}}(y)\right| \leq \eta \cdot \underbrace{q_t(y)}_{(a_1)} \left( \underbrace{\left|\log p(y) - \log q_t(y)\right|}_{(b_1)} \right.$$

$$\left. + \sum_k \underbrace{q_t(k)}_{(a_2)} \underbrace{\left|\log p(k) - \log q_t(k)\right|}_{(b_2)} \right) + \left|\mathsf{N}_t^{\mathcal{R}}(y)\right|,$$

*where $\mathsf{N}_t^{\mathcal{F}}(y)$ and $\mathsf{N}_t^{\mathcal{R}}(y)$ denote constant normalization factors independent of $y$ and vanish to zero when $p = q_t$.*

This proposition is inspired by (Tajwar et al., 2024), but our work distinguishes itself by shifting the focus towards explicitly identifying the factors that drive the decrease in $\left|\mathsf{LogR}_t^{\mathcal{A}}(y)\right|$. The proof is in App. G.1. By **minimizing** the log mass ratio, there are two types of effects hidden in the results. We first analyze their **independent** roles.

1. The first type, represented by terms $(b)$, $(b_1)$, and $(b_2)$, has the general form of $|s(p(k)) - s(q_t(k))|$, which measures the matching loss between student and teacher distribution. If considered independently, such terms control the effect of **hardness-concentration**. More precisely, a **sharper** term with a larger rate-of-change corresponds to an **aggressive** student who aims to focus on the **hardest** classes to reach a good matching performance *w.r.t.* the teacher (Fig. 1b).

2. The second type, denoted by terms $(a)$, $(a_1)$, $(a_2)$, can be expressed as the student's confidence weighting function: $q_t(y)^\beta$ ($\beta \geq 0$). If considered independently, such terms control the effect of **confidence-concentration**. In other words, a sharper weighting function corresponds to a **confident** student who only cares about the matching performance in classes that the student believes to be the ground truth (Fig. 1c).

What is the **joint** effect of the two? Prop. 3.1 provides two extreme answers, FKLD and RKLD. **FKLD** in the results picks a very weak hardness-concentration effect with $s(x) = x$ and a very weak confidence-concentration effect with $\beta = 0$. As a result, FKLD forces the student to treat all matching penalties equally for all classes since there is no weighting function ($(a) = 1$). This **fails to concentrate** on the target classes. By contrast, RKLD picks a very strong hardness-concentration effect with $s(x) = \log(x)$ (recall that $0 < x < 1$, log is much sharper than linear function) and a very strong confidence-concentration effect with $\beta = 1$. Recall that a well-trained student distribution $q_t$ primarily concentrates probability on the target class and assigns smaller probabilities to others. In this case, an overly strong confidence-concentration effect suppresses the hardness-concentration effect on non-target classes while emphasizing this effect on the target class. This results in

a trivial solution: the student **concentrates solely on the target class** and neglects the overall matching effect.

The following theorem makes the above intuition more rigorous. The results suggest an **asymmetric** mass allocation for RKLD and an **equally** important allocation for FKLD. Due to the space limit, the readers are referred to App. G.2 for a formal expression and the proof.

**Theorem 3.2 (Informal).** *Given the student distribution $q_\theta$ and teacher distribution $p$, FKLD and RKLD differ as follows within one gradient update:*

1. *FKLD allocates the mass across all classes equally.*

2. *RKLD preferentially increases the mass of underestimated $(p(y) > q_\theta(y))$ classes with higher $q_\theta(y)$.*

3. *RKLD preferentially reduces the mass of overestimated $(p(y) < q_\theta(y))$ classes with smaller $q_\theta(y)$.*

As shown in Fig. 1(d), the equally weighted matching scheme in FKLD drives students to sub-optimal modes, which induces wrong predictions. For RKLD, the theorem states that it only favors small mass classes when the teacher score is over-estimated while only favors large mass classes under the opposite scenario. As a total effect, **the small mass tends to get smaller; the large mass tends to get larger**. As an extreme result shown in Fig. 1(e), RKLD eventually forces the student to focus on one class. This makes the teacher's supervision degenerate to a ont-hot label, which loses the distributional information hidden inside the teacher's prediction. This leads to the following conclusion:

> *A proper divergence should achieve a moderate trade-off between hardness-concentration and confidence-concentration.*

### 3.3. Weighted Sum of FKLD and RKLD

In pursuit of this, a naive solution is to take a weighted sum of FKLD and RKLD, which we call the weighted sum divergence (WSD):

$$\mathbb{D}_{\mathrm{WSD}}(p\|q) \triangleq \alpha \mathbb{D}_{\mathrm{KL}}(p\|q) + \beta \mathbb{D}_{\mathrm{KL}}(q\|p), \tag{6}$$

where $\alpha$ and $\beta$ are hyperparameters. A more principled approach is to adapt the weighting coefficients dynamically during training based on the discrepancy (*e.g.*, entropy) between $p$ and $q$, as done in previous works (Amara et al., 2022; Wu et al., 2024).

Unfortunately, such a composite metric overemphasizes modes with small probabilities in $p$ and $q$. To see this, when either $q(k) \approx 0, p(k) > 0$ or $p(k) \approx 0, q(k) > 0$, we have

$\mathbb{D}_{\text{WSD}}(p\|q) \to \infty$. Hence, the algorithm must focus on extreme cases to minimize the objective function, leading to improper probability allocation. Moreover, similar to the analysis in Ko et al. (2024), one can easily show that the gradient norm in this case also grows excessively, leading to significant and potentially noisy parameter updates. Such behaviors can destabilize the optimization process and hinder convergence.

Another attractive approach is to use the Jensen-Shannon divergence (Binici et al., 2022; Agarwal et al., 2024) $\mathbb{D}_{\text{JSD}}(p\|q) \triangleq \frac{1}{2}\mathbb{D}_{\text{KL}}\left(p\|m\right) + \frac{1}{2}\mathbb{D}_{\text{KL}}\left(q\|m\right)$, where $m = \frac{1}{2}(p+q)$. However, a major drawback of JSD is that it suffers from gradient vanishing (Arjovsky et al., 2017) when the distributions $p$ and $q_\theta$ are far apart (a common scenario in early training stages), which hinders model convergence.

Above all, balancing hardness and confidence concentration is non-trivial if one only resorts to FKLD and RKLD. In the next section, we will introduce a generic notion of divergence to address this issue.

# 4. ABKD: The Proposed Method

## 4.1. ABKD

One way to pursue a harmonic utilization of hardness- and confidence-concentration is to find a subtle point between FKLD and RKLD. The following $\alpha$-$\beta$-divergence exactly serves this purpose (Cichocki et al., 2011).

**Definition 4.1** ($\alpha$-$\beta$-divergvence). *Consider $\alpha$ and $\beta \in \mathbb{R}$, satisfying $\alpha, \beta, \alpha + \beta \neq 0$. the $\alpha$-$\beta$-divergence of two distributions is given by:*

$$\mathbb{D}_{\text{AB}}^{(\alpha,\beta)}(p \parallel q) \triangleq -\frac{1}{\alpha\beta} \sum_k \left[ p(k)^\alpha q(k)^\beta - \frac{\alpha}{\alpha+\beta} p(k)^{\alpha+\beta} \right.$$
$$\left. - \frac{\beta}{\alpha+\beta} q(k)^{\alpha+\beta} \right],$$

*where $p = [p(k)]_{k=1}^C$ and $q = [q(k)]_{k=1}^C$ are two discrete distributions over $C$ classes.*

As will soon be seen in Sec.4.2, both hardness-concentration and confidence-concentration effects in $\alpha$-$\beta$-divergence could be regarded as an interpolation between the corresponding effect of FKLD and RKLD. Such ability allows the $\alpha$-$\beta$-divergence to ensure a more proper allocation of probability mass.

Inspired by this, we propose ABKD, which is formally defined as minimizing the following objective:

$$\ell = \ell_{\text{CE}} + \lambda\mathbb{D}_{\text{AB}}^{(\alpha,\beta)}(p \parallel q_\theta), \qquad (7)$$

Beyond this issue, $\alpha$-$\beta$-divergence is **also a generic notion**

*Table 1.* Some divergence functions and their corresponding choices of $\alpha$ and $\beta$. The $\alpha$-$\beta$-divergence can be extended by continuity (by applying l'Hôpital formula) to cover all the values of $\alpha, \beta \in \mathbb{R}$, as shown in App. B.

| Distribution Measure | Reference | Range |
|---|---|---|
| Kullback–Leibler (KL) divergence | Kullback & Leibler (1951) | $\alpha = 1, \beta = 0$ |
| Reverse KL divergence | Kullback & Leibler (1951) | $\alpha = 0, \beta = 1$ |
| $\alpha$-divergence | Chernoff (1952) | $\alpha + \beta = 1$ |
| $\beta$-divergence | Basu et al. (1998) | $\alpha = 1$ |
| Hellinger distance | Hellinger (1909) | $\alpha = \beta = 0.5$ |
| Squared euclidean distance | Heath (1956) | $\alpha = \beta = 1$ |

of a family distribution divergences, which includes FKLD, RKLD, and other typical divergences as special cases. For example, when ($\alpha = 1, \beta = 0$), one obtains FKLD; when ($\alpha = 0, \beta = 1$), one obtains RKLD. Please see Tab. 1 for other special cases. In this way, ABKD nature provides a generic framework for divergence-based distillation algorithms.

## 4.2. Trading off Hardness-Concentration and Confidence-Concentration via $\alpha$-$\beta$-divergence

ABKD offers a unified space to trade off the hardness-concentration and confidence-concentration effects.

To explain this, we go back to the log mass ratio, the following proposition explains how the hyperparameters $\alpha$ and $\beta$ influence the reduction of $|\text{LogR}_t^{(\alpha,\beta)}(y)|$.

**Proposition 4.2.** *The updates induced by $\alpha$-$\beta$-divergence for $q_t$ within one gradient descent step are given by:*

$$\left|\text{LogR}_t^{(\alpha,\beta)}(y)\right| \leq \eta \underbrace{q_t(y)^\beta}_{(a)} \underbrace{\left|\frac{p(y)^\alpha - q_t(y)^\alpha}{\alpha}\right|}_{(b)}$$
$$+ \eta q_t(y) \sum_k \underbrace{q_t(k)^\beta}_{(a_1)} \underbrace{\left|\frac{p(k)^\alpha - q_t(k)^\alpha}{\alpha}\right|}_{(b_1)} + \left|\text{N}_t^{(\alpha,\beta)}(y)\right|,$$

*where $\text{N}_t^{\alpha,\beta}(y)$ denotes constant normalization factor independent of $y$ and vanishes to zero when $p = q_t$.*

The proof is in App. G.5. In $(a)$ and $(a_1)$, $\alpha$-$\beta$-divergence employs a power form $q_t(k)^\beta$ for **confidence-concentration** effect. It is easy to see when $\beta \to 1$, it degenerates to the effect of RKLD, and when $\beta \to 0$ to the effect of FKLD. A larger $\beta$ provides a stronger effect of confidence-concentration, focusing the matching performance on its most confident classes (Fig. 1c). Meanwhile, terms $(b)$ and $(b_1)$ uses $|\frac{p(y)^\alpha - q_t(y)^\alpha}{\alpha}|$ for **hardness-concentration** effect. It is easy to see when $\alpha \to 1$, it degenerates to the effect of FKLD, and when $\alpha \to 0$ to the effect of RKLD. A smaller $\alpha$ amplifies the hardness-concentration

*Table 2.* ROUGE-L scores (↑) on five task-agostic instruction-following datasets. Note that ***this is an unfair comparison*** because we only train on the fixed dataset while other KD methods employ augmentation. The ***fairer results*** using our method with different augmentation strategies can be found in Fig. 3(b) and Tab. 8. All results are based on our re-implementation. We report the average and standard deviation of ROUGE-L scores across five random seeds. Better results are shown in bold, and darker colors indicate superior performance.

| Method | Dolly Eval | Self-Instruct | Vicuna Eval | Super-Natural | Unnatural |
|---|---|---|---|---|---|
| GPT-2 XL (Teacher) | 26.94 (0.23) | 13.31 (0.63) | 16.23 (0.62) | 24.28 (0.43) | 29.05 (0.14) |
| *GPT-2 XL (1.5B) → GPT-2 (0.1B)* | | | | | |
| SFT | 23.14 (0.23) | 10.22 (0.44) | 15.15 (0.31) | 17.41 (0.18) | 19.76 (0.09) |
| KD (Hinton, 2015) | 23.80 (0.37) | 10.01 (0.75) | 15.25 (0.65) | 17.69 (0.26) | 18.99 (0.05) |
| SeqKD (Kim & Rush, 2016) | 24.28 (0.22) | 11.24 (0.30) | 14.94 (0.58) | 20.66 (0.28) | 23.59 (0.13) |
| MiniLLM (Gu et al., 2024a) | 24.62 (0.33) | 12.49 (0.56) | **17.30** (0.41) | 23.76 (0.38) | 24.30 (0.14) |
| GKD (Agarwal et al., 2024) | 24.49 (0.16) | 11.41 (0.14) | 16.01 (0.37) | 18.25 (0.24) | 21.41 (0.11) |
| DISTILLM (Ko et al., 2024) | 25.32 (0.14) | 11.65 (0.28) | 16.76 (0.66) | 23.52 (0.47) | 25.79 (0.08) |
| Ours (ABKD) | **25.65** (0.24) | **13.47** (0.42) | 16.06 (0.25) | **26.47** (0.31) | **29.32** (0.08) |
| *GPT-2 XL (1.5B) → GPT-2 Medium (0.3B)* | | | | | |
| SFT | 25.30 (0.31) | 12.56 (0.62) | 16.36 (0.22) | 23.32 (0.13) | 23.42 (0.07) |
| KD (Hinton, 2015) | 24.71 (0.17) | 10.33 (0.54) | 16.23 (0.50) | 23.74 (0.32) | 23.97 (0.12) |
| SeqKD (Kim & Rush, 2016) | 25.93 (0.44) | 12.98 (0.24) | 16.68 (0.30) | 21.95 (0.19) | 25.23 (0.08) |
| MiniLLM (Gu et al., 2024a) | 25.34 (0.25) | 13.36 (0.62) | **17.25** (0.46) | 25.68 (0.41) | 26.63 (0.12) |
| GKD (Agarwal et al., 2024) | 24.75 (0.27) | 12.76 (0.85) | 16.54 (0.39) | 24.94 (0.14) | 26.42 (0.15) |
| DISTILLM (Ko et al., 2024) | **26.21** (0.29) | 13.53 (0.13) | 16.96 (0.66) | 25.78 (0.19) | 28.51 (0.26) |
| Ours (ABKD) | 26.08 (0.36) | **13.86** (0.40) | 16.63 (0.26) | **27.25** (0.38) | **29.69** (0.21) |
| *GPT-2 XL (1.5B) → GPT-2 Large (0.8B)* | | | | | |
| SFT | 25.42 (0.32) | 12.91 (0.46) | 16.31 (0.51) | 23.76 (0.28) | 25.72 (0.07) |
| KD (Hinton, 2015) | 26.02 (0.43) | 12.34 (0.52) | 16.26 (0.44) | 25.11 (0.37) | 26.44 (0.12) |
| SeqKD (Kim & Rush, 2016) | 26.29 (0.47) | 13.53 (0.34) | 16.39 (0.36) | 25.81 (0.40) | 27.51 (0.10) |
| MiniLLM (Gu et al., 2024a) | 26.12 (0.25) | 13.79 (0.31) | **17.35** (0.51) | 26.12 (0.37) | 28.53 (0.17) |
| GKD (Agarwal et al., 2024) | 26.06 (0.34) | 13.21 (0.45) | 16.64 (0.45) | 26.13 (0.41) | 27.13 (0.21) |
| DISTILLM (Ko et al., 2024) | **26.56** (0.36) | 13.97 (0.36) | 16.61 (0.45) | 26.73 (0.36) | 29.24 (0.23) |
| Ours (ABKD) | 26.51 (0.22) | **14.38** (0.43) | 16.63 (0.42) | **28.05** (0.21) | **29.92** (0.14) |

effect and tends to be more aggressive in achieving better matching by penalizing errors on hard classes (Fig. 1b).

In this sense, by tuning $\alpha$ and $\beta$, we can flexibly balance the influence of the two effects and avoid extreme cases (Fig. 1g). For a finer-grained theoretical analysis and hyper-parameter tuning guidelines, please see App. D, Thm. D.1.

**Comparing with the Weighted Sum.** As discussed earlier in Sec. 3.3, WSD often focuses excessively on the extreme values of $p/q$, leading to unstable optimization. Fortunately, one can show that the $\alpha$-$\beta$-divergence can finely adjust the focus on different likelihood ratios $p/q$, thus enjoying a more stable gradient. For further analysis, see App. E.

**Comparing with the $\alpha$-divergence.** One might also recall the $\alpha$-divergence to achieve the trade-off, which is defined as $\mathbb{D}_\alpha(p\|q) \triangleq \frac{1}{\alpha(\alpha-1)} \left[ \sum_k p(k)^\alpha q(k)^{1-\alpha} - 1 \right]$. It includes $\mathbb{D}_{\mathrm{KL}}(p\|q_\theta)$ as $\alpha \to 1$, and $\mathbb{D}_{\mathrm{KL}}(q_\theta\|p)$ as $\alpha \to 0$. Note that when $\beta = 1-\alpha$, it becomes a special case of our framework. According to Prop. 4.2, to decrease $\alpha$, one has to increase $\beta$ to ensure that they add up to 1. Such unnecessary restriction hinders its ability to achieve better performance, as shown

in Fig. 1(a) and (f). For further analysis, please see App. F.

## 5. Experiments

In the following, we investigate to what extent our theoretical results translate into practice on natural language and vision tasks. ***Due to space limitations, please see App. I for more details on datasets, competitors, and implementation.***

### 5.1. Natural Language Processing Tasks

**Datasets.** We evaluate our methods on five task-agnostic instruction-following benchmarks. Evaluation metric is based on ROUGE-L (Lin, 2004). Details about the datasets and evaluation metric can be found in App. I.1.1.

**Competitors.** We consider the following state-of-the-art (SOTA) baselines: 1) supervised fine-tuning (SFT) with only student model on fixed datasets; 2) KD with FKLD on fixed datasets; 3) SeqKD with SFT to teacher-generated output; 4) MiniLLM with RKLD using a policy gradient approach on student-generated outputs (SGOs); 5) GKD with JSD on

*Table 3.* Evaluation of the effect of different loss functions. **WSD**: weighted sum of FKLD and RKLD. **HD**: Hellinger distance. **SED**: Squared euclidean distance.

| Loss Function | Dolly Eval | Self-Instruct | Vicuna Eval | Super-Natural | Unnatural |
|---|---|---|---|---|---|
| FKLD | 23.80 (0.37) | 10.01 (0.75) | 15.25 (0.65) | 17.69 (0.26) | 18.99 (0.05) |
| RKLD | 24.77 (0.37) | 12.02 (0.48) | 15.06 (0.28) | 23.27 (0.29) | 26.01 (0.11) |
| WSD | 23.33 (0.52) | 10.52 (0.47) | 14.83 (0.61) | 19.67 (0.13) | 21.21 (0.21) |
| HD | 25.15 (0.36) | 12.39 (0.77) | 15.43 (0.20) | 24.14 (0.23) | 26.83 (0.15) |
| SED | 21.04 (0.51) | 10.00 (0.56) | 13.73 (0.17) | 19.34 (0.19) | 22.62 (0.19) |
| $\alpha$-divergence | 25.15 (0.41) | 12.92 (0.22) | 15.60 (0.27) | 24.83 (0.21) | 27.81 (0.10) |
| $\beta$-divergence | 24.12 (0.38) | 11.18 (0.27) | 14.95 (0.33) | 20.98 (0.23) | 23.15 (0.14) |
| $\alpha$-$\beta$-divergence | **25.65** (0.24) | **13.47** (0.42) | **16.06** (0.25) | **26.47** (0.31) | **29.32** (0.08) |

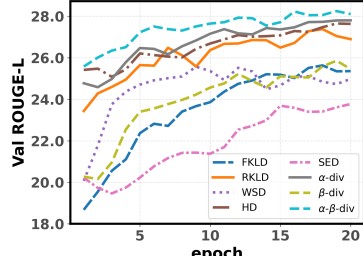

*Figure 2.* Performance across different loss functions on the validation set.

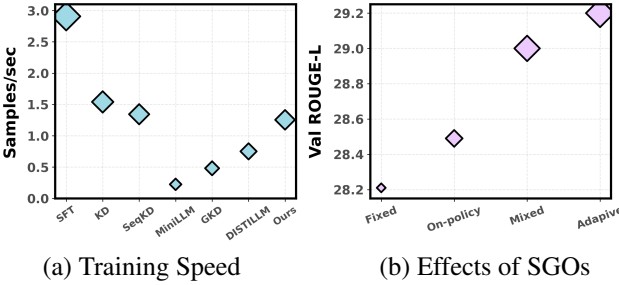

(a) Training Speed      (b) Effects of SGOs

*Figure 3.* Comparison of training speeds and the effects of using SGOs. Please see Sec. I.1.2 for details of different SGOs strategies.

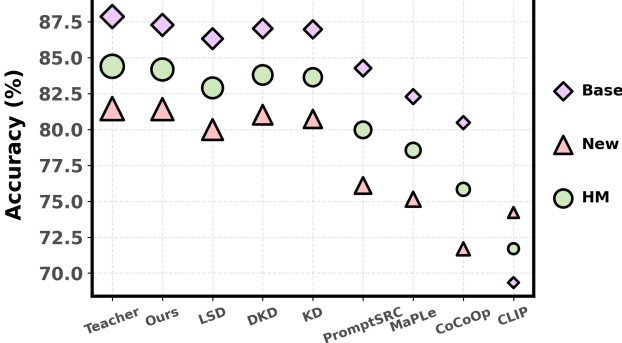

*Figure 4.* Comparison with SOTA methods on base-to-new setting. **HM** denotes the harmonic mean of base and new accuracy. Results are averaged across 11 datasets, with per-dataset details in Tab. 16. Results of baseline CLIP are evaluated on the pre-trained model. **Teacher**: ViT-L/14 CLIP; **Student**: ViT-B/16 CLIP.

a mixture of SGOs and fixed datasets; 6) DISTILLM with S(R)KL on a mixture of SGOs and fixed datasets. Please refer to App. I.1.2 for details about competitors and SGOs.

**Results.** From Tab. 2, we have the following observations: 1) Distillation methods often outperform SFT, showcasing their potential. However, they can sometimes yield worse results (*e.g.*, KD on Unnatural when distilling *GPT-2 XL* into *GPT-2*), highlighting the importance of selecting a proper distillation objective. 2) By simply modifying the distillation objective, our framework outperforms vanilla KD and SFT across various datasets when distilling *GPT-2 XL* (1.5B) to smaller-scale families of *GPT-2* (0.1B~0.8B); 3) Prior arts (Ko et al., 2024; Agarwal et al., 2024) show that training with SGOs can lead to significant improvements. However, even when compared to SGOs-based methods (*e.g.*, GKD, DISTILLM) under this inherently unfair setting, our approach consistently achieves superior or comparable results, especially on Super-Natural and Unnatural datasets.

**Efficiency Comparison**. Fig. 3(a) shows that our framework matches the training speed of vanilla KD, as it only modifies the distillation objective without introducing additional cost. This addresses concerns regarding the scalability of our method. In contrast, other distillation methods **require 1.6 to 7 times longer training time** due to the continuous need to sample student's outputs during training.

**Effects of SGOs.** We examine the robustness of our framework by evaluating its performance with various SGOs ap-

proaches. As shown in Fig. 3(b), our framework consistently delivers high performance across different settings, highlighting its adaptability and effectiveness. Further experimental results and analyses are provided in App. J.1.1.

**Effects of Loss Functions.** Tab.3 compares the performance between various loss functions. The results show that $\alpha$-$\beta$-divergence consistently outperforms the others, while using only $\alpha$- or $\beta$-divergence degrades performance due to limited expressivity. In particular, $\alpha$-$\beta$-divergence achieves improvements of 0.81 to 3.31 over FKLD and RKLD across five datasets, whereas WSD, which combines weighted FKLD and RKLD, fails to deliver comparable results. Furthermore, Fig.2 demonstrates the superior performance of $\alpha$-$\beta$-divergence during the entire training phase.

In summary, these empirical results align with the theoretical insights in Sec. 3 and show that even modest adjustments to the loss function can yield significant improvements.

### 5.2. Vision Tasks

**Datasets.** We conduct experiments on 12 popular image recognition datasets. Dataset details are referred to App. I.2.1. The evaluation metric used is accuracy. Apart

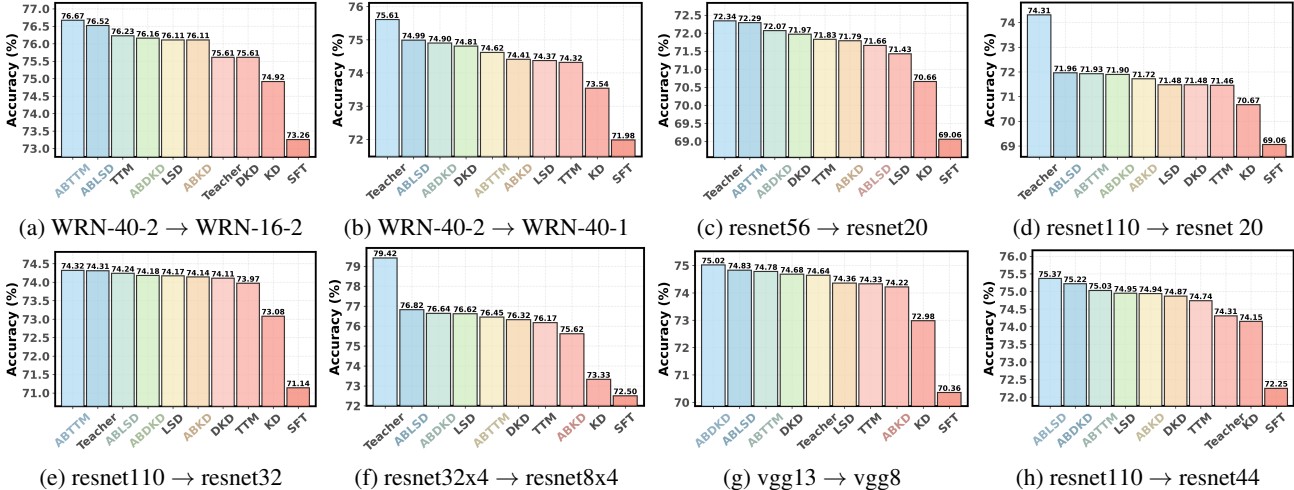

*Figure 5.* Accuracy on CIFAR-100 for student models trained with different distillation methods. ABDKD, ABTTM, ABLSD, and ABKD are our implementations by rectifying the backbone's loss function. For details on backbones, please refer to Sec. I.2.2.

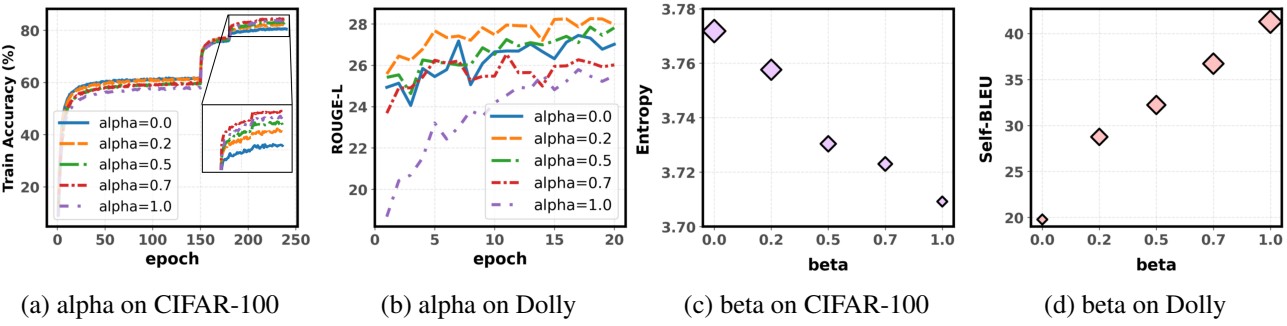

*Figure 6.* Sensitivity analysis of hyperparameters $\alpha$ and $\beta$. **(a)-(b)** For low-dimensional output distribution in CIFAR-100, a smaller $\alpha$ leads to excessive penalization for error with limited gains. However, for higher-dimensional distribution in Dolly (*e.g.*, 50,527 for GPT-2), a well-tuned smaller $\alpha$ is critical. **(c)-(d)** A larger $\beta$ sharpens output distributions by emphasizing classes with high student confidence.

from the standard training-evaluation paradigm, we further consider a novel base-to-new setting (Zhou et al., 2022; Hua et al., 2025) to more thoroughly analyze the student model's generalization across classes. In this setup, training is performed on base classes, and accuracy is evaluated on both base and new classes. Please see App. I.2.3 for more details.

**Competitors.** We consider the following SOTA distillation methods: 1) KD, 2) DKD, 3) LSD, and 4) TTM. For the base-to-new setting, we also compare with SOTA SFT methods: 5) CoCoOp, 6) MaPLe, and 7) PromptSRC. Please refer to App. I.2.2 for method details.

**Results.** Fig. 4 and Fig. 5 show results from 9 teacher-student architectures on 12 datasets. Based on these, we conclude: 1) Without modifying the distillation objective, methods that more effectively utilize teacher distribution knowledge (*e.g.*, DKD, TTM, and LSD) can outperform vanilla KD; 2) However, their scores fall short in some cases, such as LSD in base-to-new setting; 3) Orthogonal to them,

our framework selects more suitable distillation objectives for specific teacher-student pairs, showing competitive or superior results, particularly in base-to-new setting.

**Apply to Other Distillation Techniques**. Fig. 5 also shows that our framework can act as a simple plug-and-play tool to rectify the loss functions used by existing methods, yielding further improvements (*e.g.*, ABDKD vs DKD).

### 5.3. Sensitivity Analysis

We next analyze the effects of hardness-concentration and confidence-concentration, which helps validate the theoretical insights shown in Prop. 4.2 and Thm. D.1.

**Effect of $\alpha$ on hardness-concentration.** Figs. 6(a) and (b) show performance during training for different $\alpha$. In CIFAR-100, with its relatively low-dimensional output distribution, a smaller $\alpha$ (stronger hardness-concentration) aggressively penalizes errors but offers limited gains. However, in Dolly,

with a higher-dimensional output (*e.g.*, GPT-2's vocabulary size of 50,257), a well-tuned smaller $\alpha$ is crucial to avoid local optima, especially in early training stages.

**Effect of $\beta$ on confidence-concentration.** Figs. 6(c) and (d) show how $\beta$ affects Shannon entropy of the output distribution and Self-BLEU score (Zhu et al., 2018) of output sequences (100 indicates deterministic outputs and 0 denotes maximum diversity). The smaller $\beta$ (weaker confidence-concentration) places more emphasis on classes with low student confidence, encouraging the student to focus more on learning the soft label information from the teacher distribution. This leads to a smoother output distribution (higher entropy) and more diverse generated sequences (lower Self-BLEU). Thus, selecting an appropriate $\beta$ ensures a balance between focusing on the target class and learning more soft label information.

## 6. Conclusion

In this paper, we argue that the key to KD lies in trading off two mode-concentration effects: hardness-concentration and confidence-concentration. The widely used FKLD and RKLD fail to achieve this balance, instead representing two extreme cases that lead to improper probability allocation. To address this issue, we introduce ABKD, a generic distillation framework based on $\alpha$-$\beta$-divergence. ABKD generalizes FKLD and RKLD to a broader family of divergences, offering greater flexibility. Our theoretical results show that ABKD can flexibly interpolate between the above two extremes, enabling an effective trade-off. Extensive experiments further demonstrate its effectiveness.

## Acknowledgements

This work was supported in part by the Fundamental Research Funds for the Central Universities, in part by the National Key R&D Program of China under Grant 2018AAA0102000, 2024QY210004 and 2022YFC3302300, in part by National Natural Science Foundation of China: 62236008, 62441232, U21B2038, U23B2051, 62122075, 62206264 and 92370102, in part by Youth Innovation Promotion Association CAS, in part by the Strategic Priority Research Program of the Chinese Academy of Sciences, Grant No.XDB0680201, in part by the China National Postdoctoral Program for Innovative Talents under Grant BX20240384. The authors thank anonymous reviewers for their insightful comments and suggestions.

## Impact Statement

The work presented in this paper aims to advance the field of knowledge distillation (KD), a promising direction to enable knowledge transfer between different models. This potential has been demonstrated in some recent frontier works, such as DeepSeek-R1 (Guo et al., 2025) and Qwen-3 (Yang et al., 2025). Although these methods primarily adopt SeqKD-based distillation techniques (Kim & Rush, 2016) and therefore do not involve distribution matching through KL divergence, the results in Tab. 2 demonstrate that logit-based methods possess even greater potential. We believe that the applications of KD techniques will continue to expand.

In this work, we provide a theoretical analysis of the fundamentally different behaviors—mode-covering and mode-seeking—exhibited by forward and reverse KL divergences in KD, grounded in two novel mode-concentration effects. These new theoretical insights further shed light on the development of more principled and effective distillation objectives. Interestingly, our theoretical framework also aligns with findings from related studies, such as the phenomenon of *likelihood displacement* (Razin et al., 2024; Ren & Sutherland, 2024) observed in Direct Preference Optimization (DPO). This phenomenon stems from the equivalence of DPO *w.r.t.* the reverse optimization of KL divergence (Rafailov et al., 2023; Tajwar et al., 2024).

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

# Appendix

## Table of Contents

## A. Prior Arts

**Knowledge distillation** (Hinton, 2015) is a promising technology for transferring knowledge between different models. The typical setup assumes the presence of a larger teacher model with more parameters and a smaller student model with fewer parameters. To achieve this knowledge transfer, a common approach is to let the student distribution $q_\theta$ mimic the teacher distribution $p$ by minimizing a distributional measure $\mathbb{D}(p\|q_\theta)$. This approach is referred to as logit-based distillation (the focus of this work). Another promising approach is to leverage the rich information in the intermediate layers of the model, such as the attention matrix (Zagoruyko & Komodakis, 2016; Sun et al., 2020; Jiao et al., 2019; Wang et al., 2020b;a), the embedding features and their relationships (Romero et al., 2014; Liang et al., 2023b; Sun et al., 2019; Liang et al., 2023a; Lv et al., 2024; Tian et al., 2019), *etc*. These methods are known as feature-based distillation. Due to the success of KD, it often outperforms supervised fine-tuning and has led to improved performance in various downstream tasks, including image classification (Kim et al., 2024; Yang et al., 2021; 2023a; 2024), instruction generation (Gu et al., 2024b; Zhou et al., 2025b;a), neural architecture search (Wang et al., 2021), and object detection (Li et al., 2024; Lv et al., 2024).

**Logit-based methods** aim to minimize the distance between student and teacher distributions, which have achieved profound success in the past few years. To do this, one can choose different distillation objectives, such as Maximum Mean Discrepancy (Huang & Wang, 2017), Total Variation Distance (Wen et al., 2023), Wasserstein Distance (Lv et al., 2024), or Pearson correlation coefficient (Huang et al., 2022). Most prior methods (Hinton, 2015) use primarily *forward* Kullback-Leibler divergence (FKLD) to let the student distribution to mimic the teacher distribution. On this basis, a variety of methods have been proposed to help the student learn better from the teacher distribution, such as using asymmetric temperature scaling (Li et al., 2022), decomposing the teacher distribution into separate learning of target class and non-target class knowledge (Zhao et al., 2022), removing temperature scaling on the student side (Zheng & Yang, 2024), normalizing logits (Sun et al., 2024), reusing the teacher's classifier (Chen et al., 2022), utilizing inter-class relationships (Lv et al., 2024; Jin et al., 2023), and so on. Despite achieving profound success, recent research points out that due to its asymmetry, FKLD tends to cover the entire support of the teacher's distribution, leading to an overly smoothed student distribution. To address this issue, many works resort to using *reverse* Kullback-Leibler divergence (RKLD) (Lee et al., 2023; Gu et al., 2024a; Kim et al., 2024; Gu et al., 2024b), which forces the student distribution to focus on a few modes in the teacher's distribution. At the same time, some works explore more general distribution measures (Wen et al., 2023; Agarwal et al., 2024; Wang et al., 2021) and composite metrics (Wu et al., 2024; Amara et al., 2022; Binici et al., 2022). Recently, some studies (Ko et al., 2024; Wu et al., 2024; Wen et al., 2023) have found that the superiority of FKLD and RKLD depends on the task and dataset. However, systematic studies providing theoretical insights into the suboptimal performance of FKLD and RKLD are either scarce or predominantly qualitative. This limits further exploration in this field.

**Contributions**: In this paper, we propose a generic distillation framework based on $\alpha$-$\beta$-divergence. Unlike previous generic methods, our approach is 1) built on balancing hardness-concentration and confidence-concentration. Based on analysis in the unified space of our framework, we 2) theoretically explain why FKLD and RKLD lead to suboptimal performance, which further complements previous empirical observations. Fortunately, our framework 3) allows for flexible interpolation between them, ensuring better performance. Furthermore, we 4) confirm the effectiveness of the newly introduced distillation objective through extensive empirical experiments on language and vision datasets.

## B. Continuous Extension of $\alpha$-$\beta$-Divergence

The $\alpha$-$\beta$-divergence can be extended through continuous extension (by applying L'Hopital's Rule) to cover all values of $\alpha, \beta \in \mathbb{R}$. Its more explicit form is defined as follows.

$$
\mathbb{D}_{\text{AB}}^{(\alpha,\beta)} = \begin{cases}
\sum_k -\frac{1}{\alpha\beta}\left[p(k)^\alpha q(k)^\beta - \frac{\alpha}{\alpha+\beta}p(k)^{\alpha+\beta} - \frac{\beta}{\alpha+\beta}q(k)^{\alpha+\beta}\right], & \text{for} \quad \alpha, \beta, \alpha+\beta \neq 0, \\
\sum_k \frac{1}{\alpha^2}\left[p(k)^\alpha\left(\ln p(k)^\alpha - \ln q(k)^\alpha\right) - p(k)^\alpha + q(k)^\alpha\right], & \text{for} \quad \alpha \neq 0, \beta = 0, \\
\sum_k \frac{1}{\alpha^2}\left[\ln q(k)^\alpha - \ln p(k)^\alpha + \left(\frac{q(k)^\alpha}{p(k)^\alpha}\right)^{-1} - 1\right], & \text{for} \quad \alpha = -\beta \neq 0, \\
\sum_k \frac{1}{\beta^2}\left[q(k)^\beta\left(\ln q(k)^\beta - \ln p(k)^\beta\right) - q(k)^\beta + p(k)^\beta\right], & \text{for} \quad \alpha = 0, \beta \neq 0, \\
\sum_k \frac{1}{2}\left[\ln p(k) - \ln q(k)\right]^2, & \text{for} \quad \alpha, \beta = 0.
\end{cases}
\tag{8}
$$

## C. More Discussion Related to Tracking Probability Allocation with Log Mass Ratio

### C.1. The Relationship between Log Mass Ratio and Logit Gradient

For a given divergence algorithm $\ell \triangleq \mathbb{D}_{\mathcal{A}}(p \parallel q)$, consider using the gradient descent method to update the loss function $\ell$ w.r.t the logits $f_y^t$, then the distribution at the next step $p^{t+1}$ is given by:

$$
\begin{aligned}
q_{t+1}^{\mathcal{A}}(y) &= \frac{\exp(f_y^{t+1})}{\sum_k \exp(f_k^{t+1})} \\
&= \frac{\exp(f_y^t - \eta \nabla_{f_y^t} \ell)}{\sum_k \exp(f_k^t - \eta \nabla_{f_k^t} \ell)} \\
&= q_t(y) \cdot \frac{\exp(-\eta \nabla_{f_y^t} \ell)}{\sum_k q_t(k) \exp(-\eta \nabla_{f_k^t} \ell)}.
\end{aligned}
\tag{9}
$$

Observing that the denominator serves as a normalization constant, this can be rewritten as:

$$
\frac{q_{t+1}^{\mathcal{A}}(y)}{q_t(y)} \propto \exp\left(-\eta \nabla_{f_y^t} \ell\right).
\tag{10}
$$

Taking the logarithm on both sides, we get:

$$
\log \frac{q_{t+1}^{\mathcal{A}}(y)}{q_t(y)} = -\eta \cdot \nabla_{f_y^t} \ell + \mathsf{N}_t^{\mathcal{A}}(y),
\tag{11}
$$

where $\mathsf{N}_t^{\mathcal{A}}(y)$ denotes constant normalization factors independent of $y$. This indicates that the log mass ratio is proportional to $\nabla_{f_y^t} \ell$.

### C.2. The Relationship Between Overall Gradient and Logit Gradient

We first give the relationship between overall gradient and logit gradient:

$$
\nabla_{\boldsymbol{W}} \ell = \boldsymbol{J}^\top \cdot \nabla_f \ell.
\tag{12}
$$

In this case, $\boldsymbol{J}$ is the Jacobian matrix representing the gradient of logits w.r.t. model parameters, and its dimensions are $C \times M$, where $C$ is the dimensionality of logits, and $M$ is the dimensionality of the model parameters $\boldsymbol{W}$. Typically, we have $M \gg C$. For example, for an image classification task using ResNet-110 on CIFAR-100, $C = 100$ and $M = 1,110,240$. In the case of instruction generation tasks, for GPT-2 XLarge, $C = 50,257$ and $M = 1,500,000,000$. Thus, the matrix $\boldsymbol{J}$ is close to being full rank $C$.

In this case, if $\nabla_{\boldsymbol{W}} \ell \to 0$, then we must have:

$$
\boldsymbol{J}^\top \cdot \nabla_f \ell \to 0.
\tag{13}
$$

Since the Jacobian matrix $\boldsymbol{J}$ is full rank, the product can only approach zero if $\nabla_f \ell \to 0$, i.e., $\nabla_{f_y} \ell \to 0$ for all class channels $y$.

## D. $\alpha$-$\beta$-divergence: Further Analysis on Trading off Hardness-Concentration and Confidence-Concentration

Of course, there is no free lunch. A broad hyperparameter space offers more flexibility but also makes finding suitable values more difficult. While grid search could theoretically yield optimal performance, it introduces additional overhead in our framework. Fortunately, guided by the following theoretical insights, we can design a principled divergence algorithm for the target task with inductive bias more efficiently.

**Theorem D.1.** *Let $q_{t+1}^{(\alpha,\beta)}(y)$ be the distribution obtained after one gradient step, starting from $q_t$ using the $\alpha$-$\beta$-divergence. Define $\Delta_t^{(\alpha,\beta)}$ as the difference of log mass ratios across two classes $y_1$ and $y_2$, obtained from the $\alpha$-$\beta$-divergence:*

$$
\Delta_t^{(\alpha,\beta)}(y_1, y_2) \triangleq \mathsf{LogR}_t^{(\alpha,\beta)}(y_1) - \mathsf{LogR}_t^{(\alpha,\beta)}(y_2).
\tag{14}
$$

*We have the following (for appropriate positive constants $\zeta$, $\delta_1$, $\delta_2$, and any real numbers $\alpha_1$ and $\alpha_2$ in the range $[0,1]$ satisfying $\alpha_1 < \alpha_2$):*

1. *$\alpha$-$\beta$-divergence transfers probability mass from overestimated classes to underestimated classes more aggressively as $\alpha$ decreases. If $y_1$ and $y_2$ are such that $\delta_1 < q_t(y_1) = q_t(y_2) \leq p(y_1)$ (where $\delta_1 > 0$), and $p(y_1) \geq p(y_2) + \zeta$, it holds that $\Delta_t^{(\alpha_1,\beta)}(y_1, y_2) \geq \Delta_t^{(\alpha_2,\beta)}(y_1, y_2)$.*

2. *$\alpha$-$\beta$-divergence reduces the probability mass of classes with larger error $|p(y) - q_t(y)|$ more aggressively as $\alpha$ decreases. If $y_1$ and $y_2$ are such that $p(y_1) < q_t(y_1) = q_t(y_2) \leq 1 - \delta_2$ (where $\delta_2 > 0$), and $p(y_1) \geq p(y_2) + \zeta$, it holds that $\Delta_t^{(\alpha_1,\beta)}(y_1, y_2) \geq \Delta_t^{(\alpha_2,\beta)}(y_1, y_2)$.*

3. *The $\alpha$-$\beta$-divergence becomes more (less) preferential in focusing the error on classes with higher student confidence as $\beta$ increases (decreases) when reducing $\left|\mathsf{LogR}_t^{(\alpha,\beta)}(y)\right|$.*

The proof is in App. G.6. Case 1 shows that a smaller $\alpha$ (stronger hardness-concentration) leads to aggressive mass reallocation across classes when some classes are overestimated. Case 2 shows that a smaller $\alpha$ will more aggressively penalize overestimated classes with large errors (*a.k.a.*, hard classes). In this sense, a proper assignment of $\alpha$ leads to a better hardness-concentration effect. On the other hand, case 3 shows that a larger $\beta$ tends to focus more on reducing the error from classes with high student confidence and ignores the error from other classes, as shown in Fig. 6(c) and Fig. 6(d), resulting in a sharper student distribution. As such, one can select a proper $\beta$ to ensure a better confidence-concentration effect.

**Hyperparameter tuning guidelines**. In principle, one may prefer to select a larger $\beta$ than 0, which should be inversely proportional to the ratio of non-target classes in the output distribution. A proper $\beta$ allows the student to effectively learn from the teacher's soft labels while maintaining an adequate focus on the target class. On the other hand, choosing a smaller $\alpha$ than 1 leads to more aggressive probability mass reallocation across classes. Therefore, a small and proper $\alpha$ can more effectively avoid local optima. This becomes particularly important when the two distributions are far apart, as shown in Fig. 6(b).

Empirically, we find that for tasks with low-dimensional output distributions, such as image classification on CIFAR-100, selecting a large $\alpha$ and small $\beta$ is sufficient to achieve optimal performance, as shown in Tab. 7 and Tab. 4. However, for tasks with more high-dimensional output distributions, such as instruction generation on the Dolly dataset, selecting a small $\alpha$ (which leads to more aggressive reallocation of probability mass) and a large $\beta$ (to emphasize learning the soft label information) are crucial for achieving exceptional performance, as shown in Fig. 1(a) and App. I.1.3.

## E. Comparison of Gradients: FKLD, RKLD, and Our Framework

**Lemma E.1** (Cichocki et al. 2011). *Given two distributions $p$ and $q_\theta$, the gradient of the $\alpha$-$\beta$-divergence with respect to $\theta$ is calculated as:*

$$\frac{\partial \mathbb{D}_{AB}^{(\alpha,\beta)}(p \parallel q_\theta)}{\partial \theta} = -\sum_k \frac{\partial q_\theta(k)}{\partial \theta} \cdot \underbrace{q_\theta(k)^{\alpha+\beta-1}}_{weights} \underbrace{\frac{\mathbf{r}_{p,q_\theta}^\alpha - 1}{\alpha}}_{\alpha\text{-}zoom}, \tag{15}$$

*where $\mathbf{r}_{p,q_\theta}$ is the ratio between arbitrary distributions $p$ and $q_\theta$.*

*Proof.* The formula for the $\alpha$-$\beta$-divergence is defined as follows:

$$\mathbb{D}_{AB}^{(\alpha,\beta)}(p \parallel q_\theta) = -\frac{1}{\alpha\beta}\sum_k \left[ p(k)^\alpha q_\theta(k)^\beta - \frac{\alpha}{\alpha+\beta}p(k)^{\alpha+\beta} - \frac{\beta}{\alpha+\beta}q_\theta(k)^{\alpha+\beta} \right]. \tag{16}$$

Taking the derivative of each term with respect to the parameter $\theta$, we have:

$$\frac{\partial \mathbb{D}_{AB}^{(\alpha,\beta)}(p \parallel q_\theta)}{\partial \theta} = \sum_k \frac{\partial}{\partial q_\theta(k)}\mathbb{D}_{AB}^{(\alpha,\beta)}(p \parallel q_\theta) \cdot \frac{\partial q_\theta(k)}{\partial \theta}, \tag{17}$$

where

$$\frac{\partial}{\partial q_\theta(k)}\mathbb{D}_{\text{AB}}^{(\alpha,\beta)}(p \parallel q_\theta) = -\frac{1}{\alpha\beta}\left(\beta p(k)^\alpha q_\theta(k)^{\beta-1} - \beta q_\theta(k)^{\alpha+\beta-1}\right)$$

$$= -\frac{1}{\alpha}\left(p(k)^\alpha q_\theta(k)^{\beta-1} - q_\theta(k)^{\alpha+\beta-1}\right) \tag{18}$$

$$= -q_\theta(k)^{\alpha+\beta-1} \cdot \frac{\left(\frac{p(k)}{q_\theta(k)}\right)^\alpha - 1}{\alpha}.$$

Substituting Eq. 18 into Eq. 17, we obtain:

$$\frac{\partial \mathbb{D}_{\text{AB}}^{(\alpha,\beta)}(p \parallel q_\theta)}{\partial \theta} = -\sum_k \frac{\partial q_\theta(k)}{\partial \theta} \cdot q_\theta(k)^{\alpha+\beta-1}\frac{\left(\frac{p(k)}{q_\theta(k)}\right)^\alpha - 1}{\alpha}. \tag{19}$$

This concludes the proof. □

The gradient of the FKLD with respect to $\theta$ is given by

$$\frac{\partial}{\partial\theta}\mathbb{D}_{\text{KL}}(p\|q_\theta) = -\sum_k \frac{\partial q_\theta(k)}{\partial\theta} \cdot \frac{p(k)}{q_\theta(k)}. \tag{20}$$

The result is the negative gradient of the model probability, inversely weighted by its value. As Ko et al. (2024) stated, when $q_\theta(k) \approx 0$, the gradient norm increases, causing large, noisy updates that can hinder optimization. Similarly, the derivative of the reverse KL divergence with respect to $\theta$ is given by:

$$\frac{\partial}{\partial\theta}\mathbb{D}_{\text{KL}}(q_\theta\|p) = -\sum_k \frac{\partial q_\theta(k)}{\partial\theta} \cdot \left(\log\frac{q_\theta(k)}{p(k)} + 1\right). \tag{21}$$

This value becomes very large when $p(k) \approx 0$. In our framework, by Lem. E.1, the derivative of the $\alpha$-$\beta$-divergence with respect to $\theta$ is:

$$\frac{\partial \mathbb{D}_{\text{AB}}^{(\alpha,\beta)}(p \parallel q_\theta)}{\partial \theta} = -\sum_k \frac{\partial q(k)}{\partial \theta} \cdot q_\theta(k)^{\alpha+\beta-1} \cdot \frac{\left(\frac{p(k)}{q_\theta(k)}\right)^\alpha - 1}{\alpha}.$$

When $\alpha = 1$ and $\beta = 0$, FKLD becomes a special case; when $\alpha = 0$ and $\beta = 1$, RKLD becomes a special case. The parameter $\alpha$ controls the focus on large or small ratios $p/q_\theta$, while $\beta$ adjusts the weighting of these ratios through the scaling factor $q_\theta^{\alpha+\beta-1}$. When choosing $\alpha = 1$, $\frac{\partial \mathbb{D}_{\text{AB}}^{(\alpha,\beta)}(p\|q_\theta)}{\partial\theta}$ tends to excessively focus on the extreme values of $p/q$ (i.e., when $p(x) > 0$ and $q(x) \approx 0$). On the other hand, choosing a $\alpha$ close to 0 would overly focus on the extreme values of $q/p$ (i.e., when $q(x) > 0$ and $p(x) \approx 0$). Additionally, choosing a larger value of $\alpha + \beta > 1$ would place more focus on $p/q$ of classes with high student confidence, while conversely treating all classes more equally. This means they provide a **fine-grained** way to tune the model to emphasize specific likelihood ratio ranges, thereby ensuring stable gradient optimization.

## F. Pursuing a Proper Mass allocation via $\alpha$-Divergence

To achieve an effective trade-off between hardness-concentration and confidence-concentration, one can, for example, extend FKLD and RKLD to a family of generalized divergences by introducing an additional dimension $\alpha$, which is known as the $\alpha$-divergence (Chernoff, 1952).

**Definition F.1 ($\alpha$-divergence).** Consider $\alpha \in \mathbb{R} \setminus \{0, 1\}$, the $\alpha$-divergence of two distributions is given by:

$$\mathbb{D}_\alpha(p \parallel q) \triangleq \frac{1}{\alpha(\alpha-1)}\left[\sum_k p(k)^\alpha q(k)^{1-\alpha} - 1\right],$$

where $p = [p(k)]_{k=1}^C$ and $q = [q(k)]_{k=1}^C$ are two discrete distributions over $C$ classes.

**Remark.** The $\alpha$-divergence includes $\mathbb{D}_{\text{KL}}(p \parallel q_\theta)$ as $\alpha \to 1$, and $\mathbb{D}_{\text{KL}}(q_\theta \parallel p)$ as $\alpha \to 0$.

The following proposition characterizes the effect of hyperparameter $\alpha$ on reducing $|\text{LogR}_t^\alpha(y)|$. The proof is in App. G.3.

**Proposition F.2.** *The updates induced by $\alpha$-divergence for $q_t$ within one gradient descent step are given by:*

$$\left| \text{LogR}_t^\alpha(y) \right| \leq \eta \underbrace{q_t(y)^{1-\alpha}}_{(a)} \underbrace{\left| \frac{p(y)^\alpha - q_t(y)^\alpha}{\alpha} \right|}_{(b)} + q_t(y) \sum_k \underbrace{q_t(k)^{1-\alpha}}_{(a)} \underbrace{\left| \frac{p(k)^\alpha - q_t(k)^\alpha}{\alpha} \right|}_{(b)} + \left| \text{N}_t^\alpha(y) \right|,$$

*where $\text{N}_t^\alpha(y)$ denotes constant normalization factors.*

This proposition indicates that term (a) scales the relative importance of large versus small $q_t(y)$, controlling the confidence-concentration effect. Term (b) adjusts the relative emphasis of the error between $p(y)$ and $q_t(y)$, controlling the hardness-concentration effect. Together, these terms are **coupled**, with their interaction governed by $\alpha$ and $1 - \alpha$. Unfortunately, this unnecessary constraint makes it **intractable to adjust their effects independently**.

The following theorem validates our idea and shows that the $\alpha$-divergence can only inflexibly interpolate between FKLD and RKLD (Fig. 1e) in a **linear subspace** of the planar space formed by terms (a) and (b), as shown in Fig.1(a). The proof is deferred to App. G.4.

**Theorem F.3.** *Let $q_{t+1}^\alpha(y)$ be the distribution obtained after one gradient step, starting from $q_t$ using the $\alpha$-divergence. Define $\Delta_t^\alpha$ as the difference of log mass ratios across two classes $y_1$ and $y_2$, obtained from the $\alpha$-divergence:*

$$\Delta_t^\alpha(y_1, y_2) \triangleq \text{LogR}_t^\alpha(y_1) - \text{LogR}_t^\alpha(y_2).$$

*We observe the following **linear trend** (for appropriate positive constants $\zeta$, $\delta_1$, $\delta_2$, and any real numbers $\alpha_1$ and $\alpha_2$ in the range $[0, 1]$ satisfying $\alpha_1 < \alpha_2$):*

1. *The $\alpha$-divergence transfers the probability mass of overestimated classes to underestimated ones more aggressively as $\alpha$ decreases. If $y_1$ and $y_2$ are such that $p(y_2) < \delta_1 < q_t(y_1) = q_t(y_2) \leq p(y_1)$ (where $\delta_1 > 0$), and $p(y_1) \geq p(y_2) + \zeta$, it holds that $\Delta_t^{\alpha_1}(y_1, y_2) \geq \Delta_t^{\alpha_2}(y_1, y_2)$.*

2. *$\alpha$-divergence reduces the probability mass of classes with larger error $|p(y) - q_t(y)|$ more aggressively as $\alpha$ decreases. If $y_1$ and $y_2$ are such that $p(y_1) < q_t(y_1) = q_t(y_2) \leq 1 - \delta_2$ (where $\delta_2 > 0$), and $p(y_1) \geq p(y_2) + \zeta$, it holds that $\Delta_t^{\alpha_1}(y_1, y_2) \geq \Delta_t^{\alpha_2}(y_1, y_2)$.*

3. *$\alpha$-divergence increases the probability mass more preferentially on underestimated classes with larger probabilities $q_t(y)$ as $\alpha$ decreases. If $y_1$ and $y_2$ are such that $q_t(y_2) + \zeta \leq q_t(y_1) \leq 1 - \delta_2$, and $p(y_1) = p(y_2) > c_0 \cdot q_t(y_1)$, where $c_0$ is a positive constant $> 1$, it holds that $\Delta_t^{\alpha_1}(y_1, y_2) \geq \Delta_t^{\alpha_2}(y_1, y_2)$.*

4. *$\alpha$-divergence reduces the probability mass on overestimated classes with larger probabilities $q_t(y)$ more conservatively as $\alpha$ decreases. If $y_1$ and $y_2$ are such that $q_t(y_2) + \zeta \leq q_t(y_1) \leq 1 - \delta_2$, and $c_0 \cdot q_t(y_2) < p(y_1) = p(y_2) < c_1 \cdot q_t(y_1)$, where $c_0$ and $c_1$ are constants with $c_0 > 1$ and $c_1 < 1$, it holds that $\Delta_t^{\alpha_1}(y_1, y_2) \geq \Delta_t^{\alpha_2}(y_1, y_2)$.*

These cases illustrate the differences in probability mass allocation for different $\alpha$. Specifically, from Case 1 and Case 2, it can be seen that a smaller $\alpha$ leads to a more aggressive reduction of the probability mass on overestimated classes, transferring it to underestimated classes. Case 3 and Case 4 show that a smaller $\alpha$ (or larger $1 - \alpha$) tends to concentrate the probability mass more on classes with high student confidence.

In summary, we have the following conclusion.

> The $\alpha$-divergence achieves **suboptimal** balance between hardness-concentration and confidence-concentration **inflexibly**.

## G. Proofs

### G.1. Proof of Proposition 3.1

**Lemma G.1** (Tajwar et al., 2024). *For a given distribution $q_t$, the algebraic relationships between $\mathsf{LogR}$ and the gradient of the logit $f_y$ with a given learning rate $\eta$ in FKLD and RKLD are given by:*

$$\textit{FKLD:} \quad \mathsf{LogR}_t^{\mathcal{F}}(y) = \eta \cdot \big(p(y) - q_t(y)\big) + \mathsf{N}_t^{\mathcal{F}}(y),$$

$$\textit{RKLD:} \quad \mathsf{LogR}_t^{\mathcal{R}}(y) = \eta \cdot q_t(y)\Big(\log p(y) - \log q_t(y) + \sum_k q_t(k)\big(\log q_t(k) - \log p(k)\big)\Big) + \mathsf{N}_t^{\mathcal{R}}(y),$$

*where $\mathsf{N}_t^{\mathcal{F}}(y)$ and $\mathsf{N}_t^{\mathcal{R}}(y)$ denote constant normalization factors independent of $y$ and vanish to zero when $p = q_t$.*

**Restate of Proposition 3.1.** *The updates induced by FKLD and RKLD for $q_t$ within one gradient descent step are given by:*

$$\text{FKLD:} \quad \big|\mathsf{LogR}_t^{\mathcal{F}}(y)\big| \leq \eta \cdot \underbrace{1}_{(a)} \cdot \underbrace{\big|p(y) - q_t(y)\big|}_{(b)} + \big|\mathsf{N}_t^{\mathcal{F}}(y)\big|,$$

$$\text{RKLD:} \quad \big|\mathsf{LogR}_t^{\mathcal{R}}(y)\big| \leq \eta \cdot \underbrace{q_t(y)}_{(a_1)} \Big( \underbrace{\big|\log p(y) - \log q_t(y)\big|}_{(b_1)} + \sum_k \underbrace{q_t(k)}_{(a_2)} \underbrace{\big|\log p(k) - \log q_t(k)\big|}_{(b_2)} \Big) + \big|\mathsf{N}_t^{\mathcal{R}}(y)\big|,$$

*where $\mathsf{N}_t^{\mathcal{F}}(y)$ and $\mathsf{N}_t^{\mathcal{R}}(y)$ denote constant normalization factors independent of $y$ and vanish to zero when $p = q_t$.*

*Proof.* By Lemma G.1, and applying the triangle inequality, we can directly obtain the following bounds:

$$\big|\mathsf{LogR}_t^{\mathcal{F}}(y)\big| \leq \eta \cdot 1 \cdot \big|p(y) - q_t(y)\big| + \big|\mathsf{N}_t^{\mathcal{F}}(y)\big|, \tag{22}$$

$$\big|\mathsf{LogR}_t^{\mathcal{R}}(y)\big| \leq \eta \cdot q_t(y)\Big(\big|\log p(y) - \log q_t(y)\big| + \sum_k q_t(k)\big|\log p(k) - \log q_t(k)\big|\Big) + \big|\mathsf{N}_t^{\mathcal{R}}(y)\big|, \tag{23}$$

This completes the proof. $\qquad \square$

### G.2. Proof of Theorem 3.2

**Restate of Theorem 3.2.** *Let $q_{t+1}^f(y)$ be the distribution obtained after one gradient step, starting from $q_t$ using the FKLD. Likewise, let $q_{t+1}^r(y)$ be the distribution obtained using the RKLD, from $q_t$. Define $\Delta_t^f$ and $\Delta_t^r$ as the difference of log mass ratios across two classes $y_1$ and $y_2$, obtained from the forward and reverse divergences respectively:*

$$\Delta_t^f(y_1, y_2) \triangleq \mathsf{LogR}_t^{\mathcal{F}}(y_1) - \mathsf{LogR}_t^{\mathcal{F}}(y_2),$$

*and $\Delta_t^r$ is similarly defined. Then we have the following (for appropriate positive constants $\zeta$, $\delta_1$, $\delta_2$):*

1. *RKLD transfers probability mass from overestimated classes to underestimated classes more aggressively than FKLD. If $y_1$ and $y_2$ are such that $p(y_2) < \delta_1 < q_t(y_1) = q_t(y_2) < p(y_1)$ (where $\delta_1 > 0, \delta_2 > 0$), but $p(y_1) \geq p(y_2) + \zeta$, then, $\Delta_t^r(y_1, y_2) > \Delta_t^f(y_1, y_2)$.*

2. *RKLD reduces the probability mass of overestimated classes with higher error $|p(y) - q_t(y)|$ more aggressively than FKLD. If $y_1$ and $y_2$ are such that $p(y_1) < q_t(y_1) = q_t(y_2) \leq 1 - \delta_2$, but $p(y_1) \geq p(y_2) + \zeta$, then, $\Delta_t^r(y_1, y_2) > \Delta_t^f(y_1, y_2)$.*

3. *RKLD more preferentially increases probability mass on underestimated classes with larger probability $q_t(y)$ than FKLD. If $y_1$ and $y_2$ are such that $q_t(y_2) + \zeta \leq q_t(y_1) \leq 1 - \delta_2$, and $p(y_1) = p(y_2) > c_0 \cdot q_t(y_1)$, where $c_0$ is a positive constant $> 1$, then, $\Delta_t^r(y_1, y_2) > \Delta_t^f(y_1, y_2)$.*

4. *RKLD reduces probability mass on overestimated classes with larger probability $q_t(y)$ more conservatively than FKLD. If $y_1$ and $y_2$ are such that $q_t(y_2) + \zeta \leq q_t(y_1) \leq 1 - \delta_2$, and $c_0 \cdot q_t(y_2) < p(y_1) = p(y_2) < c_1 \cdot q_t(y_1)$, where $c_0$ and $c_1$ are constants with $c_0 > 1$ and $c_1 < 1$, then, $\Delta_t^r(y_1, y_2) > \Delta_t^f(y_1, y_2)$.*

*Proof.* **For Case 1**, note that $q_t(y_1) = q_t(y_2) < p(y_1)$, based on Lem. G.1, we have:

$$\Delta^f(y_1, y_2) = \eta\left(p(y_1) - p(y_2)\right), \tag{24}$$

$$\Delta^r(y_1, y_2) = \eta q(y_1)\left[\log p(y_1) - \log p(y_2)\right]. \tag{25}$$

The discrepancy between $\Delta^f$ and $\Delta^r$ is now given by:

$$\Delta^r(y_1, y_2) - \Delta^f(y_1, y_2) = \eta \cdot q(y_1)\left[\log p(y_1) - \log p(y_2) - \frac{p(y_1) - p(y_2)}{q(y_1)}\right]. \tag{26}$$

Notice that by the lagrange's mean value theorem, there exists a $c_0 \in (p(y_2), p(y_1))$ such that:

$$\log p(y_1) - \log p(y_2) = \left.\frac{d\log p}{dp}\right|_{p=c_0} \cdot \left(p(y_1) - p(y_2)\right). \tag{27}$$

Since $\left.\frac{d\log p}{dp}\right|_{p=c_0} = \frac{1}{c_0}$, we have that:

$$\Delta^r(y_1, y_2) - \Delta^f(y_1, y_2) = \eta \cdot \left(p(y_1) - p(y_2)\right) \cdot \left[\frac{q(y_1)}{c_0} - 1\right]. \tag{28}$$

This quantity is positive if $q(y_1) > c_0 = \delta_1$.

**Then, for Case 2**, similarly, since $p(y_1) < q_t(y_1) = q_t(y_2)$, we have:

$$\Delta^r(y_1, y_2) - \Delta^f(y_1, y_2) = \eta \cdot \left(p(y_1) - p(y_2)\right) \cdot \left[\frac{q(y_1)}{c_0} - 1\right]. \tag{29}$$

where $c_0 \in (p(y_2), p(y_1))$ is obtained by applying the lagrange's mean value theorem to the difference $\log p(y_1) - \log p(y_2)$. Notice that $q(y_1) > p(y_1)$, so this term is always positive.

**The proof for Case 3** is consistent with Tajwar et al. (2024), and is omitted here.

**Finally, we prove Case 4.** Noting that $q(y_2) < p(y_1) = p(y_2) < q(y_1)$, we have

$$\Delta^f(y_1, y_2) = \eta(q(y_2) - q(y_1)). \tag{30}$$

Additionally, for $\Delta^r(y_1, y_2)$, we have

$$\Delta^r(y_1, y_2) = \underbrace{\eta[q(y_1) - q(y_2)]\log p(y_1)}_{(a)} - \underbrace{\eta[q(y_1)\log(q(y_1) - q(y_2)\log(q(y_2)]}_{(b)} + \underbrace{\eta(q(y_1) - q(y_2))\mathbb{D}_{\text{KL}}(p\|q)}_{\geq 0}. \tag{31}$$

For item (b), by the lagrange's mean value theorem, there exists a point $c_0 \in (q(y_2), q(y_1))$ such that

$$\eta[q(y_1)\log(q(y_1) - q(y_2)\log(q(y_2)] = \left.\frac{dq\log q}{dq}\right|_{q=c_0}(q(y_1) - q(y_2))$$
$$= (1 + \log c_0)(q(y_1) - q(y_2)). \tag{32}$$

Substituting into Eq. 31, we obtain

$$\Delta^r(y_1, y_2) = \eta[q(y_1) - q(y_2)][\log p(y_1) - \log c_0 - 1 + \mathbb{D}_{\text{KL}}(p\|q)]. \tag{33}$$

We need to prove that

$$\eta[q(y_1) - q(y_2)][\log p(y_1) - \log c_0 - 1 + \mathbb{D}_{\text{KL}}(p\|q)] > -\eta[q(y_1) - q(y_2)]. \tag{34}$$

It suffices to choose a sufficiently large $p(y_1) \in (q(y_2), q(y_1))$ to ensure that

$$\log p(y_1) - \log c_0 + \mathbb{D}_{\text{KL}}(p\|q) > 0. \tag{35}$$

This condition can be satisfied.

$$\square$$

**Remark.** These cases highlight the contrasting behaviors of FKLD and RKLD in certain scenarios. Specifically, Case 1 shows that RKLD is more aggressive in transferring probability mass from overestimated classes (*i.e.*, $p(y) < q(y)$) to underestimated ones (*i.e.*, $p(y) > q(y)$). Case 2 indicates that when two classes have the same predicted values $q(y)$ greater than the target values $p(y)$, RKLD more aggressively penalizes hard classes with larger overestimations (*i.e.*, larger error $|p(y) - q(y)|$). Case 3 indicates that when the predicted values $q(y)$ for two classes $y_1$ and $y_2$ are both below the target value $p(y)$, RKLD preferentially increases the probability mass of the class with larger student confidence $q(y)$, even though both predictions are equal at the target value $p(y)$. Case 4 suggests that when one class $y_1$ is overestimated and another class $y_2$ is underestimated, RKLD reduces the overestimation of high-probability classes more conservatively.

In summary, combining Case 1 and Case 2, we conclude that RKLD reallocates the probability mass across classes more aggressively by penalizing errors in hard classes compared to FKLD, aiming to achieve better matching. This can help avoid local optima more quickly in some cases, ensuring superior performance and faster convergence throughout the training process, as shown in Fig. 2. From Case 3 and Case 4, it can be inferred that RKLD tends to concentrate the probability mass on a few classes with high prediction confidence. In contrast, FKLD reallocates the probability mass evenly across all classes since the weights for different classes are identical. This can lead to a sharper distribution when using RKLD, as empirically validated in Fig. 6(c) and (d) ($\beta = 0$ for FKLD and $\beta = 1$ for RKLD).

### G.3. Proof of Proposition F.2

**Lemma G.2.** *The gradient of the softmax function* $q(i) = \frac{e^{f_i}}{\sum_k e^{f_k}}$ *with respect to* $f_j$ *is given by:*

$$\frac{\partial q(i)}{\partial f_j} = \begin{cases} q(i)(1 - q(i)) & \text{if } i = j, \\ -q(i)q(j) & \text{if } i \neq j. \end{cases}$$

*Proof.* To derive $\frac{\partial q(i)}{\partial f_j}$, we consider two cases: $i = j$ and $i \neq j$.

**1. Case $i = j$:** Using the quotient rule, we have:

$$\begin{aligned} \frac{\partial q(i)}{\partial f_i} &= \frac{e^{f_i} \cdot \sum_k e^{f_k} - e^{f_i} \cdot e^{f_i}}{\left(\sum_k e^{f_k}\right)^2} \\ &= \frac{e^{f_i}}{\sum_k e^{f_k}} - \frac{e^{f_i} \cdot e^{f_i}}{\left(\sum_k e^{f_k}\right)^2}. \end{aligned} \tag{36}$$

Simplifying:

$$\frac{\partial q(i)}{\partial f_i} = q(i)(1 - q(i)). \tag{37}$$

**2. Case $i \neq j$:** We have

$$\begin{aligned} \frac{\partial q(i)}{\partial f_j} &= -\frac{e^{f_i} e^{f_j}}{\left(\sum_k e^{f_k}\right)^2} \\ &= -q(i)q(j). \end{aligned} \tag{38}$$

In conclusion, the derivative is:

$$\frac{\partial q(i)}{\partial f_j} = \begin{cases} q(i)(1 - q(i)) & \text{if } i = j, \\ -q(i)q(j) & \text{if } i \neq j. \end{cases} \tag{39}$$

$\square$

**Restate of Proposition F.2.** *The updates induced by* $\alpha$-*divergence for* $q_t$ *within one gradient descent step are given by:*

$$\log \frac{q_{t+1}^{\alpha}(x)}{q_t(x)} \leq \eta \underbrace{q_t(y)^{1-\alpha}}_{(a)} \underbrace{\left| \frac{p(y)^{\alpha} - q_t(y)^{\alpha}}{\alpha} \right|}_{(b)} + q_t(y) \sum_k \underbrace{q_t(k)^{1-\alpha}}_{(a)} \underbrace{\left| \frac{p(k)^{\alpha} - q_t(k)^{\alpha}}{\alpha} \right|}_{(b)} + \left| \mathsf{N}_t^{\alpha}(y) \right|,$$

*where* $\mathsf{N}_t^{\alpha}(y)$ *denotes constant normalization factors.*

*Proof.* The formula for the $\alpha$-divergence is:

$$\mathbb{D}_\alpha(p \parallel q) \triangleq \frac{1}{\alpha(\alpha-1)} \left[ \sum_k p(k)^\alpha q(k)^{1-\alpha} - 1 \right], \tag{40}$$

Using the chain rule, we have:

$$\frac{\partial}{\partial f_y} \mathbb{D}_\alpha(p \parallel q) = \sum_k \frac{\partial \mathbb{D}_\alpha(p \parallel q)}{\partial q(k)} \frac{\partial q(k)}{\partial f_y}, \tag{41}$$

where

$$\frac{\partial \mathbb{D}_\alpha(p \parallel q)}{\partial q(y)} = -\frac{1}{\alpha} \left( \frac{p(y)}{q(y)} \right)^\alpha, \tag{42}$$

and

$$\frac{\partial q(y)}{\partial f_y} = \frac{\partial}{\partial f_y} \frac{e^{f_y}}{\sum_k e^{f_k}}. \tag{43}$$

Combining Lem. G.2, the Eq. 41 can be expressed as

$$\begin{aligned}
\frac{\partial}{\partial f_y} \mathbb{D}_\alpha(p \parallel q) &= \sum_{k \neq y} \frac{\partial \mathbb{D}_\alpha(p \parallel q)}{\partial q(k)} \frac{\partial q(k)}{\partial f(y)} + \frac{\partial \mathbb{D}_\alpha(p \parallel q)}{\partial q(y)} \frac{\partial q(y)}{\partial f_j} \\
&= \sum_{k \neq y} \left( -\frac{1}{\alpha} \left( \frac{p(k)}{q(k)} \right)^\alpha \right) \cdot (-q(k)q(y)) + \left( -\frac{1}{\alpha} \left( \frac{p(y)}{q(y)} \right)^\alpha \right) \cdot (q(y)(1-q(y))) \\
&= -\frac{1}{\alpha} \left[ q(y)^{1-\alpha} (p(y)^\alpha - q(y)^\alpha) + q(y) \left( \sum_k q(k)^{1-\alpha}(q(k)^\alpha - p(k)^\alpha) \right) \right].
\end{aligned} \tag{44}$$

Now, consider using the gradient descent method to update the loss function $\ell$ with respect to the logits $f_y^t$, then the distribution at the next step $p^{t+1}$ is given by:

$$\begin{aligned}
q_{t+1}(y) &= \frac{\exp(f_y^{t+1})}{\sum_k \exp(f_k^{t+1})} \\
&= \frac{\exp(f_y^t - \eta \nabla_{f_y^t} \ell)}{\sum_k \exp(f_k^t - \eta \nabla_{f_k^t} \ell)} \\
&= q_t(y) \cdot \frac{\exp(-\eta \nabla_{f_y^{t+1}} \ell)}{\sum_k q_t(k) \exp(-\eta \nabla_{f_k^t} \ell)}.
\end{aligned} \tag{45}$$

Now, substituting the gradient formula of the $\alpha$-divergence, the characterization of $q_{t+1}^\alpha(x_j)$ is obtained as:

$$q_{t+1}^\alpha(y) = q_t(y) \cdot \frac{\exp\left( \frac{\eta}{\alpha} \left[ q(y)^{1-\alpha} (p(y)^\alpha - q(y)^\alpha) + q(y) \left( \sum_k q(k)^{1-\alpha}(q(k)^\alpha - p(k)^\alpha) \right) \right] \right)}{\sum_i q_t(i) \exp\left( \frac{\eta}{\alpha} \left[ q(i)^{1-\alpha} (p(i)^\alpha - q(i)^\alpha) + q(i) \left( \sum_k q(k)^{1-\alpha}(q(k)^\alpha - p(k)^\alpha) \right) \right] \right)}. \tag{46}$$

Observing that the denominator serves as a normalization constant, this can be rewritten as:

$$\frac{q_{t+1}^f(y)}{q_t(y)} \propto \exp\left( \frac{\eta}{\alpha} \left[ q(y)^{1-\alpha} (p(y)^\alpha - q(y)^\alpha) + q(y) \left( \sum_k q(k)^{1-\alpha}(q(k)^\alpha - p(k)^\alpha) \right) \right] \right). \tag{47}$$

Taking the logarithm on both sides, we get:

$$\log \frac{q_{t+1}^\alpha(y)}{q_t(y)} = \eta \underbrace{\left[ q_t(y)^{1-\alpha} \left( \frac{p(y)^\alpha - q_t(y)^\alpha}{\alpha} \right) + q_t(y) \sum_k q_t(k)^{1-\alpha} \left( \frac{q_t(k)^\alpha - p(k)^\alpha}{\alpha} \right) \right]}_{-\nabla_{f_y} \ell} + \mathsf{N}_t^\alpha(y), \tag{48}$$

where $\mathsf{N}_t^\alpha(y)$ denotes constant normalization factors. We can further derive that

$$\left| \log \frac{q_{t+1}^\alpha(y)}{q_t(y)} \right| \leq \eta \underbrace{q_t(y)^{1-\alpha}}_{(a)} \underbrace{\left| \frac{p(y)^\alpha - q_t(y)^\alpha}{\alpha} \right|}_{(b)} + q_t(y) \sum_k \underbrace{q_t(k)^{1-\alpha}}_{(a)} \underbrace{\left| \frac{p(k)^\alpha - q_t(k)^\alpha}{\alpha} \right|}_{(b)} + \left| \mathsf{N}_t^\alpha(y) \right|. \tag{49}$$

This completes the proof. $\qquad\square$

### G.4. Proof of Theorem F.3

**Lemma G.3.** *Let $p$ and $q$ be normalized probability distributions over class numbers $C$ i.e., $\sum_k^C p(k) = \sum_k^C q(k) = 1$. Define the function $F(\alpha)$ for $\alpha \in [0,1]$ as:*

$$F(\alpha) \triangleq \frac{1}{\alpha}\left(1 - \sum_k p(k)^\alpha q(k)^{1-\alpha}\right).$$

*Then, $F(\alpha)$ decreases monotonically as $\alpha$ increases, and $F(\alpha) \geq 0$ on $[0,1]$.*

*Proof.* Define:

$$S(\alpha) \triangleq \sum_k p(k)^\alpha q(k)^{1-\alpha}. \tag{50}$$

Thus, the function $F(\alpha)$ can be written as:

$$F(\alpha) = \frac{1 - S(\alpha)}{\alpha}. \tag{51}$$

First, compute the first derivative of $S(\alpha)$:

$$S'(\alpha) = \sum_k p(k)^\alpha q(k)^{1-\alpha}\left(\ln p(k) - \ln q(k)\right). \tag{52}$$

and the second derivative:

$$S''(\alpha) = \sum_k p(k)^\alpha q(k)^{1-\alpha}\left(\ln p(k) - \ln q(k)\right)^2 \tag{53}$$
$$\geq 0.$$

Since $S''(\alpha) \geq 0$ for all $\alpha$, $S(\alpha)$ is a convex function of $\alpha$.

Now, compute the derivative of $F(\alpha)$:

$$\begin{aligned}
F'(\alpha) &= \frac{d}{d\alpha}\left(\frac{1 - S(\alpha)}{\alpha}\right) \\
&= \frac{-S'(\alpha) \cdot \alpha - (1 - S(\alpha))}{\alpha^2} \\
&= \frac{N(\alpha)}{\alpha^2},
\end{aligned} \tag{54}$$

where

$$N(\alpha) \triangleq -\alpha S'(\alpha) - 1 + S(\alpha). \tag{55}$$

To show $F'(\alpha) \leq 0$, it suffices to prove $N(\alpha) \leq 0$.

Since $S(\alpha)$ is convex, $S'(\alpha)$ is non-decreasing. Additionally, considering the boundary conditions:

$$N(0) = 0, \quad N(1) = -\mathbb{D}_{\text{KL}}(p\|q) \leq 0, \tag{56}$$

and since

$$\begin{aligned}
N'(\alpha) &= \frac{d}{d\alpha}\left(S(\alpha) - \alpha S'(\alpha) - 1\right) \\
&= -\alpha S''(\alpha) \\
&\leq 0,
\end{aligned} \tag{57}$$

$N(\alpha)$ is monotonically decreasing.

Therefore, for all $\alpha \in [0,1]$:

$$N(\alpha) \leq 0. \tag{58}$$

Thus,

$$F'(\alpha) = \frac{N(\alpha)}{\alpha^2} \tag{59}$$
$$\leq 0.$$

Hence, $F(\alpha)$ is monotonically decreasing with respect to $\alpha$ on the interval $[0, 1]$.

Finally, note that $F(1) = 0$, and therefore $F(\alpha) \geq 0$. This completes the proof. □

**Lemma G.4.** *Consider the function $f(\alpha) \triangleq p(x)^\alpha - q(y_i)^\alpha$, where $p(x)$ and $q(y_i)$ are constants. The derivative of the logarithm of $f(\alpha)$ with respect to $\alpha$ is:*

$$\frac{d}{d\alpha} \ln (p(x)^\alpha - q(y_i)^\alpha) = \ln p(x) + \frac{-\left(\frac{q(y_i)}{p(x)}\right)^\alpha \ln \left(\frac{q(y_i)}{p(x)}\right)}{1 - \left(\frac{q(y_i)}{p(x)}\right)^\alpha}. \tag{60}$$

*Proof.* Start with:

$$\ln (p(x)^\alpha - q(y_i)^\alpha) = \ln p(x)^\alpha + \ln \left(1 - \left(\frac{q(y_i)}{p(x)}\right)^\alpha\right). \tag{61}$$

Differentiating with respect to $\alpha$, we get:

$$\frac{d}{d\alpha} \ln (p(x)^\alpha - q(y_i)^\alpha) = \frac{d}{d\alpha} \ln p(x)^\alpha + \frac{d}{d\alpha} \ln \left(1 - \left(\frac{q(y_i)}{p(x)}\right)^\alpha\right). \tag{62}$$

The derivative of $\ln p(x)^\alpha$ is $\ln p(x)$, and the derivative of the second term is:

$$\frac{d}{d\alpha} \ln \left(1 - \left(\frac{q(y_i)}{p(x)}\right)^\alpha\right) = \frac{-\left(\frac{q(y_i)}{p(x)}\right)^\alpha \ln \left(\frac{q(y_i)}{p(x)}\right)}{1 - \left(\frac{q(y_i)}{p(x)}\right)^\alpha}. \tag{63}$$

Combining these gives the desired result:

$$\frac{d}{d\alpha} \ln (p(x)^\alpha - q(y_i)^\alpha) = \ln p(x) + \frac{-\left(\frac{q(y_i)}{p(x)}\right)^\alpha \ln \left(\frac{q(y_i)}{p(x)}\right)}{1 - \left(\frac{q(y_i)}{p(x)}\right)^\alpha}. \tag{64}$$

□

**Lemma G.5.** *Let $1 \geq \alpha > 0$ and define*
$$h(s) \triangleq 1 - s^{2\alpha} - 2\alpha s^\alpha |\ln s| \tag{65}$$
*for $0 < s < 1$. Then $h(s)$ is strictly decreasing on $(0, 1)$.*

*Proof.* Since $0 < s < 1$, we have $\ln s < 0$, thus $|\ln s| = -\ln s$. Substitute this:

$$h(s) = 1 - s^{2\alpha} + 2\alpha s^\alpha \ln s. \tag{66}$$

Differentiating term-by-term,

$$h'(s) = -2\alpha s^{2\alpha-1} + 2\alpha(\alpha s^{\alpha-1} \ln s + s^{\alpha-1}) \tag{67}$$
$$= 2\alpha s^{\alpha-1}(\alpha \ln s + 1 - s^\alpha).$$

Set

$$q(s) = \alpha \ln s + 1 - s^\alpha. \tag{68}$$

As $s \to 0^+$, $\ln s \to -\infty$ and thus $q(s) \to -\infty$. At $s = 1$, $q(1) = \alpha \cdot 0 + 1 - 1 = 0$. Moreover,

$$q'(s) = \frac{\alpha(1 - s^\alpha)}{s} \tag{69}$$
$$> 0.$$

Since $0 < s < 1$ implies $1 - s^\alpha > 0$. Hence $q(s)$ is strictly increasing on $(0,1)$, and we have $q(1) = 0$. Thus $q(s) < 0$ for all $0 < s < 1$.

Since $2\alpha s^{\alpha-1} > 0$ and $q(s) < 0$, we have $h'(s) < 0$ for all $0 < s < 1$. Therefore, $h(s)$ is strictly decreasing on $(0,1)$. □

**Lemma G.6.** *For $0 < s < 1$ and $1 \geq \alpha > 0$, define*

$$\beta(s, \alpha) \triangleq \frac{2(1 - s^\alpha)(1 + s^\alpha)|\ln s|}{(1 + s^\alpha)^2 (\ln s)^2}. \tag{70}$$

*Then $\beta(s, \alpha)$ is strictly increasing in $s$ on $(0,1)$ and increases from 0 to $\alpha$ as $s$ goes from 0 to 1.*

*Proof.* Since $0 < s < 1$, we have $\ln s < 0$ and thus $|\ln s| = -\ln s$. Substituting this into Eq. 70 and simplifying, we obtain:

$$\beta(s, \alpha) = \frac{2(1 - s^\alpha)}{(1 + s^\alpha)|\ln s|}. \tag{71}$$

To differentiate $\beta(s, \alpha)$ with respect to $s$, let:

$$f(s) \triangleq 1 - s^\alpha, \quad f'(s) = -\alpha s^{\alpha-1}, \tag{72}$$

$$g(s) \triangleq 1 + s^\alpha, \quad g'(s) = \alpha s^{\alpha-1}, \tag{73}$$

$$h(s) \triangleq |\ln s| = -\ln s, \quad h'(s) = -\frac{1}{s}. \tag{74}$$

Thus,

$$\beta(s, \alpha) = \frac{2f(s)}{g(s)h(s)}. \tag{75}$$

Applying the quotient rule:

$$\frac{d\beta}{ds} = 2\frac{f'(s)g(s)h(s) - f(s)g'(s)h(s) - f(s)g(s)h'(s)}{[g(s)h(s)]^2}. \tag{76}$$

Plugging Eq. 72, Eq. 73 and Eq. 74 into Eq. 76 and simplifying, we arrive at:

$$\frac{d\beta}{ds} = \frac{2h(s)}{(1 + s^\alpha)^2 (\ln s)^2} \quad \text{with} \quad h(s) = 1 - s^{2\alpha} - 2\alpha s^\alpha |\ln s|. \tag{77}$$

From Lem. G.5, it follows that $h(s)$ is decreasing on $(0,1)$. As $s \to 1^-$, $h(s) \to 0$. Therefore, $h(s) > 0$ on $(0,1)$.

Since $(1 + s^\alpha)^2 (\ln s)^2 > 0$, it follows from Eq. 77 that $\frac{d\beta}{ds} > 0$. Therefore, $\beta(s, \alpha)$ is strictly increasing. Furthermore, taking limits:

$$\lim_{s \to 0^+} \beta(s, \alpha) = 0, \quad \lim_{s \to 1^-} \beta(s, \alpha) = \alpha,$$

so $\beta(s, \alpha)$ increases from 0 to $\alpha$ as $s$ goes from 0 to 1. □

**Lemma G.7.** *Let $f(s, \alpha) \triangleq \frac{s^\alpha (\ln s)^2}{(1 - s^\alpha)^2}$, where $0 < s < 1$ and $\alpha > 0$. Then, $f(s, \alpha)$ is strictly increasing with respect to $s$ for all $0 < s < 1$ and $\alpha > 0$, i.e., $\frac{\partial f}{\partial s} > 0$.*

*Proof.* To compute $\frac{\partial f}{\partial s}$, we use the quotient rule. Define:

$$u \triangleq s^\alpha (\ln s)^2, \quad v \triangleq (1 - s^\alpha)^2, \quad f(s, \alpha) \triangleq \frac{u}{v}. \tag{78}$$

The derivative is given by:

$$\frac{\partial f}{\partial s} = \frac{u'v - uv'}{v^2}. \tag{79}$$

First, compute $u'$:

$$u = s^\alpha (\ln s)^2, \quad u' = \alpha s^{\alpha-1}(\ln s)^2 + 2s^{\alpha-1}\ln s = s^{\alpha-1}\left[\alpha(\ln s)^2 + 2\ln s\right]. \tag{80}$$

Next, compute $v'$:

$$v = (1 - s^\alpha)^2, \quad v' = 2(1 - s^\alpha)\cdot(-\alpha s^{\alpha-1}) = -2\alpha s^{\alpha-1}(1 - s^\alpha). \tag{81}$$

Substituting $u'$ and $v'$ into the quotient rule:

$$\frac{\partial f}{\partial s} = \frac{s^{\alpha-1}\left[\alpha(\ln s)^2 + 2\ln s\right](1 - s^\alpha)^2 + 2\alpha s^{2\alpha-1}(\ln s)^2(1 - s^\alpha)}{(1 - s^\alpha)^4}. \tag{82}$$

Simplify the numerator:

$$\text{Numerator} = s^{\alpha-1}(1 - s^\alpha)\left[\alpha(\ln s)^2 + 2\ln s\right] + 2\alpha s^{2\alpha-1}(\ln s)^2. \tag{83}$$

Factorize:

$$\text{Numerator} = s^{\alpha-1}(1 - s^\alpha)\left[\alpha(1 + s^\alpha)(\ln s)^2 + 2(1 - s^\alpha)\ln s\right]. \tag{84}$$

Thus:

$$\frac{\partial f}{\partial s} = \frac{s^{\alpha-1}\left[\alpha(1 + s^\alpha)(\ln s)^2 + 2(1 - s^\alpha)\ln s\right]}{(1 - s^\alpha)^3}. \tag{85}$$

To determine the sign of $\frac{\partial f}{\partial s}$, note:

- $s^{\alpha-1} > 0$ since $0 < s < 1$ and $\alpha > 0$,

- $(1 - s^\alpha)^3 > 0$ since $0 < s < 1$ and $\alpha > 0$,

- $\ln s < 0$ for $0 < s < 1$, hence $(\ln s)^2 > 0$.

Denote the remaining expression inside the brackets as:

$$N \triangleq \alpha(1 + s^\alpha)(\ln s)^2 + 2(1 - s^\alpha)\ln s. \tag{86}$$

Rewrite $N$ as:

$$N = |\ln s|\left[\alpha(1 + s^\alpha)|\ln s| - 2(1 - s^\alpha)\right]. \tag{87}$$

Since $|\ln s| > 0$, we analyze $\alpha(1 + s^\alpha)|\ln s| - 2(1 - s^\alpha) > 0$:

$$\alpha > \frac{2(1 - s^\alpha)}{(1 + s^\alpha)|\ln s|}. \tag{88}$$

From Lem. G.6, it follows that for all $0 < s < 1$ and $\alpha > 0$, $\alpha > \beta(s, \alpha)$ always holds, implying:

$$N > 0. \tag{89}$$

Hence:

$$\frac{\partial f}{\partial s} > 0, \tag{90}$$

which shows that $f(s, \alpha)$ is strictly increasing with respect to $s$. $\qquad\square$

**Lemma G.8.** *Considering $p(x), q(y_1), q(y_2) \in [0, 1]$ such that $p(x) \geq q(y_1) \geq q(y_2)$, and $\alpha > 0$, the following function:*

$$F(\alpha) \triangleq \frac{q(y_1)^{1-\alpha}}{q(y_2)^{1-\alpha}} \cdot \frac{p(x)^\alpha - q(y_1)^\alpha}{p(x)^\alpha - q(y_2)^\alpha}, \tag{91}$$

*$F(\alpha)$ decreases as $\alpha$ increases.*

*Proof.* Let $h(\alpha) \triangleq \ln F(\alpha)$, we get

$$h(\alpha) = (1 - \alpha) \ln \frac{q(y_1)}{q(y_2)} + \ln (p(x)^\alpha - q(y_1)^\alpha) - \ln (p(x)^\alpha - q(y_2)^\alpha) . \tag{92}$$

Differentiating $h(\alpha)$ with respect to $\alpha$, we get

$$h'(\alpha) = -\ln \frac{q(y_1)}{q(y_2)} + \frac{d \ln (p(x)^\alpha - q(y_1)^\alpha)}{d\alpha} - \frac{d \ln (p(x)^\alpha - q(y_2)^\alpha)}{d\alpha} . \tag{93}$$

Combining Lem. G.4, we obtain the following expression for the derivative of $h(\alpha)$:

$$h'(\alpha) = -\ln \frac{q(y_1)}{q(y_2)} - \frac{\left(\frac{q(y_1)}{p(x)}\right)^\alpha \ln \left(\frac{q(y_1)}{p(x)}\right)}{1 - \left(\frac{q(y_1)}{p(x)}\right)^\alpha} + \frac{\left(\frac{q(y_2)}{p(x)}\right)^\alpha \ln \left(\frac{q(y_2)}{p(x)}\right)}{1 - \left(\frac{q(y_2)}{p(x)}\right)^\alpha} . \tag{94}$$

For convenience, define the variables $s_i = \frac{q(x_i)}{p(x)}$ for $i = 1, 2$, where $1 \geq s_1 \geq s_2$. Using this substitution, we obtain:

$$h'(\alpha) = -\ln \frac{s_1}{s_2} \underbrace{- \frac{s_1^\alpha \ln s_1}{1 - s_1^\alpha}}_{(a)} + \underbrace{\frac{s_2^\alpha \ln s_2}{1 - s_2^\alpha}}_{(b)} . \tag{95}$$

To analyze the sign of $h'(\alpha)$, consider the second derivative with respect to $\alpha$:

$$h''(\alpha) = \frac{d}{d\alpha} \left( -\frac{s_1^\alpha \ln s_1}{1 - s_1^\alpha} + \frac{s_2^\alpha \ln s_2}{1 - s_2^\alpha} \right) . \tag{96}$$

Using the quotient rule, differentiate each term separately:

$$\frac{d}{d\alpha} \left( \frac{s^\alpha \ln s}{1 - s^\alpha} \right) = \frac{s^\alpha (\ln s)^2}{(1 - s^\alpha)^2} . \tag{97}$$

Therefore:

$$h''(\alpha) = -\frac{s_1^\alpha (\ln s_1)^2}{(1 - s_1^\alpha)^2} + \frac{s_2^\alpha (\ln s_2)^2}{(1 - s_2^\alpha)^2} . \tag{98}$$

From Lem. G.7, we have

$$h''(\alpha) \leq 0.$$

Thus, we have shown that $h'(\alpha)$ is monotonically decreasing. Considering the limit of $h'(\alpha)$ as $\alpha \to 0^+$, we have:

$$\lim_{\alpha \to 0^+} h'(\alpha) = 0.$$

Therefore, $h'(\alpha) < 0$ always holds, and $F(\alpha)$ decreases as $\alpha$ increases. $\qquad\square$

**Lemma G.9.** *Define*

$$f(p(y_1)) \triangleq - (\log p(y_1) - \log q(y_2))^2 q(y_2)^{1-\alpha}$$
$$+ (\log p(y_1) - \log q(y_1))^2 q(y_1)^{1-\alpha},$$

*where $0 < q(y_2) < p(y_1) < q(y_1)$. Then:*

1. *The function $f(p(y_1))$ is monotonically decreasing with respect to $p(y_1)$.*

2. *There exists a unique constant $c_0 \in (q(y_2), q(y_1))$ such that $f(c_0) = 0$. Moreover, for all $p(y_1) > c_0$, it holds that $f(p(y_1)) < 0$.*

*Proof.* The derivative of $f$ with respect to $p(y_1)$ is:

$$f'(p(y_1)) = \frac{2}{p(y_1)} \left[ (\log p(y_1) - \log q(y_1)) q(y_1)^{1-\alpha} - (\log p(y_1) - \log q(y_2)) q(y_2)^{1-\alpha} \right]. \tag{99}$$

Noting that $0 < q(y_2) < p(y_1) < q(y_1)$, therefore $f'(p(y_1)) \leq 0$. When $p(y_1) = q(y_2)$:

$$f(q_2) = (\log q_2 - \log q_1)^2 q_1^{1-\alpha} > 0. \tag{100}$$

When $p(y_1) = q(y_1)$:

$$f(q_1) = -(\log q_1 - \log q_2)^2 q_2^{1-\alpha} < 0. \tag{101}$$

According to the intermediate value theorem, since $f(p)$ is continuous on $p \in (q_2, q_1)$ and decreases from a positive value to a negative value, there exists a unique $c_0 \in (q_2, q_1)$ such that $f(c_0) = 0$. Therefore, when $p > c_0$, $f(p) < 0$. $\qquad\square$

**Restate of Theorem F.3.** *Let $q_{t+1}^{\alpha}(y)$ be the distribution obtained after one gradient step, starting from $q_t$ using the $\alpha$-divergence. Define $\Delta_t^{\alpha}$ as the difference of log mass ratios across two classes $y_1$ and $y_2$, obtained from the $\alpha$-divergence:*

$$\Delta_t^{\alpha}(y_1, y_2) \triangleq \mathsf{LogR}_t^{\alpha}(y_1) - \mathsf{LogR}_t^{\alpha}(y_2).$$

*We observe the following **linear trend** (for appropriate positive constants $\zeta$, $\delta_1$, $\delta_2$, and any real numbers $\alpha_1$ and $\alpha_2$ in the range $[0, 1]$ satisfying $\alpha_1 < \alpha_2$):*

1. *The $\alpha$-divergence transfers the probability mass of overestimated classes to underestimated ones more aggressively as $\alpha$ decreases. If $y_1$ and $y_2$ are such that $p(y_2) < \delta_1 < q_t(y_1) = q_t(y_2) \leq p(y_1)$ (where $\delta_1 > 0$), and $p(y_1) \geq p(y_2) + \zeta$, it holds that $\Delta_t^{\alpha_1}(y_1, y_2) \geq \Delta_t^{\alpha_2}(y_1, y_2)$.*

2. *$\alpha$-divergence reduces the probability mass of classes with larger error $|p(y) - q_t(y)|$ more aggressively as $\alpha$ decreases. If $y_1$ and $y_2$ are such that $p(y_1) < q_t(y_1) = q_t(y_2) \leq 1 - \delta_2$ (where $\delta_2 > 0$), and $p(y_1) \geq p(y_2) + \zeta$, it holds that $\Delta_t^{\alpha_1}(y_1, y_2) \geq \Delta_t^{\alpha_2}(y_1, y_2)$.*

3. *$\alpha$-divergence increases the probability mass more preferentially on underestimated classes with larger probabilities $q_t(y)$ as $\alpha$ decreases. If $y_1$ and $y_2$ are such that $q_t(y_2) + \zeta \leq q_t(y_1) \leq 1 - \delta_2$, and $p(y_1) = p(y_2) > c_0 \cdot q_t(y_1)$, where $c_0$ is a positive constant $> 1$, it holds that $\Delta_t^{\alpha_1}(y_1, y_2) \geq \Delta_t^{\alpha_2}(y_1, y_2)$.*

4. *$\alpha$-divergence reduces the probability mass on overestimated classes with larger probabilities $q_t(y)$ more conservatively as $\alpha$ decreases. If $y_1$ and $y_2$ are such that $q_t(y_2) + \zeta \leq q_t(y_1) \leq 1 - \delta_2$, and $c_0 \cdot q_t(y_2) < p(y_1) = p(y_2) < c_1 \cdot q_t(y_1)$, where $c_0$ and $c_1$ are constants with $c_0 > 1$ and $c_1 < 1$, it holds that $\Delta_t^{\alpha_1}(y_1, y_2) \geq \Delta_t^{\alpha_2}(y_1, y_2)$.*

*Proof.* **First, we prove Case 1**. Note that $q_t(y_1) = q_t(y_2) = q(x)$, based on Eq. 48, we have

$$\Delta_t^{\alpha} = \eta q(x)^{1-\alpha} \cdot \frac{p(y_1)^{\alpha} - p(y_2)^{\alpha}}{\alpha}. \tag{102}$$

Let $f(\alpha) = q(x)^{1-\alpha} \cdot \frac{p(y_1)^{\alpha} - p(y_2)^{\alpha}}{\alpha}$, and then it can be rewritten as

$$f(\alpha) \triangleq q(x)^{1-\alpha} \int_{p(y_2)}^{p(y_1)} y^{\alpha-1} \, dy. \tag{103}$$

Let $t = \frac{y}{q(x)}$, substituting gives

$$f(\alpha) = q(x)^{1-\alpha} \int_{\frac{p(y_2)}{q(x)}}^{\frac{p(y_1)}{q(x)}} t^{\alpha-1} \cdot q(x)^{\alpha-1} \cdot q(x) \, dt$$

$$= q(x) \int_{\frac{p(y_2)}{q(x)}}^{\frac{p(y_1)}{q(x)}} t^{\alpha-1} \, dt. \tag{104}$$

Using Leibniz's rule, we get

$$f'(\alpha) = q(x) \int_{\frac{p(y_2)}{q(x)}}^{\frac{p(y_1)}{q(x)}} t^{\alpha-1} \ln t \, dt. \tag{105}$$

Note that the sign of $f'(\alpha)$ depends only on $\int_{\frac{p(y_2)}{q(x)}}^{\frac{p(y_1)}{q(x)}} t^{\alpha-1} \ln t \, dt$, so we define

$$h \triangleq \int_{\frac{p(y_2)}{q(x)}}^{\frac{p(y_1)}{q(x)}} t^{\alpha-1} \ln t \, dt. \tag{106}$$

Clearly, the sign of $h$ depends on the relative size of $q(x)$ with respect to $p(y_1)$ and $p(y_2)$. To differentiate $h$ with respect to $q(x)$, using Leibniz's Rule, we get

$$
\begin{aligned}
h'(q(x)) &= \frac{d}{dq(x)} \int_{\frac{p(y_2)}{q(x)}}^{\frac{p(y_1)}{q(x)}} t^{\alpha-1} \ln t \, dt \\
&= \left[ t^{\alpha-1} \ln t \right]_{t=\frac{p(y_1)}{q(x)}} \cdot \left( -\frac{p(y_1)}{q(x)^2} \right) - \left[ t^{\alpha-1} \ln t \right]_{t=\frac{p(y_2)}{q(x)}} \cdot \left( -\frac{p(y_2)}{q(x)^2} \right).
\end{aligned}
\tag{107}
$$

Simplifying, we obtain:

$$
\begin{aligned}
h'(q(x)) &= \frac{p(y_2)^\alpha \ln\left(\frac{p(y_2)}{q(x)}\right) - p(y_1)^\alpha \ln\left(\frac{p(y_1)}{q(x)}\right)}{q(x)^{\alpha+1}} \\
&= \frac{p(y_2)^\alpha \ln p(y_2) - p(y_1)^\alpha \ln p(y_1) + (p(y_1)^\alpha - p(y_2)^\alpha) \ln q(x)}{q(x)^{\alpha+1}}.
\end{aligned}
\tag{108}
$$

Note that the sign of $h'(q(x))$ depends only on the sign of the numerator, and the numerator is a monotonic increasing function of $q(x)$ $((p(y_1) \geq p(y_2))$. Note that $q(x) \leq p(y_1)$, we have

$$
\begin{aligned}
h'(q(x)) &\leq h'(p(y_1)) \\
&= \frac{p(y_2)^\alpha \ln\left(\frac{p(y_2)}{p(y_1)}\right)}{p(y_1)^{\alpha+1}} \\
&\leq 0.
\end{aligned}
\tag{109}
$$

Thus, we have proven that $h$ is monotonically decreasing as $q(x)$ increases. Also, since

$$
\begin{aligned}
h(p(y_2)) &= \int_1^{\frac{p(y_1)}{p(y_2)}} t^{\alpha-1} \ln t \, dt \\
&\geq 0,
\end{aligned}
\tag{110}
$$

and

$$
\begin{aligned}
h(p(y_1)) &= \int_{\frac{p(y_2)}{p(x_i)}}^1 t^{\alpha-1} \ln t \, dt \\
&\leq 0.
\end{aligned}
\tag{111}
$$

By the intermediate value theorem, there exists $c_\alpha \in [p(y_2), p(y_1)]$ such that when $q(x) > c_\alpha$, we have

$$h(q(x)) \leq 0 \tag{112}$$

for all values. By combining Eq. 105, we can conclude that $f'(\alpha) < 0$. Furthermore, let $\delta_1 \triangleq \max(c_\alpha)$ for any $\alpha \in [0, 1]$. Then, when $p(y_1) \geq q(x) > \delta_1$, we have $f'(\alpha) \leq 0$ for $\alpha \in [0, 1]$. Thus, we have proven that for any $\alpha_1$ and $\alpha_2 \in [0, 1]$ such that $\alpha_1 < \alpha_2$, we have

$$\Delta_t^{\alpha_1}(y_1, y_2) \geq \Delta_t^{\alpha_2}(y_1, y_2).$$

**Next, we prove Case 2**. Similarly, since $q_t(y_1) = q_t(y_2) = q(x)$, we can deduce that

$$\Delta_t^\alpha = \eta q(x)^{1-\alpha} \cdot \frac{p(y_1)^\alpha - p(y_2)^\alpha}{\alpha}. \tag{113}$$

Let $f(\alpha) \triangleq q(x)^{1-\alpha} \cdot \frac{p(y_1)^\alpha - p(y_2)^\alpha}{\alpha}$., we get

$$f(\alpha) = q(x) \int_{\frac{p(y_2)}{q(x)}}^{\frac{p(y_1)}{q(x)}} t^{\alpha-1} \, dt. \tag{114}$$

Using Leibniz's rule, we get

$$f'(\alpha) = q(x) \int_{\frac{p(y_2)}{q(x)}}^{\frac{p(y_1)}{q(x)}} t^{\alpha-1} \ln t \, dt. \tag{115}$$

Note that $q(x) > p(y_1)$, so

$$f'(\alpha) \leq 0. \tag{116}$$

Thus, this proves that for any $\alpha_1$ and $\alpha_2$ such that $\alpha_1 < \alpha_2$, we have

$$\Delta_t^{\alpha_1}(y_1, y_2) \geq \Delta_t^{\alpha_2}(y_1, y_2).$$

**We now proceed to prove Case 3**. Note that $p(y_1) = p(y_2) \geq q(y_1) \geq q(y_2) + \zeta$, thus we can deduce

$$\Delta^\alpha(y_1, y_2) = \underbrace{\eta q(y_1)^{1-\alpha} \cdot \frac{p(y_1)^\alpha - q(y_1)^\alpha}{\alpha}}_{(a)} - \underbrace{\eta q(y_2)^{1-\alpha} \cdot \frac{p(y_1)^\alpha - q(y_2)^\alpha}{\alpha}}_{(b)}$$
$$+ \underbrace{\frac{\eta \left(q(y_1) - q(y_2)\right)}{\alpha} \cdot \left[\sum_k q(k)^{1-\alpha} \left(q(k)^\alpha - p(k)^\alpha\right)\right]}_{(c)}. \tag{117}$$

First, through Lem. G.3, we know that $(c) \geq 0$, and it increases as $\alpha$ decreases on $[0, 1]$. Then, we consider the term $(a) - (b)$:

$$\Delta^\alpha(y_1, y_2) \geq (a) - (b)$$
$$= \eta q(y_1)^{1-\alpha} \cdot \frac{p(y_1)^\alpha - q(y_1)^\alpha}{\alpha} - \eta q(y_2)^{1-\alpha} \cdot \frac{p(y_2)^\alpha - q(y_2)^\alpha}{\alpha}$$
$$= \eta \underbrace{q(y_2)^{1-\alpha} \cdot \frac{p(y_2)^\alpha - q(y_2)^\alpha}{\alpha}}_{(1)} \left[\underbrace{\frac{q(y_1)^{1-\alpha}}{q(y_2)^{1-\alpha}} \cdot \frac{p(y_1)^\alpha - q(y_1)^\alpha}{p(y_2)^\alpha - q(y_2)^\alpha}}_{(2)} - 1\right]. \tag{118}$$

From Lem. G.8, it can be seen that term (2) increases as $\alpha$ decreases. Term (1) can be expressed as:

$$q(y_2)^{1-\alpha} \cdot \frac{p(y_2)^\alpha - q(y_2)^\alpha}{\alpha} = q(y_2)^{1-\alpha} \cdot \int_{q(y_2)}^{p(y_2)} t^{\alpha-1} \, dt$$
$$= q(y_2) \cdot \int_1^{\frac{p(y_2)}{q(y_2)}} t^{\alpha-1} \, dt. \tag{119}$$

Noting that

$$\frac{d}{d\alpha} \int_1^{\frac{p(y_2)}{q(y_2)}} t^{\alpha-1} \, dt = \int_1^{\frac{p(y_2)}{q(y_2)}} t^{\alpha-1} \ln t \, dt$$
$$\geq 0, \tag{120}$$

it follows that term (1) decreases as $\alpha$ decreases. Therefore, as $\alpha$ decreases, if term $(2) \leq 1$, the value of $(a) - (b)$ increases as $\alpha$ decreases, although its value remains overall less than 0. If term $(2) \geq 1$, we have $(a) - (b) \geq 0$, even though $\Delta^{\alpha=1}(y_1, y_2) < 0$.

**Finally, we prove Case 4.** Since $p(y_1) = p(y_2)$ and $q(y_2) \leq q(y_1)$, we have

$$\Delta^\alpha(y_1, y_2) = \eta q(y_1)^{1-\alpha} \cdot \frac{p(y_1)^\alpha - q(y_1)^\alpha}{\alpha} - \eta q(y_2)^{1-\alpha} \cdot \frac{p(y_1)^\alpha - q(y_2)^\alpha}{\alpha}$$
$$+ \frac{\eta \left(q(y_1) - q(y_2)\right)}{\alpha} \cdot \left[\sum_k q(k)^{1-\alpha} \left(q(k)^\alpha - p(k)^\alpha\right)\right]. \tag{121}$$

Considering $f(\alpha) \triangleq \Delta^\alpha(y_1, y_2)/\eta$, taking the derivative with respect to $\alpha$ yields:

$$f'(\alpha) = \frac{1}{\alpha^2}\Bigg[\left(-1 + \alpha \log p(y_1) - \alpha \log q(y_1)\right)p(y_1)^\alpha q(y_1)^{1-\alpha}$$
$$+ \left(1 - \alpha \log p(y_1) + \alpha \log q(y_2)\right)p(y_1)^\alpha q(y_2)^{1-\alpha}$$
$$- \left(q(y_1) - q(y_2)\right)\left(-1 + \sum_k q(k)^{1-\alpha}\left(-p(k)^\alpha + q(k)^\alpha\right)\right)$$
$$- \alpha \sum_k \left(-\log q(k)\, q(k)^{1-\alpha}\left(-p(x_k)^\alpha + q(x_k)^\alpha\right)\right.$$
$$\left. + q(k)^{1-\alpha}\left(-\log p(k)\, p(k)^\alpha + \log q(k)\, q(k)^\alpha\right)\right)\Bigg)\Bigg]. \tag{122}$$

Noting that the sign of $f'(\alpha)$ depends solely on the numerator, let $h(\alpha)$ denote its numerator. Differentiating $h(\alpha)$ with respect to $\alpha$, we obtain:

$$h'(\alpha) = \alpha p(y_1)^\alpha q(y_1)^{-\alpha} q(y_2)^{-\alpha}\Big[-\left(\log p(y_1) - \log q(y_2)\right)^2 q(y_1)^\alpha q(y_2)$$
$$+ \left(\log p(y_1) - \log q(y_1)\right)^2 q(y_1) q(y_2)^\alpha\Big]$$
$$\underbrace{- \alpha \left(q(y_1) - q(y_2)\right)\sum_k q(k)^{1-\alpha} p(k)^\alpha \left(\log q(k) - \log p(k)\right)^2}_{\geq 0} \tag{123}$$
$$\leq \alpha p(y_1)^\alpha q(y_1)^{-\alpha} q(y_2)^{-\alpha}\Big[-\left(\log p(y_1) - \log q(y_2)\right)^2 q(y_1)^\alpha q(y_2)$$
$$+ \left(\log p(y_1) - \log q(y_1)\right)^2 q(y_1) q(y_2)^\alpha\Big].$$

From Lem.G.9, it can be concluded that for any $\alpha$, there exists a point $c_\alpha \in (q(y_2), q(y_1))$ such that when $p(y_1) > c_\alpha$, $h'(\alpha) < 0$. Therefore, let $\delta_1 = \max(c_\alpha)$ for $\alpha \in [0, 1]$. Then, when $p(y_1) > \delta_1$, $h(\alpha)$ is monotonically decreasing for $\alpha \in [0, 1]$.

Noting that as $\alpha \to 0^+$, we have $\lim_{\alpha \to 0^+} h(\alpha) = 0$. Hence, $h(\alpha) < 0$ when $\alpha > 0$. In this way, we have proven that $f'(\alpha) < 0$, *i.e.*, $\Delta^\alpha(y_1, y_2)$ increases as $\alpha$ decreases.

$\square$

### G.5. Proof of Proposition 4.2

**Restate of Proposition 4.2** *The updates induced by $\alpha$-$\beta$-divergence for $q_t$ within one gradient descent step are given by:*

$$\left|\mathsf{LogR}_t^{(\alpha,\beta)}(y)\right| \leq \eta \underbrace{q_t(y)^\beta}_{(a)} \underbrace{\left|\frac{p(y)^\alpha - q_t(y)^\alpha}{\alpha}\right|}_{(b)}$$
$$+ q_t(y) \sum_k \underbrace{q_t(k)^\beta}_{(a)} \underbrace{\left|\frac{p(k)^\alpha - q_t(k)^\alpha}{\alpha}\right|}_{(b)} + \left|\mathsf{N}_t^{(\alpha,\beta)}(y)\right|,$$

*where $\mathsf{N}_t^{\alpha,\beta}(y)$ denotes constant normalization factor independent of $y$.*

*Proof.* The formula for the $\alpha$-$\beta$-divergence is:

$$\mathbb{D}_{\mathrm{AB}}^{(\alpha,\beta)}(p \parallel q) \triangleq -\frac{1}{\alpha\beta} \sum_k \left[ p(k)^\alpha q(k)^\beta - \frac{\alpha}{\alpha+\beta} p(k)^{\alpha+\beta} - \frac{\beta}{\alpha+\beta} q(k)^{\alpha+\beta} \right]. \tag{124}$$

Using the chain rule, we have:

$$\frac{\partial}{\partial f_y} \mathbb{D}_{\mathrm{AB}}^{(\alpha,\beta)}(p \parallel q) = \sum_k \frac{\partial \mathbb{D}_{\mathrm{AB}}^{(\alpha,\beta)}(p \parallel q)}{\partial q(k)} \frac{\partial q(k)}{\partial f_y}, \tag{125}$$

where

$$\frac{\partial \mathbb{D}_{\mathrm{AB}}^{(\alpha,\beta)}(p \parallel q)}{\partial q(y)} = -\frac{1}{\alpha} \left( p(y)^\alpha q(y)^{\beta-1} - q(y)^{\alpha+\beta-1} \right), \tag{126}$$

and

$$\frac{\partial q(y)}{\partial f_y} = \frac{\partial}{\partial f_y} \frac{e^{f_y}}{\sum_k e^{f_k}}. \tag{127}$$

Combining Lem. G.2, the Eq. 125 can be expressed as

$$\begin{aligned}
\frac{\partial}{\partial f_y} \mathbb{D}_\alpha(p \parallel q) &= \sum_{k \neq y} \frac{\partial \mathbb{D}_\alpha(p \parallel q)}{\partial q(k)} \frac{\partial q(k)}{\partial f(y)} + \frac{\partial \mathbb{D}_\alpha(p \parallel q)}{\partial q(y)} \frac{\partial q(y)}{\partial f_j} \\
&= -\frac{1}{\alpha} \left[ q(y)^\beta \left( p(y)^\alpha - q(y)^\alpha \right) + q(y) \left( \sum_k q(k)^\beta (q(k)^\alpha - p(k)^\alpha) \right) \right].
\end{aligned} \tag{128}$$

Now, consider using the gradient descent method to update the loss function $\ell$ with respect to the logits $f_y^t$, then the distribution at the next step $p^{t+1}$ is given by:

$$\begin{aligned}
q_{t+1}(y) &= \frac{\exp(f_y^{t+1})}{\sum_k \exp(f_k^{t+1})} \\
&= \frac{\exp(f_y^t - \eta \nabla_{f_y^t} \ell)}{\sum_k \exp(f_k^t - \eta \nabla_{f_k^t} \ell)} \\
&= q_t(y) \cdot \frac{\exp(-\eta \nabla_{f_y^{t+1}} \ell)}{\sum_k q_t(k) \exp(-\eta \nabla_{f_k^t} \ell)}.
\end{aligned} \tag{129}$$

Now, substituting the gradient formula of the $\alpha$-$\beta$-divergence, the characterization of $q_{t+1}^{(\alpha,\beta)}(x_j)$ is obtained as:

$$q_{t+1}^{(\alpha,\beta)}(y) = q_t(y) \cdot \frac{\exp\left( \frac{\eta}{\alpha} \left[ q(y)^\beta \left( p(y)^\alpha - q(y)^\alpha \right) + q(y) \left( \sum_k q(k)^\beta (q(k)^\alpha - p(k)^\alpha) \right) \right] \right)}{\sum_i q_t(i) \exp\left( \frac{\eta}{\alpha} \left[ q(i)^\beta \left( p(i)^\alpha - q(i)^\alpha \right) + q(i) \left( \sum_k q(k)^\beta (q(k)^\alpha - p(k)^\alpha) \right) \right] \right)}. \tag{130}$$

Observing that the denominator serves as a normalization constant, this can be rewritten as:

$$\frac{q_{t+1}^{(\alpha,\beta)}(y)}{q_t(y)} \propto \exp\left( \frac{\eta}{\alpha} \left[ q(y)^\beta \left( p(y)^\alpha - q(y)^\alpha \right) + q(y) \left( \sum_k q(k)^\beta (q(k)^\alpha - p(k)^\alpha) \right) \right] \right). \tag{131}$$

Taking the logarithm on both sides, we get:

$$\log \frac{q_{t+1}^{(\alpha,\beta)}(y)}{q_t(y)} = \eta \underbrace{\left[ q_t(y)^\beta \left( \frac{p(y)^\alpha - q_t(y)^\alpha}{\alpha} \right) + q_t(y) \sum_k q_t(k)^\beta \left( \frac{q_t(k)^\alpha - p(k)^\alpha}{\alpha} \right) \right]}_{-\nabla_{f_y} \ell} + \mathsf{N}_t^{(\alpha,\beta)}(y), \tag{132}$$

where $\mathsf{N}_t^{(\alpha,\beta)}(y)$ denotes constant normalization factors independent of $y$. We can further derive that

$$\left| \log \frac{q_{t+1}^{(\alpha,\beta)}(y)}{q_t(y)} \right| \leq \eta \underbrace{q_t(y)^\beta}_{(a)} \underbrace{\left| \frac{p(y)^\alpha - q_t(y)^\alpha}{\alpha} \right|}_{(b)} + q_t(y) \sum_k \underbrace{q_t(k)^\beta}_{(a)} \underbrace{\left| \frac{p(k)^\alpha - q_t(k)^\alpha}{\alpha} \right|}_{(b)} + \left| \mathsf{N}_t^{(\alpha,\beta)}(y) \right|. \tag{133}$$

This completes the proof. $\qquad\square$

### G.6. Proof of Theorem D.1

**Restate of Theorem D.1.** *Let $q_{t+1}^{(\alpha,\beta)}(y)$ be the distribution obtained after one gradient step, starting from $q_t$ using the $\alpha$-$\beta$-divergence. Define $\Delta_t^{(\alpha,\beta)}$ as the difference of log mass ratios across two classes $y_1$ and $y_2$, obtained from the $\alpha$-$\beta$-divergence:*

$$\Delta_t^{(\alpha,\beta)}(y_1, y_2) \triangleq \mathsf{LogR}_t^{(\alpha,\beta)}(y_1) - \mathsf{LogR}_t^{(\alpha,\beta)}(y_2).$$

*We have the following (for appropriate positive constants $\zeta$, $\delta_1$, $\delta_2$, and any real numbers $\alpha_1$ and $\alpha_2$ in the range $[0, 1]$ satisfying $\alpha_1 < \alpha_2$):*

1. *$\alpha$-$\beta$-divergence transfers probability mass from overestimated classes to underestimated classes more aggressively as $\alpha$ decreases. If $y_1$ and $y_2$ are such that $\delta_1 < q_t(y_1) = q_t(y_2) \le p(y_1)$ (where $\delta_1 > 0$), and $p(y_1) \ge p(y_2) + \zeta$, it holds that $\Delta_t^{(\alpha_1,\beta)}(y_1, y_2) \ge \Delta_t^{(\alpha_2,\beta)}(y_1, y_2)$.*

2. *$\alpha$-$\beta$-divergence reduces the probability mass of classes with larger error $|p(y) - q_t(y)|$ more aggressively as $\alpha$ decreases. If $y_1$ and $y_2$ are such that $p(y_1) < q_t(y_1) = q_t(y_2) \le 1 - \delta_2$ (where $\delta_2 > 0$), and $p(y_1) \ge p(y_2) + \zeta$, it holds that $\Delta_t^{(\alpha_1,\beta)}(y_1, y_2) \ge \Delta_t^{(\alpha_2,\beta)}(y_1, y_2)$.*

3. *The $\alpha$-$\beta$-divergence becomes more (less) preferential in focusing the error on classes with higher student confidence as $\beta$ increases (decreases) when reducing $\left|\mathsf{LogR}_t^{(\alpha,\beta)}(y)\right|$.*

*Proof.* **First, we prove Case 1**. Note that $q_t(y_1) = q_t(y_2) = q(x)$, Based on Eq. 132, we have

$$
\begin{aligned}
\Delta_t^{(\alpha,\beta)} &= \eta q(x)^\beta \cdot \frac{p(y_1)^\alpha - p(y_2)^\alpha}{\alpha} \\
&= \eta q(x)^\beta \cdot \int_{p(y_2)}^{p(y_1)} t^{\alpha-1}\, dt.
\end{aligned}
\tag{134}
$$

Taking the derivative with respect to $\alpha$, we get

$$
\frac{\partial}{\partial \alpha}\Delta_t^{(\alpha,\beta)} = \eta q(x)^\beta \cdot \int_{p(y_2)}^{p(y_1)} t^{\alpha-1}\ln t\, dt.
\tag{135}
$$

Note that $p(y_1) \le 1$ and $p(y_2) \le 1$, we have

$$
\frac{\partial}{\partial \alpha}\Delta_t^{(\alpha,\beta)} \le 0.
\tag{136}
$$

since $\ln t \le 0$ when $t \in (0, 1]$. Therefore, we have $\Delta_t^{(\alpha_1,\beta)} > \Delta_t^{(\alpha_2,\beta)}$ when $\alpha_1 < \alpha_2$.

The proof of **Case 2** is similar to **Case 1** and thus is omitted.

Finally, we prove **Case 3**. Recall that when reducing $\left|\mathsf{LogR}_t^{(\alpha,\beta)}(y)\right|$, we have the following relationship:

$$
\left|\mathsf{LogR}_t^{(\alpha,\beta)}(y)\right| \le \eta \underbrace{q_t(y)^\beta}_{(a)} \underbrace{\left|\frac{p(y)^\alpha - q_t(y)^\alpha}{\alpha}\right|}_{(b)} + \eta q_t(y)\sum_k \underbrace{q_t(k)^\beta}_{(a_1)} \underbrace{\left|\frac{p(k)^\alpha - q_t(k)^\alpha}{\alpha}\right|}_{(b_1)} + \left|\mathsf{N}_t^{(\alpha,\beta)}(y)\right|,
$$

where terms $(a)$ and $(a_1)$ act as weighting functions. Therefore, selecting a larger (smaller) $\beta$ will place more (less) emphasis on errors from classes with higher student confidence $q_t(y)$, as shown in Fig. 1(c). □

**Remark.** Case 1 and Case 2 show that selecting a smaller $\alpha$ leads to a more aggressive reduction of errors across classes by shifting the probability mass from overestimated to underestimated classes. On the other hand, Case 3 shows that increasing $\beta$ emphasizes minimizing errors more in classes with higher student confidence.

---

**Algorithm 1** alpha beta Divergence Function

---

**Input:** Student distribution $q_\theta$, Teacher distribution $p$, and Hyperparameters $\alpha$ and $\beta$

**Output:** Divergence value $\mathbb{D}_{\mathrm{AB}}^{(\alpha,\beta)}(p\|q_\theta)$

1: **return** $-\frac{1}{\alpha\beta}\sum_k \left( p(k)^\alpha q(k)^\beta - \frac{\alpha}{\alpha+\beta}p(k)^{\alpha+\beta} - \frac{\beta}{\alpha+\beta}q(k)^{\alpha+\beta} \right)$

---

**Algorithm 2** Generalized distillation framework with $\alpha$-$\beta$-divergence.

---

**Input:** Dataset $\mathcal{D}$ with input-target pair $\{\{\boldsymbol{x}_n, \boldsymbol{y}_n\}\}_{n=1}^N$, Teacher $f_T$, Student $f_S$, loss weight $\lambda$, $\alpha$-$\beta$-divergence function $\mathbb{D}_{\mathrm{AB}}^{(\alpha,\beta)}$ in Algo. 1, and Hyperparameters $\alpha$ and $\beta$

**Output:** Trained student model $f_S$

1: **for** each $(\boldsymbol{x}_n, \boldsymbol{y}_n)$ in $\mathcal{D}$ **do**

2:    $f^T \leftarrow f_T(\boldsymbol{x}_n), f^S \leftarrow f_S(\boldsymbol{x}_n)$

3:    $p \leftarrow \texttt{softmax}(f^T)$

4:    $q_\theta \leftarrow \texttt{softmax}(f^S)$

5:    $\ell_{\mathrm{KD}} \leftarrow \mathbb{D}_{\mathrm{AB}}^{(\alpha,\beta)}(p\|q_\theta)$

6:    Update $f_S$ towards minimizing $\ell_{CE}(\boldsymbol{y_n}, q_\theta) + \lambda\ell_{\mathrm{KD}}(p, q_\theta)$

7: **end for**

---

# H. Algorithm Protocol

Algo. 1 and Algo. 2 give the algorithmic protocol of our framework, which is easy to implement and applicable to common KD downstream tasks.

# I. Additional Experiment Settings

In this section, we provide a more detailed description of the experimental protocol.

## I.1. Natural Language Processing Tasks

### I.1.1. DATASETS

Following Gu et al. (2024a); Ko et al. (2024), we select 14K samples from `databricks-dolly-15k` (Conover et al., 2023) for training and 500 samples each for validation and testing. After distillation, the models are evaluated on five task-agnostic instruction-following benchmarks: Dolly-evaluation, Self-Instruct, Vicuna-evaluation, Super-Natural Instructions, and Unnatural Instruction. The details for each dataset are as follows:

- **databricks-dolly-15k** (Conover et al., 2023): An open-source dataset of instruction-following records created by thousands of Databricks employees. It includes several behavioral classes from Ouyang et al. (2022), such as brainstorming, classification, closed QA, generation, information extraction, open QA, and summarization.

- **Self-Instruct** (Wang et al., 2023): A framework that enhances a language model's instruction-following by using its own outputs to generate extensive instructional data. It contains 52K instructions and 82K input-output pairs for tuning, 252 expert-written tasks for practical use, and 50K public dataset examples for benchmarking.

- **Vicuna**: Utilizes 80 challenging questions for evaluating Vicuna, following (Ko et al., 2024; Gu et al., 2024a).

- **Super-Natural Instruction** (Wang et al., 2022a): A benchmark of 1,616 diverse NLP tasks with expert-written instructions, covering 76 task types. The test set includes 9K samples across 119 tasks.

- **Unnatural Instruction** (Honovich et al., 2023): AI-generated dataset with 240K instructions created with minimal human input, proving AI data can match human data for training language models. The core set has 60K samples.

For experiments on these datasets, we use ROUGE-L (Lin, 2004) as the evaluation metric. ROUGE-L measures the quality of generated text by calculating the Longest Common Subsequence (LCS) between the generated text $\boldsymbol{y}$ and the reference

text $\boldsymbol{x}$. A higher ROUGE-L score indicates that the generated text is more similar to the reference text. The metric is computed based on the harmonic mean of *recall* $R_{\mathrm{LCS}}$ and *precision* $P_{\mathrm{LCS}}$, defined as:

$$R_{\mathrm{LCS}} = \frac{\mathrm{LCS}(\boldsymbol{x}, \boldsymbol{y})}{L_{\boldsymbol{x}}},$$

$$P_{\mathrm{LCS}} = \frac{\mathrm{LCS}(\boldsymbol{x}, \boldsymbol{y})}{L_{\boldsymbol{y}}},$$

$$\mathrm{ROUGE\text{-}L} = \frac{2 \cdot R_{\mathrm{LCS}} \cdot P_{\mathrm{LCS}}}{R_{\mathrm{LCS}} + P_{\mathrm{LCS}}}.$$

Here, $\mathrm{LCS}(\boldsymbol{x}, \boldsymbol{y})$ is the length of the longest common subsequence, $L_{\boldsymbol{x}}$ is the length of the reference text, and $L_{\boldsymbol{y}}$ is the length of the generated text.

### I.1.2. COMPETITORS

Here we give a more detailed summary of the competitors mentioned in the experiments and their SGOs approaches (if exists).

- **SFT** is supervised fine-tuning of student model using ground-truth on the **Fixed** dataset (using predefined input-output pairs).

- **KD** (Hinton, 2015) trains the student distribution to mimic the teacher distribution on the **Fixed dataset** using FKLD.

- **SeqKD** (Kim & Rush, 2016) maximizes the likelihood of high probability sequences generated by the teacher, and can be viewed as SFT on teacher-generated outputs.

- **MiniLLM** (Gu et al., 2024a) trains on the student-generated sentences (SGOs) and uses an **On-policy** gradient method. Their distillation object is to minimize the RKLD between the teacher and student distributions.

- **GKD** (Agarwal et al., 2024) uses the generalized Jensen-Shannon divergence ($\mathbb{D}_{\mathrm{JSD}(\beta)}(p\|q_\theta) = \beta\mathbb{D}(p\|\beta p + (1 - \beta)q_\theta) + (1 - \beta)\mathbb{D}(q_\theta\|\beta p + (1 - \beta)q_\theta)$), training on a **Mixture** of datasets, either teacher-generated or ground-truth, and on-policy student-generated sequences.

- **DISTILLM** (Ko et al., 2024) uses Skew KL ($\mathbb{D}(p\|\alpha p + (1 - \alpha q_\theta))$) or Skew RKL ($\mathbb{D}(q_\theta\|\alpha q_\theta + (1 - \alpha p))$) and reports the better performing one. They train on a mixed dataset consisting of fixed outputs and student-generated outputs. Additionally, they use an **Adaptive** off-policy method to determine whether to use student-generated outputs for training based on validation loss, thereby removing noisy SGOs data.

### I.1.3. IMPLEMENTATION DETAILS

**Training.** For training the teacher and student models, we used four RTX 3090 24GB GPUs. Our experimental setup for training LMs on `databricks-dolly-15k` primarily follows the experimental setup for Ko et al. (2024). We search for the learning rates in {5e-4, 1e-4, 5e-5}, the batch sizes in {4, 8, 16} within the possible maximum batch size for 3090 24GB GPUs, and train these models for 20 epochs. We fully use the distillation loss for the instruction-following dataset and language modeling loss for OpenWebText (Gokaslan et al., 2019) corpus. The checkpoints of each student are selected by the ROUGE-L scores on the validation set. For all teacher-student configurations, we set $\alpha = 0.2$ and $\beta = 0.7$. Additionally, the cross-entropy loss $\ell_{\mathrm{CE}}$ was not considered to ensure a fair comparison with previous methods.

To ensure a fair comparison, for other competitors, we rerun them (with the necessary hyperparameter tuning) and select the best-performing checkpoint on the validation set.

**Evaluation.** For evaluating the teacher and student models, we applied a single RTX 3090 24GB GPU. Following (Ko et al., 2024; Gu et al., 2024a), We adopt a prompt template as shown in Fig. 7. We sample the responses from each model using a temperature of 1.0, a max-length limit of 512, and five random seeds (*i.e.*, {10, 20, 30, 40, 50}).

```
Below is an instruction that describes a task.
Write a response that appropriately completes the request.

### Instruction:
{instruction}

### Input:
{input}

### Response:
```

*Figure 7.* The prompt template for training and evaluation of instruction-following task experiments from (Ko et al., 2024; Gu et al., 2024a).

## I.2. Vision Tasks

### I.2.1. DATASETS

In this section, we provide detailed descriptions of the image datasets used.

- **CIFAR-100**: It is a generic image dataset consisting of 60,000 32×32 color images across 100 classes, with 600 images per class. It is further split into 50,000 training images and 10,000 test images.

- **ImageNet** (Deng et al., 2009): It is a widely recognized object classification dataset containing approximately 1.28 million training images and 50,000 test images across 1,000 object classes. The images are sourced from the web and organized using the WordNet hierarchy, making it a standard benchmark for evaluating object recognition models.

- **Caltech101** (Fei-Fei et al., 2004): It is an object classification dataset with 101 classes and a background class, containing approximately 7,650 training images and 3,300 test images. The images vary significantly in scale, orientation, and lighting conditions.

- **OxfordPets** (Parkhi et al., 2012): It is a fine-grained pet classification dataset with 37 pet breed classes, featuring nearly equal numbers of training (3,680) and test (3,669) images. It also includes pixel-level segmentation masks.

- **StanfordCars** (Krause et al., 2013): It is a fine-grained car model recognition dataset with 196 classes, based on make, model, and year. It contains 8,144 training images and 8,041 test images, capturing diverse vehicle angles and environments.

- **Flowers102** (Nilsback & Zisserman, 2008): It is a dataset of 102 flower species for fine-grained classification tasks. It includes 6,149 training images and 1,020 test images, posing challenges in distinguishing visually similar flower classes.

- **Food101** (Bossard et al., 2014): It is a fine-grained food classification dataset with 101 dish classes, comprising 75,750 training images and 25,250 test images. It poses challenges in recognizing overlapping ingredients and presentation styles.

- **FGVCAircraft** (Maji et al., 2013): It is a fine-grained aircraft classification dataset with 100 classes, distinguishing between models and manufacturers. It contains 6,667 training images and 3,333 test images.

- **SUN397** (Xiao et al., 2010): It is a comprehensive scene recognition dataset with 397 classes, including natural landscapes, indoor spaces, and urban environments. It contains approximately 50,000 training images and 50,000 test images.

- **UCF101** (Soomro, 2012): It is a video dataset for action recognition, featuring 101 action classes ranging from sports to daily activities. It contains approximately 9,500 training clips and 3,700 test clips, collected from YouTube.

- **DTD** (Soomro, 2012): It is a texture classification dataset with 47 texture classes described using human-interpretable attributes. It contains 3,760 training images and 1,880 test images.

- **EuroSAT** (Helber et al., 2019): It is a satellite image dataset for land-use and land-cover classification, with 10 classes such as agricultural areas, forests, and urban regions. It contains 21,600 training images and 5,400 test images.

For experiments on these datasets, we follow the popular setup in classification tasks to use accuracy as the evaluation metric.

### I.2.2. COMPETITORS

In this section, we provide a more in-depth overview of the competitors discussed in the vision experiments.

First, we introduce the distillation-based methods:

- **KD** (Hinton, 2015) directly minimizes the FKLD between the student and teacher distributions to transfer knowledge.

- **DKD** (Zhao et al., 2022) uses FKLD for distillation, where the knowledge of the teacher's distribution is decoupled into target class and non-target class knowledge for separate learning.

- **LSD** (Sun et al., 2024) uses FKLD for distillation in their experiments, where they first normalize the logit vector before obtaining the model output distribution.

- **TTM** (Zheng & Yang, 2024) uses FKLD for distillation, and they also introduce Rényi entropy as a regularization to make the student distribution smoother.

Next, we describe the SFT-based methods:

- **CLIP** is supervised fine-tuning using ground-truth on standard datasets.

- **CoCoOp** (Zhou et al., 2022) enhances new class performance by transforming the unified context into an instance-adaptive context, where each sample is assigned a specific prompt that focuses on its unique features or attributes.

- **MaPLe** (Khattak et al., 2023a) improves vision-language alignment by simultaneously adapting both the text and image encoders in CLIP using hierarchical prompts.

- **PromptSRC** (Khattak et al., 2023b) ensures better performance on both base and new classes by minimizing the task cross-entropy loss and the FKLD between the output distribution of the model and the pre-trained model.

### I.2.3. IMPLEMENTATION DETAILS

We conduct all vision experiments on a single RTX 3090 GPU. The detailed experimental setups are as follows.

**Standard Training-Evaluation setup.** In this experimental setup, we consider model architectures including VGG (Simonyan, 2014), ResNet (He et al., 2016), and WideResNet (Zagoruyko, 2016). Following (Zheng & Yang, 2024; Sun et al., 2024; Zhao et al., 2022), we train the student models on all class samples. We also consider a standard training data augmentation scheme including padding 4 pixels prior to random cropping and horizontal flipping. We set the batch size as 64 and the initial learning rate as 0.05. We train the model for 240 epochs, in which the learning rate is decayed by 10 every 30 epochs after 150 epochs. We use stochastic gradient descent (SGD) as the optimizer with weight decay 5e-4 and momentum 0.9.

For **evaluation**, we report the average accuracy across all classes on the test set. We list the hyperparameters $\alpha$ and $\beta$ used across the above experiments in Tab. 4. In our ABKD, the weight $\lambda$ of $\ell_{KD}$ is set to the default value 32. For those re-implemented methods, we only adjust $\alpha$ and $\beta$ and follow the other hyperparameters as reported in their original papers.

**Base-to-New setup.** In this experimental setup, we use the ViT-L/14 CLIP model as the teacher and the ViT-B/16 CLIP model as the student. We adopt the recently popular prompt tuning setup for CLIP, as it performs sufficiently well across many tasks, despite freezing most of the model parameters and training only a subset of the learnable prompt tokens. We split the training and testing datasets into base and new classes same as previous work (Khattak et al., 2023a; Kim et al., 2024). Tab. 5 provides the details of the number of images used for training on the base-to-new setup.

*Table 4.* Hyperparameters for different architecture distillations on CIFAR-100.

| Teacher
Student | WRN-40-2
WRN-16-2 | WRN-40-2
WRN-40-1 | resnet56
resnet20 | resnet110
resnet20 | resnet110
resnet32 | resnet32x4
resnet8x4 | vgg13
vgg8 | resnet110
resnet44 |
|---|---|---|---|---|---|---|---|---|
| ABKD | $\alpha = 0.6, \beta = 0.5$ | $\alpha = 0.9, \beta = 0.2$ | $\alpha = 0.8, \beta = 0.3$ | $\alpha = 0.8, \beta = 0.3$ | $\alpha = 0.7, \beta = 0.4$ | $\alpha = 0.5, \beta = 0.5$ | $\alpha = 0.9, \beta = 0.2$ | $\alpha = 0.8, \beta = 0.3$ |
| ABDKD | $\alpha = 0.8, \beta = 0.4$ | $\alpha = 1.0, \beta = 0.2$ | $\alpha = 1.0, \beta = 0.2$ | $\alpha = 0.8, \beta = 0.3$ | $\alpha = 0.9, \beta = 0.3$ | $\alpha = 0.8, \beta = 0.3$ | $\alpha = 0.7, \beta = 0.4$ | $\alpha = 0.8, \beta = 0.3$ |
| ABLSD | $\alpha = 0.9, \beta = 0.1$ | $\alpha = 0.8, \beta = 0.4$ | $\alpha = 0.9, \beta = 0.3$ | $\alpha = 1.2, \beta = -0.1$ | $\alpha = 0.9, \beta = 0.2$ | $\alpha = 1.2, \beta = -0.2$ | $\alpha = 1.0, \beta = 0.2$ | $\alpha = 1.0, \beta = 0.2$ |
| ABTTM | $\alpha = 0.8, \beta = 0.3$ | $\alpha = 1.0, \beta = 0.1$ | $\alpha = 0.7, \beta = 0.5$ | $\alpha = 0.9, \beta = 0.2$ | $\alpha = 0.8, \beta = 0.3$ | $\alpha = 0.8, \beta = 0.3$ | $\alpha = 0.7, \beta = 0.5$ | $\alpha = 0.8, \beta = 0.2$ |

*Table 5.* Number of images used for distillation and testing per-dataset. To ensure a fair comparison, we follow the same data split as prior arts (Kim et al., 2024).

| Dataset | ImageNet | Caltech101 | OxfordPets | StanfordCars | Flowers102 | Food101 | FGVCAircraft | SUN397 | DTD | EuroSAT | UCF101 |
|---|---|---|---|---|---|---|---|---|---|---|---|
| Train | 1,281,167 | 4,128 | 2,944 | 6,509 | 4,093 | 50,500 | 3,334 | 15,880 | 2,820 | 13,500 | 7,639 |
| Test Base | 25,000 | 1,549 | 1,881 | 4,002 | 1,053 | 15,300 | 1,666 | 9,950 | 864 | 4,200 | 1,934 |
| Test New | 25,000 | 916 | 1,788 | 4,039 | 1,410 | 15,000 | 1,667 | 9,900 | 828 | 3,900 | 1,849 |

The teacher is pre-trained using the PromptSRC (Khattak et al., 2023b) method, following which it is fine-tuned on the base classes using ground truth supervision. Following (Kim et al., 2024), all distillation-based methods use the unlabeled training set to train students for a fair comparison, and we search for $\lambda$ in $\{100, 200, 300, 500, 1000, 2000, 3000\}$. We set the prompt depth to 9 and the vision and language prompt lengths to 4. We use SGD as the optimizer. All student models are trained for 20 epochs with a batch size of 8 and a learning rate of 0.005. We follow the data augmentation scheme as in Khattak et al. (2023b), *i.e.*, random resized cropping and random flipping. The text prompts of the first layer are initialized with the word embeddings of "a photo of a {classname}".

For **evaluation**, we report the model's accuracy on both the base classes and the new classes separately. Additionally, we report the Harmonic Mean (HM) of the two accuracies, defined as:

$$\text{HM} = \frac{2 \times \text{BaseAcc} \times \text{NewAcc}}{\text{BaseAcc} + \text{NewAcc}}. \tag{137}$$

We list the hyperparameters $\alpha$ and $\beta$ used across different datasets in Tab. 7. In addition, since LSD, DKD, and TTM did not report performance under the base-to-new setting, we reran their source code and report the best results (with necessary hyperparameter tuning).

*Table 6.* Hyperparameters for different image datasets

| Dataset | ImageNet | Caltech101 | OxfordPets | StanfordCars | Flowers102 | Food101 | FGVCAircraft | SUN397 | DTD | EuroSAT | UCF101 |
|---|---|---|---|---|---|---|---|---|---|---|---|
| $\alpha$ | 0.5 | 0.8 | 0.8 | 0.6 | 0.9 | 0.5 | 0.6 | 0.8 | 1.0 | 0.6 | 0.8 |
| $\beta$ | 0.5 | 0.2 | 0.4 | 0.4 | 0.1 | 0.5 | 0.5 | 0.2 | 0.2 | 0.5 | 0.2 |

# J. Additional Experiment Analysis

In this section, we present additional experimental results on language and vision tasks.

## J.1. Natural Language Processing Tasks

### J.1.1. EFFECTS OF SGOS

Prior arts (Ko et al., 2024; Agarwal et al., 2024) highlight that existing KD methods suffer from distribution mismatch between the output sequences seen during training and those generated by the student during inference in auto-regressive language models. To address this, these works incorporate student-generated outputs (SGOs) along with teacher feedback (*i.e.*, token-level predict distribution for these sentences) during training, leading to significant improvements. To assess the applicability of our framework, we evaluate the performance after training with different SGO strategies, as shown in Tab. 8. The results indicate that by integrating these promising techniques, our framework achieves further improvements across most datasets compared to training with fixed data.

Table 7. Hyperparameters for different instruction datasets

| Dataset | Dolly Eval | Self-Instruct | Vicuna Eval | Super-Natural | Unnatural |
|---------|-----------|---------------|-------------|---------------|-----------|
| $\alpha$ | 0.2 | 0.2 | 0.2 | 0.2 | 0.2 |
| $\beta$ | 0.7 | 0.7 | 0.7 | 0.7 | 0.7 |

Table 8. Effects of our framework using different SGOs strategies (*i.e.*, On-policy, Mixed, and Adaptive Off-policy). Fixed denotes that our framework uses only the original dataset for training without augmentation. Following (Ko et al., 2024), in the mixed strategy, we apply the on-policy optimization method with a probability of 0.5. Otherwise, we sample from the fixed dataset.

| Method | Dolly Eval | Self-Instruct | Vicuna Eval | Super-Natural | Unnatural |
|--------|-----------|---------------|-------------|---------------|-----------|
| Prior SOTA result | 25.32 (0.14) | 12.49 (0.56) | 17.30 (0.41) | 23.76 (0.38) | 25.79 (0.08) |
| Fixed + Ours | 25.65 (0.24) | 13.47 (0.42) | 16.06 (0.25) | 26.47 (0.31) | 29.32 (0.08) |
| On-policy (Gu et al., 2024a) + Ours | 25.96 (0.42) | 13.44 (0.37) | 17.32 (0.38) | 26.86 (0.26) | 29.57 (0.13) |
| Mixed (Agarwal et al., 2024) + Ours | 26.49 (0.23) | **14.62** (0.27) | 17.14 (0.26) | 27.54 (0.44) | 30.98 (0.09) |
| Adaptive Off-policy (Ko et al., 2024) + Ours | **26.58** (0.18) | 14.25 (0.25) | **17.79** (0.35) | **27.79** (0.26) | **31.13** (0.12) |

### J.1.2. DISTILLING FROM STRONGER TEACHER

Recent research (Cho & Hariharan, 2019; Huang et al., 2022) shows that as the size of the teacher model increases, the distillation performance does not always improve for the student models and may even degrade due to the capacity gap between them. It is not clear how our framework performs when scaling up the teacher models' sizes. To this end, we report the performance of our method using teacher models of varying sizes while keeping the student model size fixed, as shown in Tab. 9. From the results, we have the following observations: 1) Our $\alpha$-$\beta$-divergence consistently outperforms FKLD and RKLD across different teacher models; 2) FKLD and RKLD fail to ensure that the student model consistently benefits from the rich supervision provided by larger teacher models and thus leads to suboptimal performance. 3) In contrast, the $\alpha$-$\beta$-divergence can maintain the student model's performance nearly positively correlated with teacher model size by smoothly interpolating between FKLD and RKLD.

Table 9. Performance of *GPT-2* on five task-agostic instruction-following datasets with different teacher model sizes.

| Method | Dolly Eval | Self-Instruct | Vicuna Eval | Super-Natural | Unnatural |
|--------|-----------|---------------|-------------|---------------|-----------|
| *GPT-2 Medium (0.3B) $\rightarrow$ GPT-2 (0.1B)* | | | | | |
| FKLD | 23.68 (0.29) | 10.14 (0.53) | 15.44 (0.48) | 18.54 (0.30) | 20.44 (0.20) |
| RKLD | 24.66 (0.20) | 11.73 (0.31) | 15.27 (0.41) | 22.65 (0.25) | 25.27 (0.24) |
| $\alpha$-$\beta$-divergence (Ours) | **25.47** (0.25) | **13.13** (0.46) | **15.84** (0.21) | **26.29** (0.13) | **28.31** (0.14) |
| *GPT-2 Large (0.8B) $\rightarrow$ GPT-2 (0.1B)* | | | | | |
| FKLD | 20.01 (0.23) | 9.89 (0.59) | 14.98 (0.35) | 19.00 (0.26) | 18.72 (0.13) |
| RKLD | 25.27 (0.28) | 11.77 (0.24) | 14.78 (0.26) | 23.61 (0.36) | 26.41 (0.13) |
| $\alpha$-$\beta$-divergence (Ours) | **25.72** (0.52) | **13.08** (0.45) | **15.80** (0.62) | **26.44** (0.32) | **29.25** (0.10) |
| *GPT-2 XL (1.5B) $\rightarrow$ GPT-2 (0.1B)* | | | | | |
| FKLD | 23.80 (0.37) | 10.01 (0.75) | 15.25 (0.65) | 17.69 (0.26) | 18.99 (0.05) |
| RKLD | 24.77 (0.37) | 12.02 (0.48) | 15.06 (0.28) | 23.27 (0.29) | 26.01 (0.11) |
| $\alpha$-$\beta$-divergence (Ours) | **25.65** (0.24) | **13.47** (0.42) | **16.06** (0.25) | **26.47** (0.31) | **29.32** (0.08) |

### J.1.3. COMPARISON WITH MORE BASELINES

To further demonstrate the effectiveness of the proposed method, we additionally consider several KD baselines that were not included in the main text, namely (1) AlphaNet (Wang et al., 2021), (2) BDKD (Amara et al., 2022), (3) AKL (Wu et al., 2024), and (4) Jensen's KL (Binici et al., 2022).

The results are shown in Tab. 10. ABKD outperforms all other baselines across different benchmarks, with improvements ranging from 0.42 to 1.76. In addition, we visualize the performance dynamics of different methods throughout the training process, as shown in Fig. 8. The $\alpha$-$\beta$ divergence consistently achieves the highest performance throughout training, especially in the early stages, demonstrating its faster convergence ability.

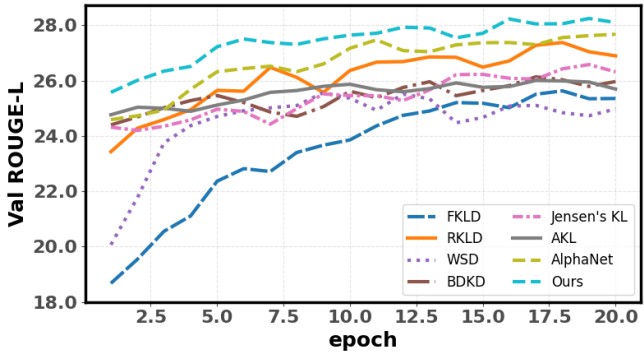

*Figure 8.* Performance on the validation set when distilling GPT-2 XL (1.5B) to GPT-2 (0.1B).

Finally, it is worth noting that although AlphaNet achieves performance comparable to ours, it requires tuning three hyperparameters (while ours only has two), which significantly increases the search overhead. In addition, a key advantage of our approach is that its hyperparameters have clear physical interpretations. This allows one to leverage inductive bias and follow principled guidelines to more efficiently search for suitable hyperparameters for the target task (App. D).

*Table 10.* ROUGE-L scores (↑) of different loss functions on five task-agnostic instruction-following datasets when distilling GPT-2 XL (1.5B) into GPT-2 (0.1B). We report the average and standard deviation of ROUGE-L scores across five random seeds [10, 20, 30, 40, 50].

| Loss Function | Dolly | Self-Instruct | Vicuna | Super-Natural | Unnatural |
|---|---|---|---|---|---|
| FKLD | 23.80 (0.37) | 10.01 (0.75) | 15.25 (0.65) | 17.69 (0.26) | 18.99 (0.05) |
| RKLD | 24.77 (0.37) | 12.02 (0.48) | 15.06 (0.28) | 23.27 (0.29) | 26.01 (0.11) |
| WSD | 23.33 (0.52) | 10.52 (0.47) | 14.83 (0.61) | 19.67 (0.13) | 21.21 (0.21) |
| BDKD | 23.94 (0.24) | 11.83 (0.39) | 15.21 (0.23) | 19.56 (0.23) | 21.66 (0.23) |
| Jensen-Shannon divergence | 23.79 (0.24) | 11.52 (0.18) | 15.35 (0.80) | 21.36 (0.17) | 21.97 (0.10) |
| AKL | 23.83 (0.59) | 10.87 (0.42) | 15.63 (0.66) | 20.07 (0.32) | 21.97 (0.13) |
| AlphaNet | 25.13 (0.27) | 12.46 (0.46) | 15.64 (0.40) | 25.27 (0.20) | 27.56 (0.15) |
| SKL | 25.01 (0.23) | 12.47 (0.29) | 15.98 (0.84) | 25.56 (0.31) | 27.51 (0.07) |
| SRKL | 25.75 (0.39) | 11.58 (0.49) | 15.56 (0.17) | 26.13 (0.25) | 27.37 (0.18) |
| $\alpha$-divergence | 25.15 (0.41) | 12.92 (0.22) | 15.60 (0.27) | 24.83 (0.21) | 27.81 (0.10) |
| $\beta$-divergence | 24.12 (0.38) | 11.18 (0.27) | 14.95 (0.33) | 20.98 (0.23) | 23.15 (0.14) |
| $\alpha$-$\beta$-divergence (Ours) | **25.65** (0.24) | **13.47** (0.42) | **16.06** (0.25) | **26.47** (0.31) | **29.32** (0.08) |

### J.1.4. LLAMA FAMILY DISTILLATION

The following analysis aims to investigate whether the proposed method remains effective in distillation experiments involving larger-scale models. To this end, we conducted distillation from OpenLLaMA2-7B (Touvron et al., 2023b) to 3B and compared our approach with various KD baselines.

The results are presented in Tab. 11. ABKD outperforms others by 0.65-3.26 ROUGE-L scores, especially excelling in Dolly and Unnatural.

### J.1.5. QUALITATIVE EVALUATION

In this section, we present several case studies to illustrate the effectiveness of ABKD. As shown in Tab. 17, ABKD is better at generating more accurate responses according to the predefined requirements specified in the instructions.

*Table 11.* ROUGE-L scores (↑) on five task-agnostic instruction-following datasets when distilling OpenLLaMA2-7B into OpenLLaMA2-3B. Experiments are conducted on eight RTX 3090 24GB GPUs. * indicates that SGOs are used.

| Method | Dolly | Self-Instruct | Vicuna | Super-Natural | Unnatural |
|---|---|---|---|---|---|
| SFT | 24.54 (0.51) | 16.80 (0.64) | 16.15 (0.15) | 29.29 (0.13) | 27.43 (0.21) |
| FKLD | 25.23 (0.44) | 18.90 (1.20) | 16.67 (0.35) | 31.68 (0.22) | 29.36 (0.13) |
| RKLD | 27.74 (0.45) | 20.61 (0.80) | 18.83 (0.40) | 35.31 (0.24) | 33.86 (0.16) |
| Jensen's KL | 26.28 (0.43) | 18.84 (0.66) | 17.81 (0.38) | 30.92 (0.12) | 29.79 (0.17) |
| BDKD | 26.78 (0.53) | 18.94 (0.68) | 17.81 (0.52) | 32.15 (0.34) | 30.89 (0.24) |
| AKL | 26.38 (0.41) | 17.69 (0.46) | 16.72 (0.48) | 33.02 (0.16) | 31.29 (0.08) |
| DISTILLM$^*$ | 28.24 (0.48) | 21.00 (0.72) | 19.12 (0.53) | 37.06 (0.35) | 35.05 (0.13) |
| AlphaNet | 28.11 (0.29) | 21.30 (0.63) | 18.70 (0.23) | 37.86 (0.44) | 35.40 (0.17) |
| **Ours (ABKD)** | **30.25** (0.37) | **22.39** (0.62) | **20.83** (0.42) | **38.51** (0.32) | **38.66** (0.10) |

*Table 12.* The distillation results of Qwen2.5-Math on English mathematical benchmarks. Models are evaluated with chain-of-thought prompting.

| BENCHMARK / MODEL | GSM8K | MATH | GaoKao 2023 En | Olympiad Bench | College Math | Avg. |
|---|---|---|---|---|---|---|
| Qwen2.5-Math-7B-Instruct (Teacher) | 95.5 | 82.8 | 66.8 | 38.5 | 37.7 | 64.3 |
| Qwen2.5-1.5B-Instruct (Student) | 73.3 | 54.9 | 45.9 | 18.9 | 30.3 | 44.7 |
| SeqKD | 75.8 | 57.3 | 47.3 | 17.7 | 31.3 | 45.9 |
| KD | 75.9 | 58.1 | 45.5 | **21.1** | 31.3 | 46.3 |
| ABKD | **77.4** | **58.6** | **48.5** | 20.4 | **32.0** | **47.4** |

## J.2. Vision Tasks

### J.2.1. DISTILLING FROM STRONGER TEACHER

To evaluate the potential of our framework to benefit from a larger teacher, we examine the distillation effect when different-sized teacher models are used to distill a student model of the same size, as shown in Tab. 13. Encouragingly, our framework consistently outperforms FKLD and RKLD, with performance gains remaining stable as the teacher model size increases.

*Table 13.* Performance of resnet20 on CIFAR-100 with different teacher model sizes. For a fair comparison, we set $\alpha = 0.8$ and $\beta = 0.3$ across all teacher-student configurations.

| Student | Teacher | Accuracy (%) | | | | |
|---|---|---|---|---|---|---|
| | | Student | Teacher | FKLD | RKLD | $\alpha$-$\beta$-divergence (Ours) |
| resnet20 | resnet32 | 69.06 | 71.93 | 71.03 (0.23) | 70.91 (0.29) | **71.46** (0.15) |
| | resnet44 | | 72.25 | 71.51 (0.11) | 71.29 (0.16) | **71.76** (0.25) |
| | resnet56 | | 72.34 | 70.66 (0.24) | 71.43 (0.16) | **71.79** (0.16) |
| | resnet110 | | 74.31 | 70.67 (0.27) | 71.41 (0.23) | **71.72** (0.18) |

### J.2.2. CROSS-ARCHITECTURE DISTILLATION

Although the analysis in the main text has demonstrated the effectiveness of the proposed method for distillation within the same architecture, it remains unclear how much improvement our method can achieve when the teacher and student have different architectures. To this end, we performed distillation from ResNet50 to VGG8. The results in Tab. 14 show that our method outperforms previous approaches by a margin of 0.15 to 0.89.

*Table 14.* Accuracy (%) comparison of different distillation methods from ResNet50 to VGG8 on CIFAR-100.

| Method | Accuracy (%) |
|---|---|
| KD | 73.81 |
| ABKD (Ours) | 74.62 (0.81) |
| DKD | 74.37 |
| ABDKD (Ours) | **75.26** (0.89) |
| LSD | 74.52 |
| ABLSD (Ours) | 74.77 (0.25) |
| TTM | 74.87 |
| ABTTM (Ours) | 75.02 (0.15) |

## J.2.3. HOW DOES ABKD PERFORM WITH ALPHA/BETA OUTSIDE [0,1]?

Another interesting question is how ABKD performs when $\alpha$ or $\beta$ fall outside the range $[0, 1]$ (*e.g.*, $\alpha = 1.5$, $\beta = -0.5$). To address this concern, we tested the settings with $\alpha > 1$ and $\beta < 0$ when distilling ResNet56 to ResNet20, as shown in Tab. 15.

*Table 15.* Accuracy (%) of ABKD with $\alpha > 1$ and $\beta < 0$ when distilling ResNet56 to ResNet20.

| $\alpha \backslash \beta$ | $-0.1$ | $-0.3$ | $-0.5$ |
|---|---|---|---|
| 1.2 | 70.81 | 71.10 | 70.35 |
| 1.4 | 71.29 | 71.24 | 70.92 |
| 1.6 | 70.55 | 70.53 | 70.34 |

The results indicate that excessively large values of $\alpha$ weaken the hardness-concentration effect, while overly small values of $\beta$ diminish the confidence-concentration effect. Both cases can lead to degraded distillation performance. This observation aligns with our objective: we aim to balance FKLD and RKLD, which correspond to the extreme cases of $\alpha = 1$, $\alpha = 0$, $\beta = 0$, and $\beta = 1$. Therefore, a natural approach is to search for parameters within the range $[0, 1]$.

*Table 16.* Comparison with existing SOTA methods on base-to-new generalization. **Teacher: ViT-L/14 CLIP. Student: ViT-B/16 CLIP.**

| ViT-B/16 | Base | Novel | HM |
|---|---|---|---|
| Teacher | 87.85 | 81.45 | 84.39 |
| CLIP | 69.34 | 74.22 | 71.70 |
| CoCoOp | 80.47 | 71.69 | 75.83 |
| MaPLe | 82.28 | 75.14 | 78.55 |
| PromptSRC | 84.26 | 76.10 | 79.97 |
| KD | 86.96 | 80.73 | 83.63 |
| DKD | 87.02 | 81.02 | 83.79 |
| LSD | 86.31 | 79.99 | 82.89 |
| Ours | **87.27** | **81.41** | **84.17** |

(a) Average over 11 datasets

| ViT-B/16 | Base | Novel | HM |
|---|---|---|---|
| Teacher | 83.24 | 76.83 | 79.91 |
| CLIP | 72.43 | 68.14 | 70.22 |
| CoCoOp | 76.66 | 70.54 | 73.47 |
| MaPLe | 77.00 | 74.05 | 75.49 |
| PromptSRC | 77.63 | 74.97 | 76.26 |
| KD | 80.83 | 74.66 | 77.62 |
| DKD | 80.98 | 74.85 | 77.79 |
| LSD | 80.85 | 74.62 | 77.61 |
| Ours | **81.23** | **75.02** | **78.00** |

(b) ImageNet

| ViT-B/16 | Base | Novel | HM |
|---|---|---|---|
| Teacher | 98.71 | 98.03 | 98.37 |
| CLIP | 96.84 | 94.00 | 95.40 |
| CoCoOp | 97.96 | 93.81 | 95.84 |
| MaPLe | 97.74 | 94.36 | 96.02 |
| PromptSRC | 98.10 | 94.03 | 96.02 |
| KD | 98.91 | 96.65 | 97.77 |
| DKD | 99.12 | 96.52 | 97.80 |
| LSD | 99.05 | 96.24 | 97.62 |
| Ours | **99.46** | **96.93** | **98.18** |

(c) Caltech101

| ViT-B/16 | Base | Novel | HM |
|---|---|---|---|
| Teacher | 96.86 | 98.82 | 97.83 |
| CLIP | 91.17 | 97.26 | 94.12 |
| CoCoOp | 95.23 | 97.69 | 96.43 |
| MaPLe | 95.43 | 97.96 | 96.68 |
| PromptSRC | 95.33 | 97.30 | 96.30 |
| KD | 96.30 | 98.01 | 97.15 |
| DKD | 96.36 | 98.52 | 97.43 |
| LSD | 95.96 | 98.32 | 97.13 |
| Ours | **96.49** | **98.55** | **97.51** |

(d) OxfordPets

| ViT-B/16 | Base | Novel | HM |
|---|---|---|---|
| Teacher | 84.53 | 84.25 | 84.39 |
| CLIP | 63.37 | 74.89 | 68.65 |
| CoCoOp | 70.49 | 73.59 | 72.01 |
| MaPLe | 72.94 | 74.00 | 73.47 |
| PromptSRC | 78.27 | 74.97 | 76.56 |
| KD | 82.80 | 83.37 | 83.13 |
| DKD | 82.23 | **84.20** | 83.21 |
| LSD | 78.29 | 79.48 | 78.88 |
| Ours | **83.43** | 84.01 | **83.72** |

(e) StanfordCars

| ViT-B/16 | Base | Novel | HM |
|---|---|---|---|
| Teacher | 99.05 | 82.60 | 90.08 |
| CLIP | 72.08 | 77.80 | 74.83 |
| CoCoOp | 94.47 | 71.75 | 81.01 |
| MaPLe | 95.92 | 72.64 | 82.56 |
| PromptSRC | 98.02 | 76.50 | 85.92 |
| KD | **99.42** | 82.62 | 90.24 |
| DKD | 99.15 | 82.64 | 90.15 |
| LSD | 98.86 | 81.84 | 89.55 |
| Ours | 99.24 | **83.47** | **90.67** |

(f) Flowers102

| ViT-B/16 | Base | Novel | HM |
|---|---|---|---|
| Teacher | 94.56 | 95.15 | 94.85 |
| CLIP | 90.10 | 91.22 | 90.66 |
| CoCoOp | 90.70 | 91.29 | 90.99 |
| MaPLe | 90.91 | 91.25 | 91.08 |
| PromptSRC | 90.67 | 91.53 | 91.10 |
| KD | 92.43 | 93.68 | 93.05 |
| DKD | 92.35 | 93.72 | 93.03 |
| LSD | 92.07 | 93.07 | 92.57 |
| Ours | **92.46** | **93.84** | **93.14** |

(g) Food101

| ViT-B/16 | Base | Novel | HM |
|---|---|---|---|
| Teacher | 54.44 | 43.07 | 48.09 |
| CLIP | 27.19 | 36.29 | 31.09 |
| CoCoOp | 33.41 | 23.71 | 27.74 |
| MaPLe | 37.44 | 35.41 | 36.50 |
| PromptSRC | 42.73 | 37.87 | 40.15 |
| KD | **49.12** | 41.81 | 45.17 |
| DKD | 48.92 | 42.43 | 45.44 |
| LSD | 47.76 | 39.84 | 43.44 |
| Ours | 49.06 | **43.05** | **45.86** |

(h) FGVCAircraft

| ViT-B/16 | Base | Novel | HM |
|---|---|---|---|
| Teacher | 84.97 | 81.09 | 82.98 |
| CLIP | 69.36 | 75.35 | 72.23 |
| CoCoOp | 79.74 | 76.86 | 78.27 |
| MaPLe | 82.88 | 78.70 | 80.75 |
| PromptSRC | 82.67 | 78.47 | 80.52 |
| KD | 83.69 | 81.54 | 82.60 |
| DKD | 83.87 | 81.32 | 82.58 |
| LSD | 83.34 | 80.62 | 81.96 |
| Ours | **83.88** | **81.75** | **82.80** |

(i) SUN397

| ViT-B/16 | Base | Novel | HM |
|---|---|---|---|
| Teacher | 85.76 | 70.65 | 77.48 |
| CLIP | 53.24 | 59.90 | 56.37 |
| CoCoOp | 77.01 | 56.00 | 64.85 |
| MaPLe | 80.36 | 59.18 | 68.16 |
| PromptSRC | 83.37 | 62.97 | 71.75 |
| KD | 85.84 | 71.37 | 77.94 |
| DKD | **86.83** | 71.91 | 78.67 |
| LSD | 86.35 | 70.97 | 77.91 |
| Ours | 86.52 | **72.65** | **78.98** |

(j) DTD

| ViT-B/16 | Base | Novel | HM |
|---|---|---|---|
| Teacher | 94.79 | 83.15 | 88.59 |
| CLIP | 56.48 | 64.05 | 60.03 |
| CoCoOp | 90.89 | 71.64 | 80.11 |
| MaPLe | 93.32 | 71.26 | 80.94 |
| PromptSRC | 92.90 | 73.90 | 82.32 |
| KD | 97.54 | 82.08 | 89.14 |
| DKD | 97.14 | 83.72 | 89.93 |
| LSD | 97.31 | 83.31 | 89.77 |
| Ours | **97.75** | **83.92** | **90.30** |

(k) EuroSAT

| ViT-B/16 | Base | Novel | HM |
|---|---|---|---|
| Teacher | 89.50 | 82.26 | 85.73 |
| CLIP | 70.53 | 77.50 | 73.85 |
| CoCoOp | 82.33 | 74.95 | 78.47 |
| MaPLe | 83.08 | 78.60 | 80.76 |
| PromptSRC | 87.10 | 78.01 | 82.30 |
| KD | 89.71 | 82.27 | 86.10 |
| DKD | 90.28 | 81.40 | 85.61 |
| LSD | 89.62 | 81.53 | 85.38 |
| Ours | **90.44** | **82.31** | **86.73** |

(l) UCF101

*Table 17.* Instruction-following cases from the Unnatural (Honovich et al., 2023) dataset. ABKD is better at generating responses that meet the requirements of the given instructions.

| | Case # 1 |
|---|---|
| Instruction | In this task, you're given a short description of an event. Your job is to order the steps involved in the event from first to last. Note that there may be multiple correct answers for each event. |
| Input | Description: A man walks into a bar and orders a drink. He pays for his drink and leaves the bar. |
| Ground-truth | The man walks into the bar.\n He orders a drink. \n He pays for his drink.\n He leaves the bar. |
| SFT | A man walks into a bar, asks a bartender who is in the bar and drinks. |
| KD | A man walks into a bar, pays for his drink, and leaves the bar with a short description of the event. \n \n A woman walks into a coffee shop, opens the door, and asks the bartender for him to order them. \n \n A man asks the bartender for their order and alcohol. The woman hands the coffee, and begins making her coffee. At the end, the men exit, and the woman returns to the bar. |
| SeqKD | A man walks into a bar with a drink and leaves the bar; he pays for his drink and enters the bar. |
| MiniLLM | The event was a man walking into a bar and ordering a drink. |
| GKD | A man walks into a bar. He paid for his drink and leaves The bar. |
| DISTILLM | A man walks into a bar, orders a drink and leaves the bar |
| Ours (ABKD) | A man walks into a bar & orders a drink. He pays for his drink & leaves the bar. |
| | Case # 2 |
| Instruction | You need to output the text of the passage after capitalizing all proper nouns. |
| Input | Rami went to new york city on business. While he was there, he met his good friend ali, who showed him around town. |
| Ground-truth | Rami went to New York City on business. While he was there, he met his good friend Ali, who showed him around town. |
| SFT | Rami went to new yORK city on business. While she was there, he met His good friend ali, who shows him around town. |
| KD | Rami went to new yORK city on business. While she was there, he met His good friend ali, who taught him how to go city. |
| SeqKD | Rami went to new yorks city on business. While being there, he met his bad friend ali, who showed you around town. |
| MiniLLM | Rami went to new yorks city on business. While being there, he met his friend ali, who showed he around town. |
| GKD | Rami went to new yORK city on business. While there, he met his bad friend Ali, who showed him around Town. |
| DISTILLM | York City, Holds: ali, who showed him that you are a good friend. |
| Ours (ABKD) | Rami went to New York Citys on business. While he was there, he met her good friend Ali, who shown him around town. |
| | Case # 3 |
| Instruction | You will be given a text consisting of multiple sentences. The task is to find the number of questions present in the text, and then print them out separately. A question is defined as a sentence that ends with a question mark ('?'). |
| Input | Text: I was born on October 3rd, 1984. Do you know when you were born? What month are we in currently? |
| Ground-truth | Do you know when you were born? \n What month are we in currently? |
| SFT | I was born on October 3, 1984. What year are we in currently? |
| KD | I was born on October 3nd, 1984. Do you actually know? I'm not sure because I don't know when I was born. |
| SeqKD | You are in currently. |
| MiniLLM | The text of the question is: I was born on September 3rd, 1984. |
| GKD | I am born on October 3rd 1984. Do you know what month are we in currently, year? |
| DISTILLM | The task is to find the answer to a question that begins with a question mark ('!'). The task is to find the answer to a question that begins with a question mark ('!'). The task is to find the answer to a question that begins with a question mark ('!'). |
| Ours (ABKD) | I was born on October 3nd, 1984. Do you remember when you were born? |

