# OpenReview forum: "ABKD: Pursuing a Proper Allocation of the Probability Mass in Knowledge Distillation via $\alpha$-$\beta$-Divergence"
_ICML.cc/2025/Conference — ICML 2025 oral_

### Official Review · Reviewer_VNjn · 2025-03-08

**Overall Recommendation:** 5

**Summary:**

This paper investigates a fundamental challenge in Knowledge Distillation (KD): the improper allocation of probability mass when using traditional divergences like Forward KL Divergence (FKLD) and Reverse KL Divergence (RKLD). FKLD tends to spread probability mass too broadly, failing to pay sufficient attention to the target class, while RKLD overly concentrates on the target class, neglecting the broader distributional information from the teacher model. The authors analyze this issue from the view of two effects:  hardness-concentration, which focuses on modes where the student model has high error, and confidence-concentration, which emphasizes modes where the student is already confident. To better balance the two effects, the paper introduces $\alpha$-$\beta$-Divergence (ABKD), which generalizes FKLD, RKLD, and other divergences, thus providing a better trade-off between the two effects. Theoretical results demonstrate that ABKD provides a more balanced allocation of probability mass, leading to improved student learning. Finally, extensive experiments across 17 language and vision datasets with 12 teacher-student model pairs validate its effectiveness.

**Claims And Evidence:**

The claims in the submission are well-supported by both theoretical analysis and empirical results:

- Theoretical justifications explain the limitations of FKLD and RKLD and demonstrate how ABKD provides a more flexible probability mass allocation.
- Empirical evaluations across 17 language and vision datasets with 12 teacher-student configurations further validate these claims, showing consistent performance improvements over existing methods.

**Essential References Not Discussed:**

To my best of knowledge, this paper has covered essential references in this field.

**Experimental Designs Or Analyses:**

Yes, please refer to `Methods And Evaluation Criteria`.

**Methods And Evaluation Criteria:**

Yes. The evaluation protocol is comprehensive, spanning 17 language/vision datasets with 12 teacher-student configurations, ensuring that the findings generalize across different settings. The selection of benchmarks, including instruction-following datasets for NLP and classification datasets for vision tasks, is appropriate for assessing the effectiveness of distillation methods. Moreover, the proposed method is compared against the state-of-the-art KD techniques, also providing meaningful results.

**Other Comments Or Suggestions:**

Please refer to `weankess`.

**Other Strengths And Weaknesses:**

The strengths of this paper are summarized as follows:

- **Rigorous theoretical insights**: This paper provides a rigorous theoretical analysis of the limitations of traditional divergences widely used in KD. The key idea behind is to reveal how these divergences allocate probability mass during training. By introducing the concepts of hardness-concentration and confidence-concentration, the authors show that FKLD and RKLD represent two extreme cases: FKLD fails to concentrate on the target class and RKLD overly concentrates on the target class. The proposed $\alpha$-$\beta$-Divergence (ABKD) balances the two effects, offering a novel perspective on why model-based guidance (i.e., distilling from a teacher model) can outperform hard label guidance (i.e., learning from one-hot labels).

- **Flexibile framework based on $\alpha$-$\beta$-divergence**: The proposed framework, based on $\alpha$-$\beta$-divergence, unifies existing divergence-based KD approaches. This flexibility allows for fine-grained control over probability mass allocation, leading to improved performance.

- **Comprehensive empirical validation**: Extensive experiments are conducted across 17 language/vision datasets with 12 teacher-student model configurations. The results demonstrate consistent improvements over state-of-the-art KD methods.

My minor concerns are as follows:

- In Sec 5.3, the authors provide empirical justifications for the choice of $\alpha$ and $\beta$. Although the results are valuable for hyperparameter tuning, I wonder whether the observation is consistent across different datasets. I hope the author can provide more clues on some other datasets, which could further strengthen the practical applicability of ABKD.

**Questions For Authors:**

Please refer to `weankess`.

**Relation To Broader Scientific Literature:**

Nowadays, knowledge distillation is an important topic due to the development of the large-scale foundation model. This paper investigates a foundational issue in the field of knowledge distillation. Hence, I believe it could relate to broader literature in the future.

**Theoretical Claims:**

Yes. I have carefully checked the theoretical claims presented in the paper, particularly ABKD's role in balancing the two effects.

---

> ### Author Rebuttal · Authors · 2025-04-01
>
> We sincerely thank the reviewer for recognizing the theoretical foundations, clarity of contributions and experiments, and the improved performance demonstrated by our method. Our response follows:
>
> > Q1: I wonder whether the observation is consistent across different datasets. I hope the author can provide more clues on some other datasets, which could further strengthen the practical applicability of ABKD
>
> **A1**: Thank you again for your insightful question. The table below outlines the hyperparam settings across datasets, which generally align with the theoretical results in Sec.3 (e.g., small α and large β for language modeling tasks and large α and small β for image classification tasks).
>
> |GPT-2 XL (1.5B) -> GPT-2 (0.1B, 0.3B, 0.8B) | Dolly Eval | Self-Instruct|Vicuna Eval|Super-Natural|Unnatural|
> |-|-|-|-|-|-|
> | **α**  | 0.2        | 0.2         | 0.2        | 0.2          | 0.2         |
> | **β**  | 0.7        | 0.7         | 0.7        | 0.7          | 0.7         |
>
>
> | **Dataset**      | ImageNet   | Caltech101  | OxfordPets | StanfordCars | Flowers102  | Food101    | FGVCAircraft | SUN397   | DTD   | EuroSAT | UCF101  |
> |-|-|-|-|-|-|-|-|-|-|-|-|
> | **α**  | 0.5        | 0.8         | 0.8        | 0.6          | 0.9         | 0.5        | 0.6          | 0.8      | 1.0   | 0.6     | 0.8     |
> | **β**  | 0.5        | 0.2         | 0.4        | 0.4          | 0.1         | 0.5        | 0.5          | 0.2      | 0.2   | 0.5     | 0.2     |
>
> To further support our claim and validate the effects of our method, we made our best effort in the past few days to study **OpenLLaMA-8B→3B** distillation. For a thorough comparison, we also considered several baselines mentioned by other reviewers (i.e., AlphaNet, BDKD, Jensen's KL, and AKL) during the rebuttal period.
>
> Based on the hyperparam tuning guidelines (App.D) derived from theoretical guidance, we simply set the hyperparams to α = 0.3 and β = 0.6 for all datasets (without further tuning).
>
> The results below show that our method outperforms others by 0.65-3.26, especially excelling in Dolly and Unnatural.
>
> | Method    | Dolly Eval | Self-Instruct | Vicuna Eval | Super-Natural | Unnatural |
> |-----------|-----------|---------------|-------------|---------------|-----------|
> | SFT       | 24.54 (0.51) | 16.80 (0.64) | 16.15 (0.15) | 29.29 (0.13) | 27.43 (0.21) |
> | FKLD        | 25.23 (0.44) | 18.90 (1.20) | 16.67 (0.35) | 31.68 (0.22) | 29.36 (0.13) |
> | RKLD   | 27.74 (0.45) | 20.61 (0.80) | 18.83 (0.40) | 35.31 (0.24) | 33.86 (0.16) |
> | Jensen's KL       | 26.28 (0.43) | 18.84 (0.66) | 17.81 (0.38) | 30.92 (0.12) | 29.79 (0.17) |
> | BDKD       | 26.78 (0.53) | 18.94 (0.68) | 17.81 (0.52) | 32.15 (0.34) | 30.89 (0.24)  |
> | AKL       | 26.38 (0.41) | 17.69 (0.46) | 16.72 (0.48) | 33.02 (0.16) | 31.29 (0.08) |
> | DISTILLM  | 28.24 (0.48) | 21.00 (0.72) | 19.12 (0.53) | 37.06 (0.35) | 35.05 (0.13) |
> | AlphaNet  | 28.11 (0.29) | 21.30 (0.63) | 18.70 (0.23) | 37.86 (0.44) | 35.40 (0.17) |
> | Ours (ABKD) | **30.25** (0.37) | **22.39** (0.62) | **20.83** (0.42) | **38.51** (0.32) | **38.66** (0.10) |

---

> > ### Comment · Reviewer_VNjn · 2025-04-05
> >
> > The authors have addressed my concerns on hyperparameters, and the new experiments on the  OpenLLaMA-8B→3B distillation is also impressive. I'll raise my score.

---

> > > ### Author Response · Authors · 2025-04-05
> > >
> > > Thank you very much for your recognition of our work and for the improved score. We will do our best to further enrich and improve the final version of the article.

---

### Official Review · Reviewer_Ty7Q · 2025-03-13

**Overall Recommendation:** 4

**Summary:**

The paper introduces ABKD, a knowledge distillation (KD) framework using alpha-beta-divergence to balance the "hardness-concentration" (focus on high-error classes) and "confidence-concentration" (focus on high-confidence classes) effects. Theoretical analysis shows that FKLD and RKLD represent extreme cases of these effects, leading to suboptimal performance. ABKD generalizes these divergences, enabling smooth interpolation via alpha and beta.

**Claims And Evidence:**

Supported Claims:
Balancing mode-concentration effects (Sec. 3–4): Theoretical analysis (Prop. 3.1, 4.2) and visualizations (Fig. 1d–g) justify the limitations of FKLD/RKLD and ABKD’s trade-off mechanism.
Potential Issues:
The paper claims that ABKD encompasses FKLD (when alpha=1, beta=0) and RKLD (when alpha=0, beta=1), and thus the experiments should include a comparison of the performance of ABKD in these degenerate cases with the original FKLD and RKLD.

**Essential References Not Discussed:**

Wu, T., Tao, C., Wang, J., Zhao, Z., & Wong, N. (2024). Rethinking Kullback-Leibler Divergence in Knowledge Distillation for Large Language Models. arXiv.Org. https://doi.org/10.48550/arxiv.2404.02657
Cui, X., Qin, Y., Gao, Y., Zhang, E., Xu, Z., Wu, T., Li, K., Sun, X., Zhou, W., & Li, H. (2024). SinKD: Sinkhorn Distance Minimization for Knowledge Distillation. IEEE Transactions on Neural Networks and Learning Systems, 1–15. https://doi.org/10.1109/tnnls.2024.3501335
These papers are representative works that improve distillation effects based on different loss distance calculations, but they are not cited in this paper. The adaptive KL divergence(AKL) proposed in the former is more suitable than the WSD proposed in this paper as a baseline.

**Experimental Designs Or Analyses:**

Hyperparameter Sensitivity: Fig. 6 shows alpha and beta impact entropy and Self-BLEU, but optimal values vary across tasks (e.g., alpha=0.2, beta=0.7 for NLP vs. alpha=0.6, beta=0.5 for CIFAR-100 in Tabs. 4,6). This suggests task-specific tuning, weakening the "universal" claim.
Moreover, Table 4 in Appendix I.2.3 also shows that ABKD requires specific hyperparameter selection, which limits its scalability to more datasets and architectures.

**Methods And Evaluation Criteria:**

17 datasets and 12 teacher-student pairs (e.g., GPT-2 XL→GPT-2) are reasonable. Metrics (ROUGE-L, accuracy) align with standard practice.
However, the validation on the CIFAR100 dataset only involves cases where the teacher and student have the same architecture; the performance when they have different architectures should also be examined.

**Other Comments Or Suggestions:**

If you have any other comments or suggestions (e.g., a list of typos), please write them here. Clarity: The paper is well-structured, but theoretical sections (Sec. 3–4) are dense.

**Other Strengths And Weaknesses:**

Strengths:
Theoretical-empirical synergy: Clear connection between gradient analysis (Sec. 3) and empirical results.
Weaknesses:
Baseline Variance: Some baselines (e.g., LSD, TTM) underperform in Fig. 5; unclear if hyperparameters were optimized.

**Questions For Authors:**

Q1: How does ABKD perform with alpha/beta outside [0,1] (e.g., alpha=1.5, beta=-0.5)? Does the framework still hold? (Clarifies generality claims.)
Q2: Were computational costs (e.g., GPU hours) comparable between ABKD and baselines? (Addresses scalability concerns.)

**Relation To Broader Scientific Literature:**

Builds on classical KD (Hinton, 2015) and recent divergence variants (MiniLLM, DISTILLM). The alpha-beta-divergence generalizes prior work (Table 1), addressing gaps in FKLD/RKLD trade-offs.

**Theoretical Claims:**

Theorem 3.2(mass allocation differences): Informal statement lacks rigor; formal proof needs verification.

---

> ### Author Rebuttal · Authors · 2025-04-01
>
> We thank the reviewer for the thoughtful comments and suggestions. Our response follows:
> > Q1: The experiments should include a comparison of the performance of ABKD in these degenerate cases with the original FKLD and RKLD
>
> **A1**: When ABKD degenerates to FKLD and RKLD, its performance matches the original FKLD and RKLD (Tab.3 in the main text). Thus, we don't separately consider these cases.
> > Q2: Theorem 3.2: Informal statement lacks rigor; formal proof needs verification
>
> **A2**: Thank you for your question! The complete version of Thm.3.2 is in App.G.2. We will include them in the main text and improve readability upon acceptance.
> > Q3: This suggests task-specific tuning, challenging the "universal" claim. Tab.4 shows that ABKD requires specific hyperparam selection, limiting its scalability to additional datasets and architectures
>
> **A3**: It's a pity that the term 'universal' is misunderstood. Actually, we aim to unify a series of existing divergences in KD, not provide a set of task-agnostic parameters. In fact, most related works require parameter tuning for different datasets and architectures. As per the 'no free lunch' theorem, universal parameters are impractical. Please see A4 of Reviewer 71pK for more details on 'universal'.
>
> To further support our claim, we completed the distillation experiment from **OpenLLaMA-8B to 3B** within the limited time, including AKL mentioned by the reviewer, as well as BDKD, AlphaNet, and Jensen's KL mentioned by other reviewers. The results below show that our method outperforms others by 0.65-3.26, especially excelling in Dolly and Unnatural.
>
> ||Dolly|Self-Instruct|Vicuna|Super-Natural|Unnatural|
> |-|-|-|-|-|-|
> |SFT|24.54|16.80|16.15|29.29|27.43|
> |FKLD|25.23|18.90|16.67|31.68|29.36|
> |RKLD|27.74|20.61|18.83|35.31|33.86|
> |Jensen's KL|26.28|18.84|17.81|30.92|29.79|
> |BDKD|26.78|18.94|17.81|32.15|30.89|
> |AKL|26.38|17.69|16.72|33.02| 31.29|
> |DISTILLM|28.24|21.00|19.12|37.06|35.05|
> |AlphaNet|28.11|21.30|18.70|37.86|35.40|
> |Ours|**30.25**|**22.39**|**20.83**|**38.51**|**38.66**|
> > Q4: The Need for Ablation on Model Architectures.
>
> **A4**: Our method applies to various architectures (e.g., 17 teacher-student pairs used in our experiments). It only needs adjusting the distillation objective. We also provide an ablation study of α-β divergence (Tab.3 in the main text), which shows using only α or β imposes unnecessary constraints on hardness- or confidence-concentration, leading to suboptimal solutions.
>
> > Q5: Essential References Not Discussed
>
> **A5**: Thank you for your valuable insight! We will cite these works in the final version and discuss the differences between AKL and our work.
> We also conducted experiments comparing AKL with our method **when distilling GPT-2 XL into GPT-2**, as shown below. Our method outperforms AKL across datasets by 0.43-7.35.
>
> ||Dolly|Self-Instruct|Vicuna|Super-Natural|Unnatural|
> |-|-|-|-|-|-|
> |AKL|23.83|10.87|15.63|20.07|21.97|
> |Ours|**25.65**|**13.47**|**16.06**|**26.47**|**29.32**|
> > Q6: Baseline Variance: Some baselines (e.g., LSD, TTM) underperform in Fig.5; unclear if hyperparams were optimized.
>
> **A6**: All baseline results in Fig.5 are from the original papers. Missing values are obtained by rerunning their code with necessary hyperparam tuning and reporting the average of 3 random runs.
> > Q7: Cross-Architecture Experiment
>
> **A7**: We conducted distillation from ResNet50 to VGG8 within the limited time. The results below show that our method can improve the performance of previous methods by 0.15-0.89.
> ||KD|ABKD (Ours)|DKD|ABDKD (Ours)|LSD|ABLSD (Ours)|TTM|ABTTM (Ours)|
> |-|-|-|-|-|-|-|-|-|
> |Accuracy|73.81|74.62|74.37|**75.26**|74.52|74.77|74.87|75.02|
>
> > **Q8**: How does ABKD perform with α/β outside [0,1] (e.g., α=1.5, β=-0.5)?
>
> **A8**: This is an interesting question. We focus on balancing FKLD and RKLD, which correspond to extreme cases for α = 1, α = 0, β = 0, and β = 1. Thus, an intuitive method is to search for params between [0, 1]. Our experiments validate this. To address the reviewers' concern, we also tested α > 1 and β < 0 when distilling ResNet56 to ResNet20 within the limited time, as shown below:
>
> |α\β|-0.1|-0.3|-0.5|
> |-|-|-|-|
> |1.2 |70.81|71.10|70.35 |
> |1.4|71.29| 71.24| 70.92|
> |1.6|70.55|70.53|70.34|
>
> An overly large α weakens the hardness-concentration and an overly small β weakens the confidence-concentration effect, both may degrade distillation performance.
> > Q9: Computational Cost Analysis
>
> **A9**: We compared the training costs of different methods on language modeling, as shown below.
>
> ||Training Cost (second/sample)|
> |-|-|
> |SFT|0.344|
> |KD|0.649|
> |MiniLLM|4.452|
> |GKD|2.078|
> |DISTILLM|1.331|
> |AlphaNet|0.882|
> |Ours|0.768|
>
> Our method takes a similar amount of time as Vanilla KD but is **1.15x to 5.80x faster** than others due to its simplicity. It only modifies the optimization objective without adding extra cost, while others like GKD and DISTILLM require sampling student outputs during training.

---

### Official Review · Reviewer_71pK · 2025-03-14

**Overall Recommendation:** 4

**Summary:**

The paper discusses the main challenges in knowledge distillation, which lies under the proper balance between two modes (1) hardness concentration and (2)confidence concentration. They provided a smoother transition between the reverse and forward KL divergences via the integration of alpha-beta divergence They performed analyses on both language and vision tasks.

**Claims And Evidence:**

- The claim of using alpha-beta divergence for better distillation seems to ambiguous and not empirically well supported in the paper.
Indeed, introducing both alpha and beta as hyper-parameters adds more complexity to the already challenging problem of distillation.
Different alpha and beta can lead to very different distillation behavior. This could eventually impact the stability and the convergence of the student network while performing distillation.

**Essential References Not Discussed:**

Some key references to consider are earlier works that explored the use of Jensen-KL (symmetric KL) for knowledge distillation. [1]

Additionally, there are prior studies on adaptive KL divergence, which, similar to the idea presented in this paper, aim to balance forward KL (FKLD) and reverse KL (RKLD). A proper discussion and citation of these works would provide better context and highlight the connections between this study and existing research. [2]


[1] Binici, Kuluhan, et al. "Preventing catastrophic forgetting and distribution mismatch in knowledge distillation via synthetic data." Proceedings of the IEEE/CVF winter conference on applications of computer vision. 2022.

[2] Amara, Ibtihel, et al. "Bd-kd: balancing the divergences for online knowledge distillation." arXiv preprint arXiv:2212.12965 (2022).

**Experimental Designs Or Analyses:**

The experimental design and the analyses is somewhat comprehensive. The authors tried to empirically compare their proposed technique alpha-beta KD to other KD techniques. However, I would like to address certain comments with respect to the experiments:
- There are some missing baselines. Before balancing the divergences, previous literature have proposed to implement Jensen's KL divergence in which they symmetrically use both the RKLD and FKLD. Adding this as a baseline in their experiments could contrast the performance of the distillation.
- With respect to analyses, as heavy user of distillation, I would like to see the computational and complexity trade-off that the alpha-beta KD offers. From the results of the paper, the reported improvements over baselines like vanilla KD and other distillation methods appear to be marginal or even lower than that of alternative KD methods. This really raises concerns about the added complexity and actual performance gains. It would have been better if the authors could add other analyses and justification of why this method would be preferable despite the noticeable limited improvements.

**Methods And Evaluation Criteria:**

The paper evaluates their technique on both language and vision tasks, which I think is good and makes their work more comprehensive.

**Other Comments Or Suggestions:**

Please refer to previous cell.

**Other Strengths And Weaknesses:**

I would like to summarize what I have mentioned above:
Strengths:
- The paper is well written and easy to follow.
- The paper evaluates on various language and vision tasks.

Weaknesses:
- It would have been better if the authors could justify more about the hyper-parameters of alpha and beta. While the authors specify the values they used in their experiments, it remains unclear how extensively they tuned these parameters to achieve optimal performance.
- Another key aspect missing is the discussion on training time. Based on my expertise, training with alpha divergence tends to have slower convergence compared to traditional KD losses. Aside from the added complexity of alpha and beta, it would be interesting to see insights into the computational cost and the convergence behavior especially for language model training.
- Questionable novelty. The key contributions of the paper closely resembles existing techniques that have explored using different divergences for distillation (similar to the alphanet paper, which the authors have cited). The proposed approach does not introduce a fundamentally novel concept. I would suggest to specify in the paper a more thorough comparison with prior work and clearly stating what would be the unique contribution.
- Baselines in the experiments. I encourage the authors to add more KD baselines like the ones referenced above.

**Questions For Authors:**

Please refer to the strengths and weaknesses section.

**Relation To Broader Scientific Literature:**

The key contribution of the paper closely resembles existing techniques that have already been explored and discussed in earlier literature on knowledge distillation. The proposed approach does not introduce a fundamentally novel concept but rather builds upon previously established methods.

**Theoretical Claims:**

Yes. I checked the theoretical proofs. I have not noticed any issues.

---

> ### Author Rebuttal · Authors · 2025-04-01
>
> We thank the reviewer for taking the time to provide their helpful and valuable feedback. Our response follows (**please see https://anonymous.4open.science/r/ICML-rebuttal-experiments/results.md for all rebuttal experiments**):
> >Q1: Essential References Not Discussed
>
> **A1:** We will include the discussion of these baselines (the experiments are in Q5) in the new version:
>
> - Jensen's KL is a popular method that combines FKLD and RKLD: $D_{JSD}(p|q)=1/2D_{KL}(p|m)+1/2D_{KL}(q|m)$, where $p$ is teacher distribution, $q$ is student distribution and $m=(p+q)/2$.
>
> - BDKD adjusts the weights of FKLD and RKLD during training based on the entropy difference between $p$ and $q$.
>
> - We also note AKL mentioned by the reviewer Reviewer Ty7Q, which adjusts the weights of FKLD and RKLD based on class differences between $p$ and $q$.
>
> The problem with Jensen's KL is that when $p$ and $q$ are far apart, the loss becomes constant, causing vanishing gradients and hindering convergence. BDKD and AKL can be seen as variants of our baseline, WSD. However, they still tend to overemphasize small probability in $p$ and $q$, as shown in Sec.3.3.
>
> Overall, a simple linear addition of FKLD and RKLD can't fully resolve their issues. This work aims to first theoretically analyze their limitations and then explore a more effective divergence to address them.
> >Q2: More Details on hyperparams α and β (Weakness #1)
>
> **A2:** Thank you for your insightful question! We agree on the importance of clarifying hyperparam selection.
>
> First, our method needs comparable or fewer hyperparams than prior works: AlphaNet has 3 (ICML 21), DISTILLM has 2 (ICML 24), and GKD has 2 (ICLR 24).
>
> Second, while additional hyperparams increase search cost, **our theoretical insights (Secs.3, 4) provide tuning guidelines for different tasks (App.D)**. These insights help eliminate less likely hyperparams, improving search efficiency. Empirical hyperparam selections (e.g., α=0.2, β=0.7 for NLP and α=0.9, β=0.2 for vision) validate these.
>
> Overall, our search cost is not higher than prior works, and our hyperparams have clear theoretical and practical significance, helping one better understand the effectiveness of their distillation objective.
> >Q3: Training Cost and Convergence Behavior (Weakness #2)
>
>  **A3**: Thank you for your insightful question!
>
>  **Training Cost**: Tab.1 in the link shows our method requires a similar time to Vanilla KD, but is **1.15x to 5.80x faster** than other SOTA methods, as it only modifies the distillation loss, while others need sampling student outputs during training, adding extra computational cost.
>
>  **Convergence**: Fig.1 in the link shows that our method outperforms others at all training stages (**especially in the early stages**), showing **faster convergence**.
> >Q4: Novelty issue (Weakness #3)
>
> **A4**: Our main contribution lies in unifying FKLD, RKLD, and other potential variants from **a novel theoretical perspective: the hardness- and confidence-concentration effects**. This new theory helps explain why traditional divergences fail and our method succeeds:
>
> - FKLD has overly weak hardness- and confidence-concentration effects. It fails to focus on the target class, leading to wrong predictions.
> - RKLD has overly strong hardness- and confidence-concentration effects. It overemphasizes the target class while ignoring the broader knowledge from non-target classes.
> - The weighted sum of FKLD and RKLD tends to overemphasize the minimal values in both teacher and student distributions.
> - α-divergence imposes an unnecessary sum-to-one constraint on hardness- and confidence-concentration effects, potentially limiting model performance.
> - α-β-divergence offers a flexible way to adjust hardness- and confidence-concentration separately, enabling smooth interpolation between FKLD and RKLD.
>
> These insights are rarely explored in existing literature.
>
> > Q5: Performance Improvement & More Baselines (Weakness #4)
>
> **A5:** Thank you for your valuable suggestion! We added more baselines (the suggested AlphaNet, BDKD, Jensen's KL, and AKL mentioned by other reviewers) with the limited time. Results are averaged over 5 seeds.
> ||Dolly|Self-Instruct|Vicuna|Super-Natural|Unnatural|
> |-|-|-|-|-|-|
> |Prior SOTA|25.13 (0.27)|12.46 (0.46)|15.64 (0.40)|25.27 (0.20)|27.56 (0.15)|
> |Ours|**25.65** (0.24)|**13.47** (0.42)|**16.06** (0.25)|**26.47** (0.31)|**29.32** (0.08)|
>
> Tab.2 (see the link for the full version) shows our method outperforms others by 0.42-1.76.
>
> To further validate our method, we made our best effort in the past few days to study **OpenLLaMA-8B→3B** distillation.
> ||Dolly|Self-Instruct|Vicuna|Super-Natural|Unnatural|
> |-|-|-|-|-|-|
> |Prior SOTA|28.24 (0.48)|21.30 (0.63)|19.12 (0.53)	|37.86 (0.44)|35.40 (0.17)|
> |Ours|**30.25** (0.37)|**22.39** (0.62)|**20.83** (0.42)|**38.51** (0.32)|**38.66** (0.10)|
>
> Tab.3 (see the link for the full version) shows it outperforms others by 0.65-3.26, especially excelling in Dolly and Unnatural.

---

> > ### Comment · Reviewer_71pK · 2025-04-04
> >
> > Thank you for your thorough response and the valuable clarifications provided.
> >
> > I have carefully reviewed your replies, along with the feedback from the other reviewers, and I am particularly appreciative of the expanded comparisons with baseline models. This additional context has significantly strengthened the manuscript.
> >
> > Consequently, I am pleased to revise my recommendation **from a weak reject to an accept**.
> >
> > However, to ensure the manuscript's completeness and maximize its impact, **it is crucial that all benchmark comparisons presented in your responses to the reviewers are incorporated into the final version of the paper**. This will provide a comprehensive and transparent evaluation of your method's performance and address the concerns raised by all reviewers.
> >
> > Again, thank you for your efforts in addressing the feedback. I believe these revisions will greatly enhance the quality and clarity of your work.

---

> > > ### Author Response · Authors · 2025-04-04
> > >
> > > Thank you so much for your valuable feedback and support. Following your suggestions, we will strive to further enrich the content of our paper in the final version to contribute to the continued development of the KD community.

---

### Decision · Program_Chairs · 2025-05-01

**Decision:**

Accept (oral)

**Comment:**

Reviewers were unanimously positive in their assessment of the paper, which studies popular KL and reverse-KL based distillation objectives through a unified framework. The framework highlights the importance of Hardness- and Confidence-Concentration, which motivates a proposed $(\alpha,\beta)$-divergence based objective. The paper demonstrates results on a range of language and vision settings, with several model families.

The author response included some new results comparing the method to a further set of baselines, including the Jensen-Shannon divergence. These results are strongly encouraged to feature in the final submission.

Overall, the paper is well written, studies an important problem, and makes interesting contributions that are likely of interest to the community. Thus, we believe it is worthy of presentation at the conference.